# Excess PrP^C inhibits muscle cell differentiation via miRNA-enhanced liquid–liquid phase separation implicated in myopathy

Jing Tao [1], Yanping Zeng[2], Bin Dai[1], Yin Liu[2], Xiaohan Pan[3], Li-Qiang Wang [1,4], Jie Chen[1], Yu Zhou [1], Zuneng Lu[2], Liwei Xie [3] & Yi Liang [1,4] ✉

The cellular prion protein (PrP^C) is required for skeletal muscle function. Here, we report that a higher level of PrP^C accumulates in the cytoplasm of the skeletal muscle of six myopathy patients compared to controls. PrP^C inhibits skeletal muscle cell autophagy, and blocks myoblast differentiation. PrP^C selectively binds to a subset of miRNAs during myoblast differentiation, and the colocalization of PrP^C and miR-214-3p was observed in the skeletal muscle of six myopathy patients with excessive PrP^C. We demonstrate that PrP^C is overexpressed in skeletal muscle cells under pathological conditions, inhibits muscle cell differentiation by physically interacting with a subset of miRNAs, and selectively recruits these miRNAs into its phase-separated condensate in living myoblasts, which in turn enhances liquid–liquid phase separation of PrP^C, promotes pathological aggregation of PrP, and results in the inhibition of autophagy-related protein 5-dependent autophagy and muscle bundle formation in myopathy patients characterized by incomplete muscle regeneration.

Mammalian prion protein (PrP) has two forms that are distinct in their structure and function, the cellular prion protein (PrP^C) and its pathological aggregated form PrP^Sc (refs. 1–6). Prion diseases are a group of transmissible neurodegenerative diseases primarily caused by the conformational conversion of PrP from α-helix-dominant PrP^C to β-sheet–rich PrP^Sc in many mammalian species[1–12]. The benign cellular form PrP^C, a glycosylphosphatidylinositol (GPI)-anchored glycoprotein existing in membrane-bound and cytoplasmic forms with multifaceted functions[13,14], has functional importance in skeletal muscle[15–22], such as maintaining normal muscle size and function[15], promoting muscle regeneration[16], and linking to myoblast differentiation[15,18]. PrP^C dysfunction is observed in the skeletal muscle

of patients with inclusion-body myositis, dermatomyositis, and other myopathies[19,20,22] or transgenic mice developing a rapidly progressive primary myopathy[21], but the mechanism behind the phenomenon remains unclear.

Autophagy, an evolutionarily conserved catabolic pathway through lysosomal degradation of intracellular components[23,24], appears to increase and is required during myoblast differentiation[25,26] and related to muscle pathophysiology[27]. Importantly, impaired autophagy is observed in aged muscle satellite cells[28]. Moreover, scientists have identified at least 35 genes encoding autophagy-related (ATG) proteins, including autophagy-related protein 5 (ATG5) and microtubule-associated protein 1 light chain 3 (LC3), among which the

[1]Hubei Key Laboratory of Cell Homeostasis, College of Life Sciences, TaiKang Center for Life and Medical Sciences, Wuhan University, Wuhan 430072, China. [2]Department of Neurology, Renmin Hospital of Wuhan University, Wuhan 430060, China. [3]Guangdong Provincial Key Laboratory of Microbial Culture Collection and Application, State Key Laboratory of Applied Microbiology Southern China, Institute of Microbiology, Guangdong Academy of Sciences, Guangzhou 510070, China. [4]Wuhan University Shenzhen Research Institute, Shenzhen 518057, China. ✉e-mail: liangyi@whu.edu.cn

E3 ubiquitin ligase ATG5 is essential for autophagosome elongation[23,24,29–32]. Currently, pathological prion aggregates have been shown to trigger autophagy in skeletal muscle and can be degraded by autophagy[33]. However, it is unclear whether the benign cellular form of PrP regulates autophagy and differentiation of skeletal muscle cells.

MicroRNAs (miRNAs) are small noncoding RNAs that are highly enriched in skeletal muscle and highly conserved from plants to mammals[34,35]. miRNAs play roles in regulating differentiation, atrophy, and regeneration of skeletal muscle via interaction with specific proteins[35–38]. Three miRNAs, miR-181a-5p, miR-324-5p, and miR-451a, are overexpressed in the skeletal muscle of patients with spinal muscular atrophy, suggesting that targeting the overexpressed miRNAs may be a novel therapeutic approach against the disease[37]. However, it is unclear whether miRNAs regulate autophagy and differentiation of skeletal muscle cells via specific interactions with PrP$^C$.

Protein and RNA molecules tend to form supramolecular assemblies called membrane-less organelles via liquid–liquid phase separation (LLPS) of proteins in cells to perform key functions[39–43]. The liquid droplets formed by biological macromolecules, called biomolecular condensates, have fusion properties[40,44], and RNA may control or buffer LLPS of proteins with natively unfolded and/or low-complexity domains[43–46]. Because PrP$^C$ contains a low-complexity and intrinsically disordered region (IDR) in its N-terminal domain, it undergoes LLPS in vitro and forms protein condensates[47–56]. PrP$^C$ liquid-phase condensation is modulated by amyloid-β oligomers, neutralizing mutations, pathological mutations, RNA (polyU RNA, crude tRNA, and yeast total RNA), and other factors[47–56]. However, it is unclear whether PrP$^C$ undergoes LLPS in vivo and whether the LLPS of PrP$^C$ leads to pathological aggregation of the protein in cells. It also remains unknown whether PrP$^C$ selectively recruits specific miRNAs into phase-separated condensates, which in turn regulates the LLPS of PrP$^C$.

Here, we report that a higher level of PrP$^C$ accumulates in the cytoplasm of the skeletal muscle of myopathy patients compared to controls. We demonstrate that PrP$^C$ located in the cytoplasm blocks muscle cell differentiation by selectively recruiting a subset of miRNAs, including miR-214-3p, into its phase-separated condensate in living myoblasts, which in turn enhances the phase separation and aggregation of PrP and significantly inhibits ATG5-related autophagy. Our findings provide insights into the regulation of PrP$^C$ on muscle cell differentiation under pathological conditions via LLPS of the protein enhanced by a subset of miRNAs, such as miR-214-3p, which has implications for myopathy etiology.

## Results

### Accumulation of PrP$^C$ is observed in the skeletal muscle of myopathy patients

We first took confocal images of frozen skeletal muscle sections from six myopathy patients with dermatomyositis, neurogenic myopathy or muscular atrophy, one healthy control, two controls with lipid storage myopathy, and one control with glycogen storage disease (Figs. 1a and 2a and Supplementary Table 1). The frozen skeletal muscle sections were immunostained with the anti-PrP antibody 8H4 (red), stained with DAPI (blue), and visualized by confocal microscopy. A higher level of PrP$^C$ (red) accumulated in the cytoplasm of the skeletal muscle of the six myopathy patients (Fig. 1a). In sharp contrast, accumulation of PrP$^C$ was not observed in the skeletal muscle of these four controls and the control samples only had normal levels of PrP$^C$ (red) in most fields of view (Fig. 2a). The skeletal muscles were then analyzed by H&E staining and immunohistochemistry using an anti-NCAM antibody (brown) (Figs. 1b, c and 2b, c). H&E staining and immunohistochemical staining of the frozen skeletal muscle sections showed that the skeletal muscle of the six myopathy patients had morphological features of regeneration, such as internalized nuclei (in dashed black loops, Fig. 1b), and expressed high levels of neural cell adhesion molecule

(NCAM) (brown) (Fig. 1c), a marker of muscle regeneration[57]. In the skeletal muscles of the six patients, many centrally nucleated fibers were observed in each sample (Fig. 1b). In sharp contrast, the skeletal muscle of these four controls did not have any morphological features of regeneration (Fig. 2b) and showed lack of expression of NCAM (Fig. 2c). Thus, the skeletal muscle of the six myopathy patients was characterized by incomplete muscle regeneration (cases 2 and 6 with obvious muscular atrophy) but muscle regeneration was not observed in the skeletal muscle of these four controls (Figs. 1b, c and 2b, c). Increased PrP$^C$ expression is seen in inclusion-body myositis, dermatomyositis, and other myopathies[19–22], in agreement with our experimental observations (Fig. 1a).

### Maintenance of PrP$^C$ homeostasis is important for myoblast differentiation

A central hypothesis for PrP$^C$ function is that its overexpression or knockout may impair its function in regulating myoblast differentiation. To test this hypothesis at the cellular level, we used murine-derived C2C12 myoblast cells, which provide a fascinating possibility as myoblasts are proliferative but differentiate into multinucleated, elongated myotubes after serum deprivation[58]. Moreover, PrP$^C$ expression is upregulated during myoblast differentiation[59]. C2C12 myoblasts stably overexpressing full-length wild-type (WT) mouse PrP$^C$, C2C12 myoblasts stably overexpressing F198S PrP$^C$, a genetic Gerstmann–Sträussler–Scheinker disease–related mutation[7,12], and C2C12 myoblasts knockout (KO) for PrP$^C$ were cultured until their confluence reached 90% and then incubated with differentiation medium for 3 and 5 days, respectively (Fig. 3a–f), using C2C12 myoblasts as a control (Fig. 3a–f). Here, we used mouse PrP$^C$ instead of the human counterpart because these two proteins have more than 90% sequence homology[60]. We used anti-MyHC antibody and anti-MyoG antibody to detect the expression of myogenic differentiation markers, myosin heavy chain (MyHC) and myogenin (MyoG), respectively. To make direct comparisons, different samples were run on the same blot, and the cell lysates were probed with the anti-PrP monoclonal antibody 8H4, anti-MyHC antibody, anti-MyoG antibody, and anti-β-actin antibody (Fig. 3a). PrP$^C$ (WT PrP$^C$ and familial mutation F198S PrP$^C$) expression was significantly upregulated during myoblast differentiation (Fig. 3b). The above cells were immunostained with anti-PrP antibody (red) (Fig. 3e) or the anti-MyHC antibody MF-20 (red) (Fig. 3f), stained with DAPI (blue), and observed by confocal microscopy. Endogenous PrP$^C$ located on the plasma membrane, including its diglycosylated, monoglycosylated, and unglycosylated forms (Fig. 3a, e), and myoblast differentiation (Fig. 3a, f) were both observed when C2C12 cells (control) were incubated with differentiation medium for 3 days. After 5 days of differentiation, however, endogenous PrP$^C$ located on the plasma membrane, endogenous PrP$^C$ located in the cytoplasm (Fig. 3e), and abundant multinucleated, elongated myotubes (Fig. 3f) differentiated from C2C12 myoblasts were observed. Importantly, excess WT PrP$^C$ and F198S PrP$^C$ (Fig. 3a, e, f) located in the cytoplasm (Fig. 3e) both significantly inhibited and blocked muscle cell differentiation when C2C12 myoblasts stably expressing PrP$^C$ were incubated with differentiation medium for 3 and 5 days (Fig. 3a, c, d, f). After 3 days of differentiation, excess WT PrP$^C$ was partly located on the plasma membrane and partly located in the cytoplasm. After 5 days of differentiation, however, excess WT PrP$^C$ was mainly located in the cytoplasm (Fig. 3e). Moreover, PrP$^C$ deficiency completely blocked muscle cell differentiation when C2C12 myoblasts KO for PrP$^C$ was incubated with this medium for 3 and 5 days (Fig. 3a, c–f). These data demonstrate that in C2C12 myoblasts, PrP$^C$ knockout prevents myoblast differentiation. Overexpression of wild-type or mutant PrP$^C$ also prevents myoblast differentiation and most of the PrP$^C$ is in the cytoplasm. Together, the data showed that maintenance of PrP$^C$ homeostasis is important for myoblast differentiation.

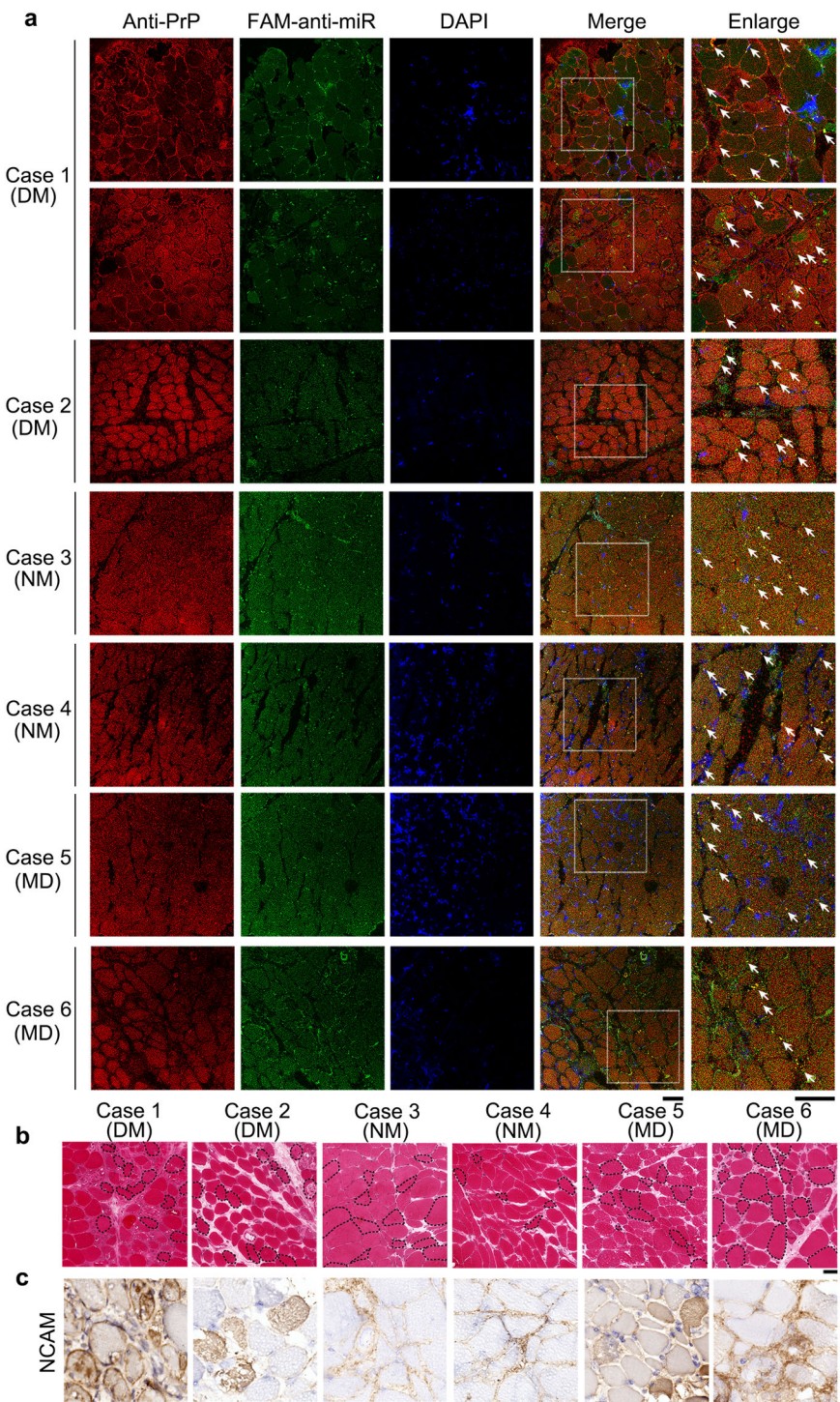

**Fig. 1 | Accumulation of PrP^C and the colocalization of PrP^C and miR-214-3p were observed in the skeletal muscle of six myopathy patients. a** Confocal images of frozen skeletal muscle sections from six myopathy patients. These myopathy patients included two patients with dermatomyositis (DM) (Cases 1 and 2), two with neurogenic myopathy (NM) (Cases 3 and 4), and two with muscular dystrophy (MD) (Cases 5 and 6). Shown are nuclei stained with DAPI (blue); signals were detected with the anti-PrP antibody 8H4 (red), and miR-214-3p was detected by FISH (green) using an FAM-labeled miR-214-3p probe (FAM-anti-miR).

The enlarged regions (right) show 4-fold enlarged images from the merged images. White arrows indicate colocalization of PrP^C and miR-214-3p in granules. Scale bars, 75 μm. **b** H&E staining of the frozen sections showed that the skeletal muscle of the six myopathy patients was characterized by incomplete muscle regeneration. Dashed black loops indicate centrally nucleated fibers. Scale bar, 400 μm.
**c** Immunohistochemical analysis of NCAM expression in the frozen skeletal muscle sections showed a positive signal (brown) in the muscle bundles. Scale bar, 100 μm.

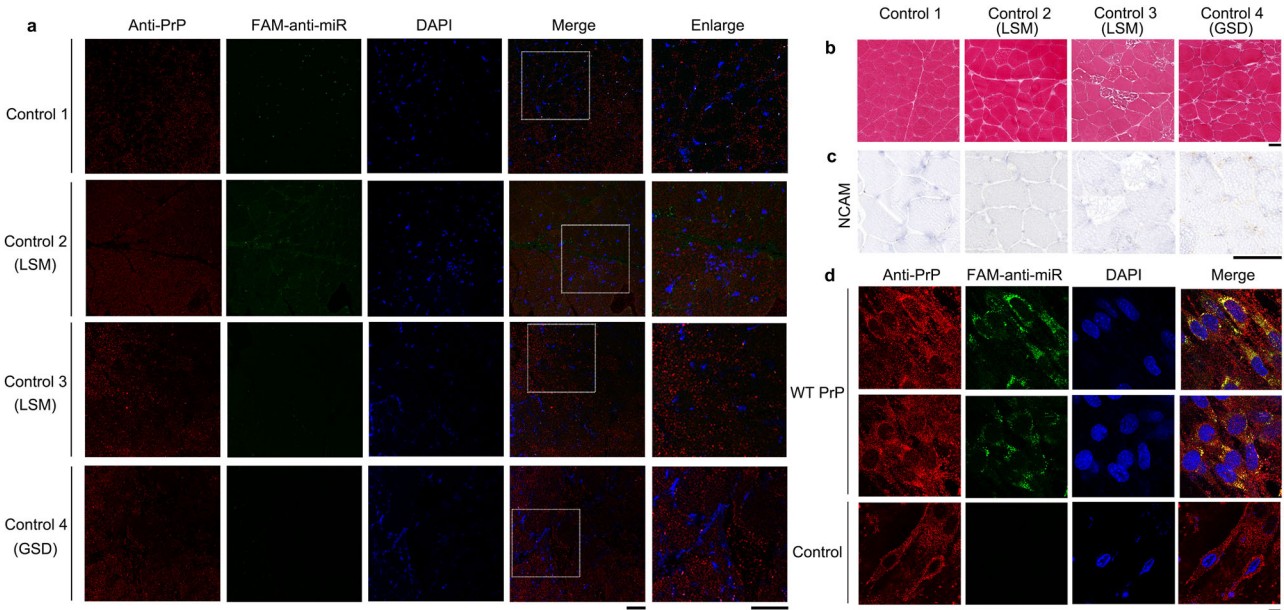

**Fig. 2 | Accumulation of PrP^C and the colocalization of PrP^C and miR-214-3p were not observed in the skeletal muscle of one healthy control (Control 1), two controls with lipid storage myopathy (LSM) (Controls 2 and 3), and one control with glycogen storage disease (GSD) (Control 4). a** Confocal images of frozen skeletal muscle sections from these four controls. We have replaced panel a with a correct version of these four controls, in which the control samples do have normal levels of PrP^C (red). **b, c** H&E staining and immunohistochemical staining of the frozen sections showed that muscle regeneration was not observed in the skeletal muscle of these four controls. The experimental conditions are the same as those in Fig. 1a–c. **d** Colocalization of PrP^C and miR-214-3p in C2C12 mouse myoblasts (control) and C2C12 myoblasts stably expressing WT PrP^C upon differentiation for 4 days. Shown are nuclei stained with DAPI (blue); signals were detected with the anti-PrP antibody 8H4 (red), and miR-214-3p was detected by FISH (green). Scale bar, 10 μm.

## Excess PrP^C strongly inhibits skeletal muscle cell autophagy and blocks myoblast differentiation

Given that the benign cellular form PrP^C does precisely regulate the differentiation of skeletal muscle cells (Fig. 3), we predicted that PrP^C might directly regulate skeletal muscle cell autophagy, an important pathway for cell differentiation. We next used confocal microscopy and western blotting to test this hypothesis. Endogenous PrP^C (red) was mainly attached to the plasma membrane and did not colocalize with the autophagy marker LC3B (green) in C2C12 cells (control) upon differentiation for 3 days (Fig. 4a). However, the colocalization of endogenous PrP^C and LC3B puncta (green dots, Fig. 4b) was observed in the elongated myotubes differentiated from C2C12 myoblasts after 5 days of differentiation (yellow dots in the merged image, Fig. 4b). Importantly, excess PrP^C strongly inhibited skeletal muscle cell autophagy and blocked myoblast differentiation, producing much fewer LC3B-positive puncta (green dots, Fig. 4a, b) when C2C12 cells stably expressing WT PrP^C were incubated with the differentiation medium for 3 and 5 days. After 3 days of differentiation, excess PrP^C was partly located on the plasma membrane and partly located in the cytoplasm (Fig. 4a); after 5 days of differentiation, however, excess PrP^C was primarily located in the cytoplasm (Fig. 4b); under both conditions, the colocalization of PrP^C and LC3B (the merged images, Fig. 4a, b) was not observed. Moreover, PrP^C deficiency resulted in increased autophagic activity in skeletal muscle cells (green dots, Fig. 4a, b) and blocked muscle cell differentiation when C2C12 myoblasts KO for PrP^C were incubated with this medium for 3 and 5 days. To gain a quantitative understanding of how PrP^C regulates skeletal muscle cell autophagy, we detected two autophagy-related proteins, ATG5 and LC3B, in the aforementioned cells using anti-ATG5 and anti-LC3B antibodies, respectively (Fig. 4c). Upon differentiation for 3 and 5 days, the relative amounts of ATG5 and LC3B-II in the cell lysates from C2C12 myoblasts stably expressing WT PrP^C were significantly lower than those in the control cell lysates from C2C12 myoblasts (orange) ($p < 0.05$) (Fig. 4d, e). Thus, excess PrP^C significantly inhibited

ATG5-dependent and LC3B-dependent autophagy in differentiating C2C12 cells stably expressing WT PrP^C compared to those in differentiating C2C12 myoblasts (control). In C2C12 cells, overexpression of WT PrP^C strongly inhibited autophagy-related proteins (ATG5 and LC3B). Moreover, PrP^C deficiency increased skeletal muscle cell autophagy when C2C12 myoblast KO for PrP^C was incubated with this medium for 3 and 5 days (Fig. 4d, e). Intriguingly, we observed that the levels of ATG5 and LC3B-II were strongly enhanced during myoblast differentiation (Fig. 4d, e). These results demonstrate that excess PrP^C strongly inhibits skeletal muscle cell autophagy and blocks myoblast differentiation and indicate that PrP^C deficiency increases skeletal muscle cell autophagy and blocks muscle differentiation.

We performed additional experiments to investigate whether autophagy is responsible for the effect of overexpression of PrP^C on myoblast differentiation (Supplementary Fig. 1). Rapamycin, a widely used autophagy enhancer, does not seem to be an appropriate tool here because rapamycin is well documented to inhibit C2C12 cell differentiation through various mammalian (or mechanistic) target of rapamycin (mTOR) mechanisms that are independent of autophagy[61,62]. Instead, trehalose, a mTOR-independent autophagy enhancer[63,64], was used to induce ATG5-dependent and LC3B-dependent autophagy in differentiating C2C12 cells stably expressing WT PrP^C (Supplementary Fig. 1a, d, e). Overexpression of PrP^C significantly inhibited skeletal muscle cell autophagy and blocked myoblast differentiation (Supplementary Fig. 1a–e). Compared with C2C12 myoblasts (control), incubation of C2C12 myoblasts stably expressing WT PrP^C with 50 mM trehalose restored their autophagic activity and thus partially restored muscle cell differentiation (Supplementary Fig. 1a–e). Therefore, the inhibitory effect of excess PrP^C on autophagy in skeletal muscle partially results in an inhibitory effect of excess PrP^C on myoblast differentiation. Compared with C2C12 myoblasts (control), incubation of C2C12 myoblasts stably expressing WT PrP^C with 100 mM trehalose significantly increased their autophagic activity and thus blocked muscle cell differentiation (Supplementary Fig. 1a–e).

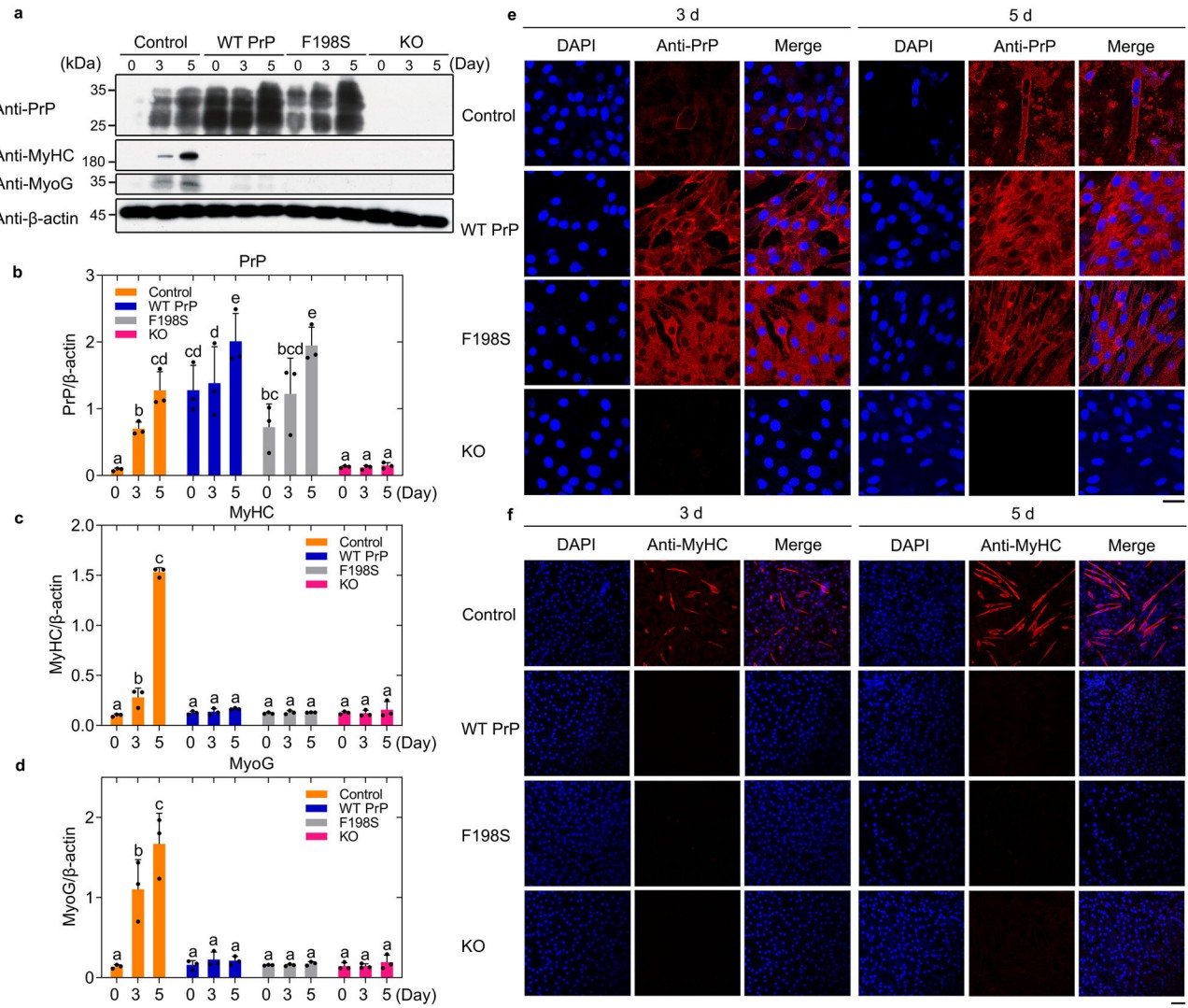

**Fig. 3 | Maintenance of PrP^C homeostasis is important for myoblast differentiation. a** Western blot for PrP^C and the myogenic differentiation markers MyHC and MyoG during C2C12 mouse myoblasts (control), C2C12 myoblasts stably expressing full-length wild-type mouse PrP^C (WT PrP^C), C2C12 myoblasts stably expressing F198S PrP^C, and C2C12 myoblasts KO for PrP^C cultured until their confluence reached 90% and then incubated with differentiation medium for 0, 3, and 5 days, respectively. β-actin served as the protein loading control. The relative amount of PrP^C (**b**), MyHC (**c**), or MyoG (**d**) in the above cell lines (control, orange; WT PrP^C, blue; F198S PrP^C, gray; and KO for PrP^C, pink) (solid black circles shown in scatter plots) was expressed as the mean ± SD (with error bars) of values obtained in *n* = 3 independent experiments. One-way two-sided ANOVA and multiple comparisons with no adjustments were performed by SPSS 19.0, and different letters indicate significant differences at the level of *p* < 0.05. Immunofluorescence imaging of the above four cell lines incubated with differentiation medium for 3 and 5 days, respectively, using antibody against PrP^C (red) (**e**) or MyHC (red) (**f**) and staining with DAPI (blue). Scale bars, 50 (**e**) and 75 μm (**f**), respectively. We have replaced the second row for Day 3 in (**f**) with a correct version of WT PrP^C, in which WT PrP^C-expressing cells did not have any MyHC signal at day 3 in most fields of view. Source data are provided as a Source Data file.

Together, the data showed that autophagy is partially responsible for the effect of overexpression of PrP^C on myoblast differentiation and that maintenance of autophagy homeostasis is also important for myoblast differentiation.

## PrP^C selectively binds to a subset of miRNAs during myoblast differentiation

To explore the mechanism underlying a potential relationship between PrP^C and miRNAs during myoblast differentiation, we pursued PrP^C-associated miRNAs in differentiating C2C12 cells. To this end, we performed formaldehyde crosslinking and RNA immunoprecipitation (RIP) using anti-mouse antibody-binding beads incubated with anti-PrP antibody or mouse IgG to pulldown a subset of PrP^C-bound miRNAs from C2C12 cells stably expressing WT PrP^C and C2C12 myoblasts (control) differentiated for 4 days and identified many

selectively enriched miRNAs based on sorted ratios of RIP to input. A subset of PrP^C-bound miRNAs (red) was significantly enriched in WT PrP^C RIP upon differentiation for 4 days compared to control RIP and two inputs (Fig. 5a). It should be mentioned that the overlap of miRNAs pulled down between the two sets of RIP is small (one third or one fifth of miRNAs identified for control RIP or WT PrP RIP). Since over-expression of PrP^C inhibited myoblast differentiation, it is possible that the difference in miRNAs pulled down might be mainly due to the differentiation versus undifferentiation state. The selective pulldown of specific miRNAs is not correlated with their abundance, because in our heat map, some miRNAs with low abundance (light blue) in two inputs were also strongly enriched in WT PrP^C RIP (deep red) (Fig. 5a). The above data suggest that PrP^C pulldown of miRNAs is specific. Our western blotting experiments indicated that the antibody-binding beads IPed endogenous PrP^C in C2C12 myoblasts after 4 days of

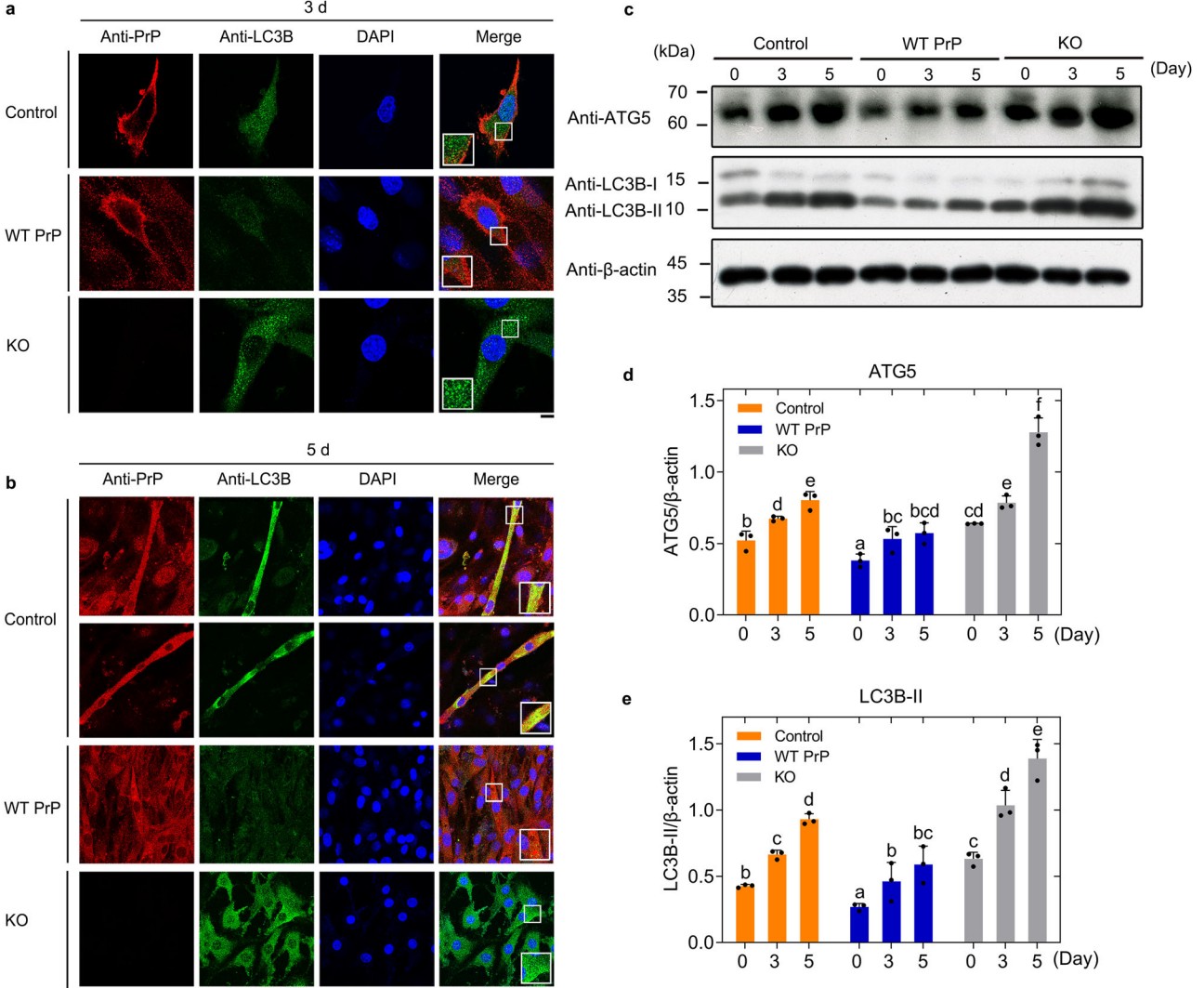

**Fig. 4 | Excess PrP$^C$ strongly inhibits skeletal muscle cell autophagy and blocks myoblast differentiation.** Immunofluorescence imaging of C2C12 mouse myoblasts (control), C2C12 myoblasts stably overexpressing WT PrP$^C$, and C2C12 myoblasts KO for PrP$^C$ cultured until their confluence reached 90% and then incubated with differentiation medium for 3 (**a**) and 5 (**b**) days, respectively, using antibodies against PrP$^C$ (red) and LC3B (green) and staining with DAPI (blue). The enlarged regions in the lower left corner (**a**) or the lower right corner (**b**) of the merged images show 4-fold enlarged images from the same images. Scale bars, 7.5 (**a**) and 50 (**b**) μm, respectively. We have replaced the first row in (**a**) with a correct version of control, in which LC3B did not overlap with PrP$^C$ in most fields of view.

**c** Western blot for PrP$^C$ and the autophagy markers ATG5 and LC3B during the above three cell lines incubated with differentiation medium for 3 and 5 days, respectively. β-actin served as the protein loading control. The relative amount of ATG5 (**d**) or LC3B-II (**e**) in the above cell lines (control, orange; WT PrP$^C$, blue; and KO for PrP$^C$, gray) (solid black circles shown in scatter plots) was expressed as the mean ± SD (with error bars) of values obtained in $n = 3$ independent experiments. One-way two-sided ANOVA and multiple comparisons with no adjustments were performed by SPSS 19.0, and different letters indicate significant differences at the level of $p < 0.05$. Source data are provided as a Source Data file.

differentiation (Fig. 5b). Displaying the small RNA-seq data in two volcano plots (Fig. 5c, d) and using a log$_2$ fold-change cutoff of 0.5 and a $p$-value cutoff of 0.05, we identified 51 miRNAs bound by PrP$^C$ in differentiating C2C12 cells stably expressing WT PrP$^C$ versus 31 miRNAs bound by endogenous PrP$^C$ in differentiating C2C12 cells (control) (Supplementary Data 1). From a total of 519 and 455 miRNAs that we identified, 51 and 31 were significantly enriched in differentiating C2C12 cells stably expressing WT PrP$^C$ (Fig. 5d) and differentiating C2C12 cells (control) (Fig. 5c), respectively, and 468 and 424 were depleted, respectively. Ten overlaps were identified between the two subsets of miRNAs (Fig. 5e), and these miRNAs were further verified by RT−qPCR (Fig. 5f) and Gene Ontology (GO) analyses (Fig. 5h, i). Forty-one miRNAs bound by PrP$^C$, such as miR-214-3p, miR-204-5p, miR-499-5p, and miR-92b-3p, which have inhibitory effects on cell differentiation and autophagy[65–68], were identified for WT PrP RIP only,

and 21 miRNAs bound by endogenous PrP$^C$, such as miR-486a-5p, miR-181b-5p, and miR-206-3p, which have enhancing effects on cell differentiation and autophagy[69–71], were identified for control RIP only (Fig. 5c, d and Supplementary Data 1). A subset of PrP$^C$-bound miRNAs, including miR-214-3p, miR-204-5p, and miR-499-5p, and a subset of endogenous PrP$^C$-bound miRNAs, such as miR-486a-5p, miR-181b-5p, and miR-206-3p, but not U6 or any of the non-target miRNAs tested, were significantly enriched in differentiating C2C12 cells stably expressing WT PrP$^C$ (Fig. 5g) and the differentiating control cells (Fig. 5f). We then identified the top six biological processes (BP) of GO enrichment in the control cells, including "Regulation of cellular process", "Regulation of cell differentiation", and "Cell differentiation" (Fig. 5h), and the top eight BPs of GO enrichment in C2C12 cells stably overexpressing WT PrP$^C$, including "Regulation of cellular process", "Cell differentiation", "Autophagy", and "Process utilizing autophagic

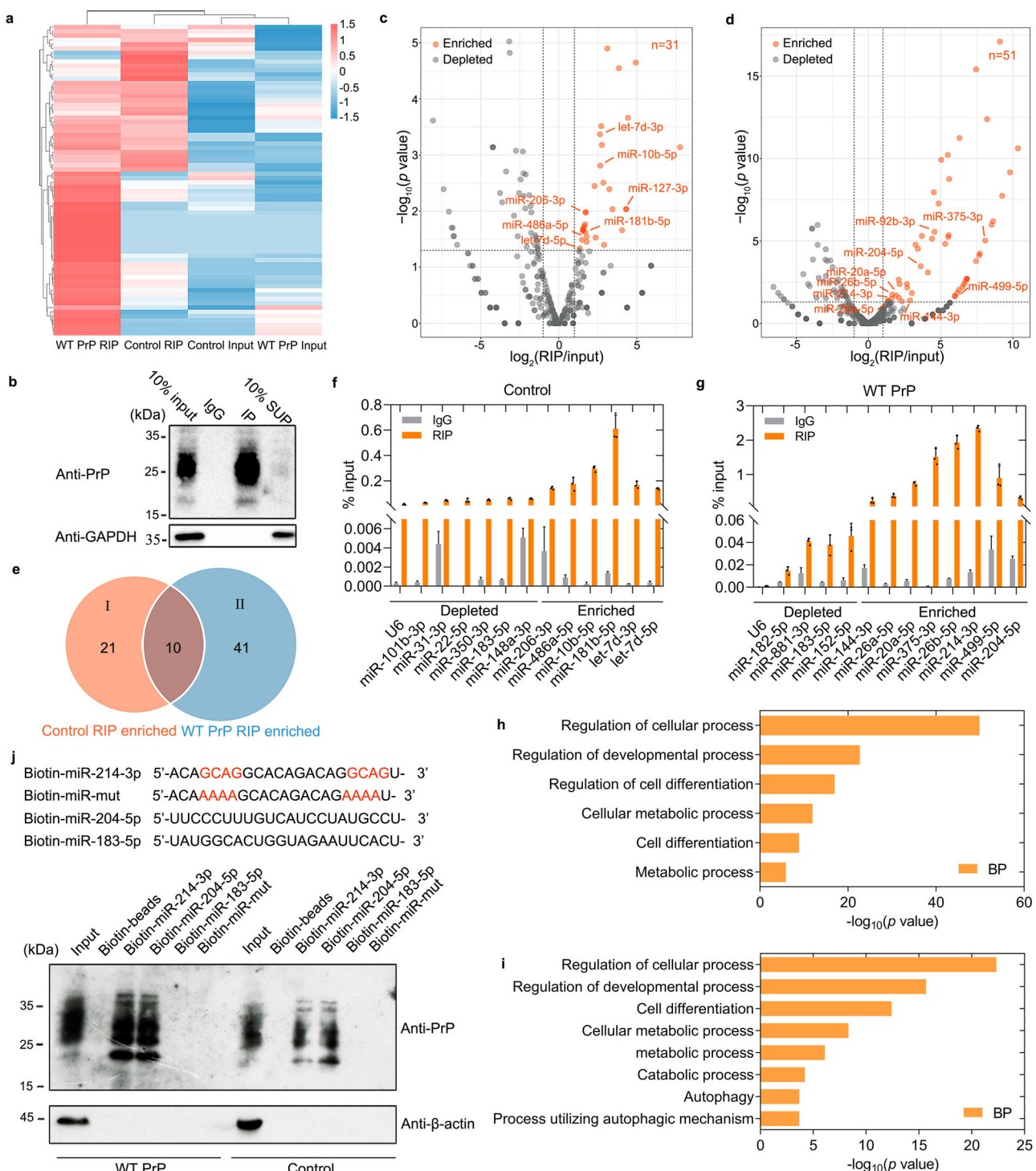

**Fig. 5 | PrP^C selectively binds to a subset of miRNAs during myoblast differentiation. a** Profile of total miRNAs and PrP^C IP-enriched miRNAs at day 4 of differentiation in C2C12 myoblasts stably overexpressing WT PrP^C (WT PrP^C) and C2C12 myoblasts (control). Heat map shows the expression levels of miRNAs (high abundance, red; and low abundance, blue) and fold-enrichment compared to Inputs in the above two cell lines. **b** Western blotting analysis of PrP^C immuno-precipitation in C2C12 myoblasts at day 4 of differentiation by probing PrP^C and GAPDH. Volcano plot of 31 (**c**) and 51 (**d**) PrP^C-bound miRNAs identified by small RNA-seq in differentiating C2C12 cells (control) and differentiating C2C12 cells stably expressing WT PrP^C, respectively. Representative PrP^C RIP-enriched miRNAs and depleted miRNAs are highlighted in orange and gray, respectively. Two-sided Student's *t*-test was used to compare two sets of the small RNA-seq data. **e** Venn diagram of miRNAs enriched by PrP^C RIP in C2C12 cells stably expressing WT PrP^C versus miRNAs enriched by PrP^C RIP in C2C12 myoblasts (control). Group I (*n* = 21,

Control RIP only, orange), *n* = 10 (overlap, brown), and group II (*n* = 41, WT PrP RIP only, blue). RT−qPCR validation of depleted versus enriched miRNAs in differentiating C2C12 myoblasts (control) (**f**) and those stably expressing WT PrP^C (**g**) normalized to total input. An axis break was introduced in the ordinate so that each orange bar is clearly paired with a gray bar as control (IgG). U6 snRNA served as a negative control. Data are presented as means ± SD (*n* = 3 biologically independent measurements). The top six biological processes (BP, orange) of Gene Ontology (GO) enrichment in the control cells (**h**) and the top eight BPs of GO enrichment in C2C12 cells stably overexpressing WT PrP^C (**i**). The differential genes annotated by GO terms were used to calculate the gene list and number of genes for each term. **j** Pulldown of endogenous PrP^C and excess PrP^C with WT and mutant biotin-labeled miR-214-3p or biotin-labeled miR-204-5p in the above two cell lines at day 4 of differentiation.

mechanism" (Fig. 5i). These significantly enriched pathways are related to cell differentiation and autophagy and may reflect the regulatory roles of miRNAs enriched by PrP$^C$.

The secondary structure of miR-214-3p, an example of 41 miRNAs bound by PrP$^C$, was predicted by RNAComposer[72] (Supplementary Fig. 2a), and the structure of mouse PrP (PDB 1XYX)[73] shows three α-helices (α1, α2, and α3) in its C-terminal domain (Supplementary Fig. 2b). We next used HDOCK, a protein–protein/nucleic acid protein docking web server that combines template-based and free docking[74], to predict the binding sites of miR-214-3p for PrP$^C$, and the molecular docking of the protein with miR-214-3p was performed using HDOCK (Supplementary Fig. 2b). The interface between PrP$^C$ and miR-214-3p features two π-bonds between Tyr225 in PrP$^C$ and GMP8 in miR-214-3p and a salt bridge between Arg228 in PrP$^C$ and CMP9 in miR-214-3p (Supplementary Fig. 2c, d). Asp166, Gln167, Tyr168, Ser169, Asn170, Gln171, Val214, Thr 215, Tyr217, Gln218, Gln222, Tyr224, and Tyr225 in PrP$^C$ form abundant hydrogen bonds with two GCAG sequences present in miR-214-3p (Supplementary Fig. 2d), which contribute to maintenance of the structure of the PrP$^C$-miR-214-3p complex. Therefore, two GCAG sequences (orange) present in miR-214-3p are the predicted binding sites for PrP$^C$ and were mutated into two AAAA sequences in the miR-214-3p mutant. We then pulled down endogenous PrP$^C$ and excess PrP$^C$ with biotin-labeled WT miR-214-3p or biotin-labeled miR-204-5p from C2C12 cell extracts using nonspecific miR-183-5p and miR-214-3p mutant as negative controls (Fig. 5j). Of note, miR-214-3p and miR-204-5p were able to capture both endogenous PrP$^C$ and excess PrP$^C$, and mutating the GCAG sequences in miR-214-3p completely abolished its ability to capture PrP$^C$. Together, these results demonstrate that PrP$^C$ selectively binds to a subset of miRNAs, especially miR-214-3p and miR-204-5p in differentiating C2C12 myoblasts.

### PrP$^C$ increases the stability of mature miR-214-3p to significantly inhibit autophagy and myoblast differentiation

Given that PrP$^C$ selectively binds to miR-214-3p and miR-204-5p during myoblast differentiation (Fig. 5), we predicted that these two miRNAs might regulate autophagy and differentiation of skeletal muscle cells via interaction with PrP$^C$. We next used miRNA reporter gene assays and western blotting to test this hypothesis. Using TargetScan and miRanda algorithms[75], we focused on two candidate target genes, *ATG5* for miR-214-3p and *LC3B* for miR-204-5p, and tested their functional responses. We used miR-214-3p and miR-204-5p reporter gene assays, in which miR-214-3p represses the wild-type 3' end of the untranslated region (WT 3'UTR) of the gene *ATG5* associated with luciferase (Fig. 6a) and miR-204-5p represses the WT 3'UTR of the gene *LC3B* associated with luciferase (Fig. 6b). The mutant versions of the 3'UTR (Mut 3'UTR), in which the miR-214-3p binding site (Fig. 6a) and the miR-204-5p binding site (Fig. 6b) were mutated, served as negative controls in miRNA reporter gene assays. Compared with transfection of control (NC) in C2C12 myoblasts KO for PrP$^C$, transfection of miR-214-3p mimic (Fig. 6a) or miR-204-5p mimic (Fig. 6b) at 10 μM significantly inhibited the relative luciferase activity of the reporter ($p = 0.001$ or 0.035) but did not significantly change that of the mutant version of the reporter ($p = 0.27$ or 0.93) in this cell line. Importantly, compared with transfection of NC in C2C12 myoblasts stably expressing WT PrP$^C$, transfection of miR-214-3p mimic (Fig. 6a) or miR-204-5p mimic (Fig. 6b) at 10 μM more significantly inhibited the relative luciferase activity of the reporter ($p = 0.00027$ or 0.001) but did not significantly change that of the mutated reporter ($p = 0.47$ or 0.37) in the C2C12 myoblasts. Compared with that in C2C12 myoblasts KO for PrP$^C$, excess PrP$^C$ significantly enhanced not only the binding affinity of miR-214-3p toward its downstream target, the autophagy marker ATG5 (Fig. 6a) ($p = 0.00021$), but also the binding affinity of miR-204-5p toward its downstream target, the autophagy marker LC3B (Fig. 6b) ($p = 0.0047$), in C2C12 myoblasts stably expressing WT PrP$^C$ when incubated with the differentiation medium for 4 days. Moreover, we

wanted to determine whether PrP$^C$ could modulate the stability of mature miR-214-3p and miR-204-5p in C2C12 myoblasts (Fig. 6c–e). C2C12 myoblasts stably expressing WT PrP$^C$ or PrP$^C$ KO were incubated with differentiation medium for 4 days and then treated with 20 mg/ml α-amanitin, an inhibitor of RNA polymerase II. Compared with that in C2C12 myoblasts KO for PrP$^C$, excess PrP$^C$ significantly enhanced the relative stability of miR-214-3p and miR-204-5p (Fig. 6c, d), two examples of 41 miRNAs bound by PrP$^C$, in C2C12 myoblasts stably expressing WT PrP$^C$ during myoblast differentiation but did not significantly change the relative stability of nonspecific miR-183-5p (Fig. 6e), a negative control, in this cell line. To gain a quantitative understanding of how miR-214-3p and miR-204-5p regulate skeletal muscle cell autophagy, we measured ATG5 and LC3B levels as proxies for autophagy and performed western blot analysis for ATG5, LC3B, and the myogenic differentiation marker MyHC after transfection with NC, miR-214-3p inhibitor, or miR-204-5p inhibitor at 10 μM in the above cell lines (Fig. 6f, g). Upon differentiation for 4 days, the relative amounts of ATG5 (Fig. 6h) and MyHC (Fig. 6i) in the cell lysates from C2C12 myoblasts stably expressing WT PrP$^C$ or KO for PrP$^C$ transfected with 10 μM miR-214-3p inhibitor were significantly higher than those in the cell lysates from the aforementioned cell lines transfected with NC. Thus, compared with transfection of NC in C2C12 myoblasts KO for PrP$^C$ and C2C12 myoblasts stably expressing WT PrP$^C$, transfection of 10 μM miR-214-3p inhibitor significantly promoted ATG5-related autophagy and myoblast differentiation in the aforementioned cell lines (Fig. 6f, h, i), suggesting that miR-214-3p significantly inhibits autophagy and differentiation of skeletal muscle cells via specific interaction with PrP$^C$. Upon differentiation for 4 days, transfection of 10 μM miR-204-5p inhibitor significantly promoted LC3B-related autophagy in C2C12 myoblasts KO for PrP$^C$ and C2C12 myoblasts stably expressing WT PrP$^C$ compared to transfection of NC in the aforementioned cell lines (Fig. 6j). Importantly, PrP$^C$ deficiency completely blocked muscle cell differentiation after transfection with NC or miR-204-5p inhibitor at 10 μM in C2C12 myoblasts KO for PrP$^C$ upon differentiation for 4 days (Fig. 6k). Compared with transfection of NC in C2C12 myoblasts stably expressing WT PrP$^C$, however, transfection of miR-204-5p inhibitor significantly promoted myoblast differentiation in the above cell line (Fig. 6k), suggesting that miR-204-5p, another PrP$^C$-bound miRNA, significantly inhibits myoblast differentiation. Therefore, inhibition of miR-214-3p significantly enhances both autophagy (ATG5) and myoblast differentiation in C2C12 cells (even when PrP$^C$ is knocked out), and such enhancement is reduced when PrP$^C$ is overexpressed. In PrP$^C$-KO C2C12 cells, inhibition of miR-204-5p significantly promotes autophagy (LC3B-II), but had no effect on myoblast differentiation (no differentiation); in C2C12 cells overexpressing PrP$^C$, inhibition of miR-204-5p enhances both autophagy (LC3B-II) and myoblast differentiation. Moreover, PrP$^C$ overexpression enhanced the stability of miR-204-5p and to a lesser degree for miR-214-3p, and the inhibitory effect of these miRNAs on their target proteins (ATG5 for miR-214-3p and LC3B for miR-204-5p) under PrP$^C$ overexpression conditions is significantly stronger than that of the KO group, suggesting that the miRNA actions are enhanced by their binding to PrP$^C$. However, because the effect of miRNA inhibitor is there with either KO or PrP$^C$ overexpression, the possibility that the miRNA actions are not dependent on their binding to PrP$^C$ cannot be excluded. These results demonstrate that PrP$^C$ increases the stability of mature miR-214-3p and miR-204-5p and enhances miRNA repression of their downstream mRNA targets to significantly inhibit autophagy and myoblast differentiation.

### A higher level of PrP$^C$ colocalized with miR-214-3p in the skeletal muscle of myopathy patients

PrP$^C$ dysfunction and miRNA overexpression are both observed in the skeletal muscle of myopathy patients[19,22,37]. We found that miR-214-3p, an example of 41 miRNAs bound by PrP$^C$, significantly inhibited

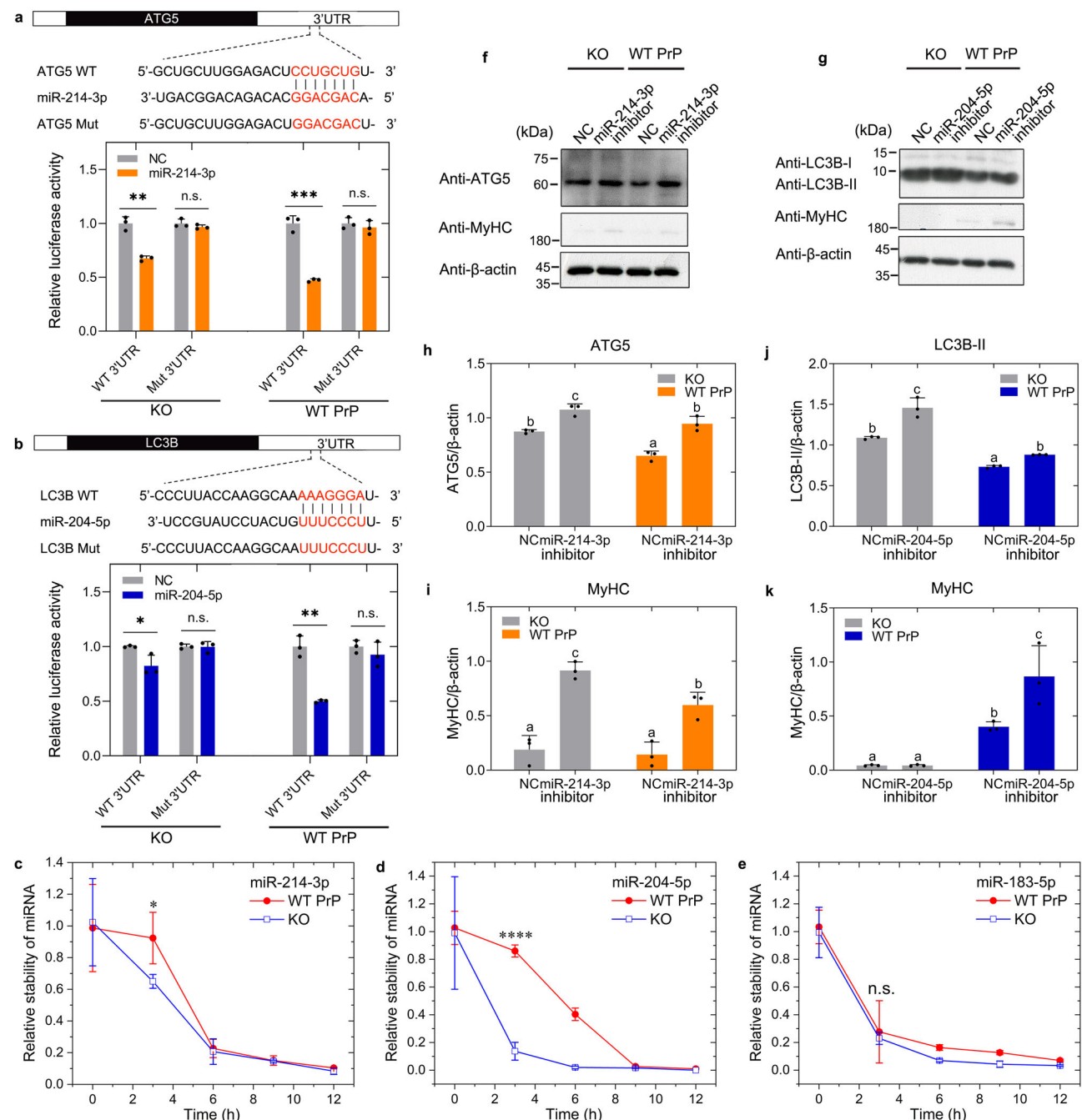

**Fig. 6 | PrP^C increases the stability of mature miR-214-3p to inhibit autophagy and myoblast differentiation.** Dual luciferase reporter assays, in which miR-214-3p (**a**) or miR-204-5p (**b**) represses the wild-type 3′ end of untranslated region (WT 3′UTR) of the gene *ATG5* or *LC3B* associated with luciferase. The relative luciferase activity of the reporters after transfection with NC (gray), miR-214-3p mimic (**a**, orange), or miR-204-5p mimic (**b**, blue) at day 4 of differentiation in C2C12 myoblasts stably overexpressing WT PrP^C or KO for PrP^C (solid black circles shown in scatter plots), and the relative stability of miR-214-3p (**c**), miR-204-5p (**d**), and nonspecific miR-183-5p (**e**) in C2C12 myoblasts stably overexpressing WT PrP^C (red) or KO for PrP^C (blue) incubated with differentiation medium for 4 days and then treated with 20 mg/ml α-amanitin, were expressed as the mean ± SD (with error bars) of values obtained in three independent experiments. $p = 0.001$, 0.268, 0.000226, 0.467 (**a**), 0.035, 0.926, 0.001, 0.372 (**b**), 0.048 (**c**), 0.000086 (**d**), and 0.742 (**e**), respectively. Western blot for ATG5 (**f**), LC3B (**g**), and MyHC after

transfection with NC, miR-214-3p inhibitor (**f**), or miR-204-5p inhibitor (**g**) at 10 μM in the aforementioned cells upon differentiation for 4 days. β-actin served as the protein loading control. The relative amount of ATG5 (**h**), LC3B-II (**j**), or MyHC (**i**, **k**) in the aforementioned cells (solid black circles shown in scatter plots) was expressed as the mean ± SD (with error bars) of values obtained in three independent experiments. Transfection of NC, miR-214-3p inhibitor (**h**, **i**), or miR-204-5p inhibitor (**j**, **k**) in C2C12 myoblasts stably overexpressing WT PrP^C (orange or blue) or KO for PrP^C (gray). **a**–**e** Statistical analyses were performed using two-sided Student's *t*-test. Values of $p < 0.05$ indicate statistically significant differences. The following notation is used throughout: *$p < 0.05$; **$p < 0.01$; ***$p < 0.001$; and ****$p < 0.0001$ relative to controls. n.s., no significance. **h**–**k** One-way two-sided ANOVA and multiple comparisons with no adjustments were performed by SPSS 19.0, and different letters indicate significant differences at the level of $p < 0.05$.

autophagy and differentiation of skeletal muscle cells (Figs. 5 and 6a, b, f–k). Based on these observations, we hypothesized that PrP$^C$ colocalizes with miR-214-3p in the skeletal muscle of myopathy patients. To test this hypothesis, we took confocal images of frozen skeletal muscle sections from the six myopathy patients and these four controls (Figs. 1a and 2a), and obtained confocal images of C2C12 mouse myoblasts (control) and C2C12 myoblasts stably overexpressing WT PrP$^C$ upon differentiation for 4 days (Fig. 2d). The frozen skeletal muscle sections and the above two cell lines, in which miR-214-3p was detected by FISH (green) using an FAM-labeled miR-214-3p probe, were immunostained with the anti-PrP antibody 8H4 (red), stained with DAPI (blue), and visualized by confocal microscopy. Excess PrP$^C$ (red) was mainly located in the cytoplasm, and abundant miR-214-3p (green dots) and the colocalization of excess PrP$^C$ and miR-214-3p (yellow dots in the merged images) were observed not only in the skeletal muscle of six myopathy patients with excessive PrP$^C$ (Fig. 1a) but also in differentiating C2C12 cells stably expressing WT PrP$^C$ (Fig. 2d). In sharp contrast, accumulation of PrP$^C$, miR-214-3p expression, and the colocalization of PrP$^C$ and miR-214-3p were not observed in the skeletal muscle of four controls with normal levels of PrP$^C$ (Fig. 2a). Similarly, both miR-214-3p expression and the colocalization of endogenous PrP$^C$ and miR-214-3p (the merged images) were not observed in differentiating C2C12 cells (control) (Fig. 2d). Thus, a higher level of PrP$^C$ colocalized with miR-214-3p in the skeletal muscle of the six myopathy patients (Fig. 1a). Similarly, overexpressed PrP$^C$ colocalized with miR-214-3p in differentiating C2C12 cells stably expressing WT PrP$^C$ (Fig. 2d). In sharp contrast, such a phenomenon was not observed not only in these four controls (Fig. 2a) but also in control C2C12 cells (Fig. 2d). Moreover, we determined that PrP$^C$ could modulate the stability of mature miR-214-3p in C2C12 myoblasts (Fig. 6c). Together, the data showed that under pathological conditions, overexpressed PrP$^C$ colocalized with miR-214-3p in skeletal muscle cells to increase the stability of mature miR-214-3p.

## PrP$^C$ selectively recruits a subset of miRNAs into its phase-separated condensate, which in turn enhances the LLPS of PrP$^C$

Given that PrP$^C$ specifically interacts with a subset of miRNAs, such as miR-214-3p and miR-204-5p, during myoblast differentiation (Fig. 5), and the miRNA actions are enhanced by their binding to PrP$^C$ (Fig. 6), we predicted that miR-214-3p and miR-204-5p might regulate the LLPS of PrP$^C$ via interaction with the protein. We next used confocal microscopy and fluorescence recovery after photobleaching (FRAP)[48,49,51,52,54–56] to test this hypothesis. Bacterial-purified WT mouse PrP$^C$, labeled by 5(6)-carboxy-tetramethylrhodamine N-succinimidyl ester (TAMRA, red fluorescence) and incubated with 1 × PBS (pH 7.4) on ice, underwent LLPS in vitro and formed protein condensates (Supplementary Fig. 3). PrP$^C$ formed abundant liquid droplets, and protein condensates formed by PrP$^C$ became much larger in the presence of low concentrations of miR-214-3p or miR-204-5p than in the absence of miRNA (Fig. 7a). Low concentrations of miR-214-3p and miR-204-5p strongly promoted in vitro LLPS of PrP$^C$ (Fig. 7a). In sharp contrast, low concentrations of nonspecific miR-183-5p, a negative control, only mildly enhanced in vitro LLPS of PrP$^C$ (Fig. 7a). To further test this hypothesis, we next took fluorescence images of in vitro phase-separated PrP$^C$ droplets with three miRNAs (Fig. 7b). 50 μM WT PrP$^C$, labeled by TAMRA (red fluorescence) and incubated with 1 × PBS (pH 7.4) containing 10 μM FAM-labeled miRNA (green fluorescence) on ice, also underwent LLPS in vitro (Fig. 7b). PrP$^C$ demixed droplets (red; Merge: yellow) fused with droplets of miR-214-3p or those of miR-204-5p (green) were observed by confocal microscopy, with excitation at 561 nm and 488 nm, respectively (Fig. 7b). Importantly, PrP$^C$ selectively recruited and concentrated a subset of miRNAs into its phase-separated condensate (Fig. 7b). The recruitment ability of PrP$^C$ on miR-214-3p and miR-204-5p (green; Merge: yellow) was much stronger than that on two PrP$^C$-unbound miRNAs (negative controls), miR-183-5p and mutant miR-214-3p (light green; Merge: orange) (Fig. 7b). The addition of miR-214-3p or miR-204-5p at low concentrations strongly promoted but the addition of miR-183-5p or miR-214-3p mutant did not strongly enhance the phase separation (Fig. 7a) and droplet fusion ability (Fig. 7b) of PrP$^C$. Therefore, recombinant PrP$^C$ selectively recruits miR-214-3p and miR-204-5p into its phase-separated condensate, which stimulates the LLPS of PrP$^C$ and the fusion of PrP droplets in vitro. Together, the data showed that a subset of miRNAs recruited by PrP$^C$ forms a positive feed-forward loop with PrP$^C$ to enhance in vitro LLPS of PrP$^C$.

We then investigated and evaluated the dynamics of in vitro phase-separated droplets of PrP$^C$ with/without miRNA by FRAP (Fig. 7c, d). FRAP of phase-separated PrP$^C$ droplets without miRNA or with a negative control miR-183-5p revealed fluorescence recovery of $(82.5 ± 0.3)\%$ or $(65.1 ± 0.3)\%$ within 30 s (Fig. 7d). In sharp contrast, FRAP of phase-separated PrP$^C$ droplets coacervated with miR-214-3p or miR-204-5p revealed much lower fluorescence recovery, $(34.5 ± 0.8)\%$ or $(24.9 ± 0.3)\%$, within 30 s (Fig. 7d). According to Fig. 7c, d, miR-214-3p and miR-204-5p reduced fluorescence recovery. This means that miR-214-3p and miR-204-5p decrease the fluidity of LLPS condensates, possibly because these miRNAs could modulate liquid-to-solid transitions in phase-separated PrP$^C$ condensates. The aforementioned experiments help drive the narrative that miR-214-3p and miR-204-5p at low concentrations decrease fluorescence recovery and modulate the liquid nature of PrP$^C$ droplets in vitro.

Altogether, these data strongly suggest that the interactions between PrP$^C$ and miR-214-3p or miR-204-5p control liquidity and that miR-214-3p and miR-204-5p reduce PrP$^C$ mobility via specific interactions with PrP$^C$. Therefore, miR-214-3p and miR-204-5p are key factors in modulating PrP$^C$ liquid-phase condensation.

## PrP$^C$ recruits miR-214-3p into its phase-separated condensate in living skeletal muscle cells, which results in the inhibition of autophagy

Given that PrP$^C$ selectively recruits a subset of miRNAs into its phase-separated condensate, which in turn enhances in vitro LLPS of PrP$^C$ (Fig. 7), we predicted that miR-214-3p, an example of these PrP$^C$-associated miRNAs, might regulate in vivo LLPS of PrP$^C$ and thus suppress skeletal muscle cell autophagy. PrP$^C$ contains an IDR in its N-terminal domain[7,48–52,54–56,73]. We next employed an optogenetic tool that uses blue light (488-nm laser) to activate IDR-mediated LLPS of proteins in living cells[76–78] and western blotting to test this hypothesis. We used this "optoDroplet" system (Fig. 8a) to take time-lapse images of living C2C12 cells expressing the mCherry-Cry2-WT PrP$^C$ construct, which contains PrP$_{1-37}$ IDR (residues 1–37) linked to mCherry fluorescent protein and Cry2 and then linked to PrP$_{38-230}$ IDR (residues 38–230), and mCherry-Cry2 fusion alone was used as a control (Fig. 8b, c). We also took time-lapse images (0–100 s) of a C2C12 cell expressing the mCherry-Cry2-WT PrP$^C$ construct (Fig. 8d, e). PrP$^C$ underwent light-activated LLPS and formed abundant liquid droplets (white puncta) in the cytoplasm of living C2C12 cells (Fig. 8c, d). Two small liquid condensates gradually fused into one larger liquid droplet in living C2C12 cells (Fig. 8e and Supplementary Movie 1). To test the first half of this hypothesis, we took live-cell images of PrP$^C$-miRNA speck formation when 10 μM FAM-labeled miRNA (green) was transfected into C2C12 cells stably expressing mCherry (red) or WT PrP$^C$-mCherry (red) (Fig. 8f). PrP$^C$ colocalized with miR-214-3p in phase-separated condensates (Fig. 8f and Supplementary Movies 2–5). PrP$^C$ demixed droplets (red; Merge: yellow) fused with droplets of miR-214-3p (green), and the colocalization of PrP$^C$ and miR-214-3p (yellow puncta in the merged images) in phase-separated condensates was observed by confocal microscopy, with excitation at 561 nm and 488 nm, respectively (Fig. 8f). Together, the data showed that PrP$^C$ recruited and concentrated miR-214-3p into its phase-separated condensate in living skeletal muscle cells.

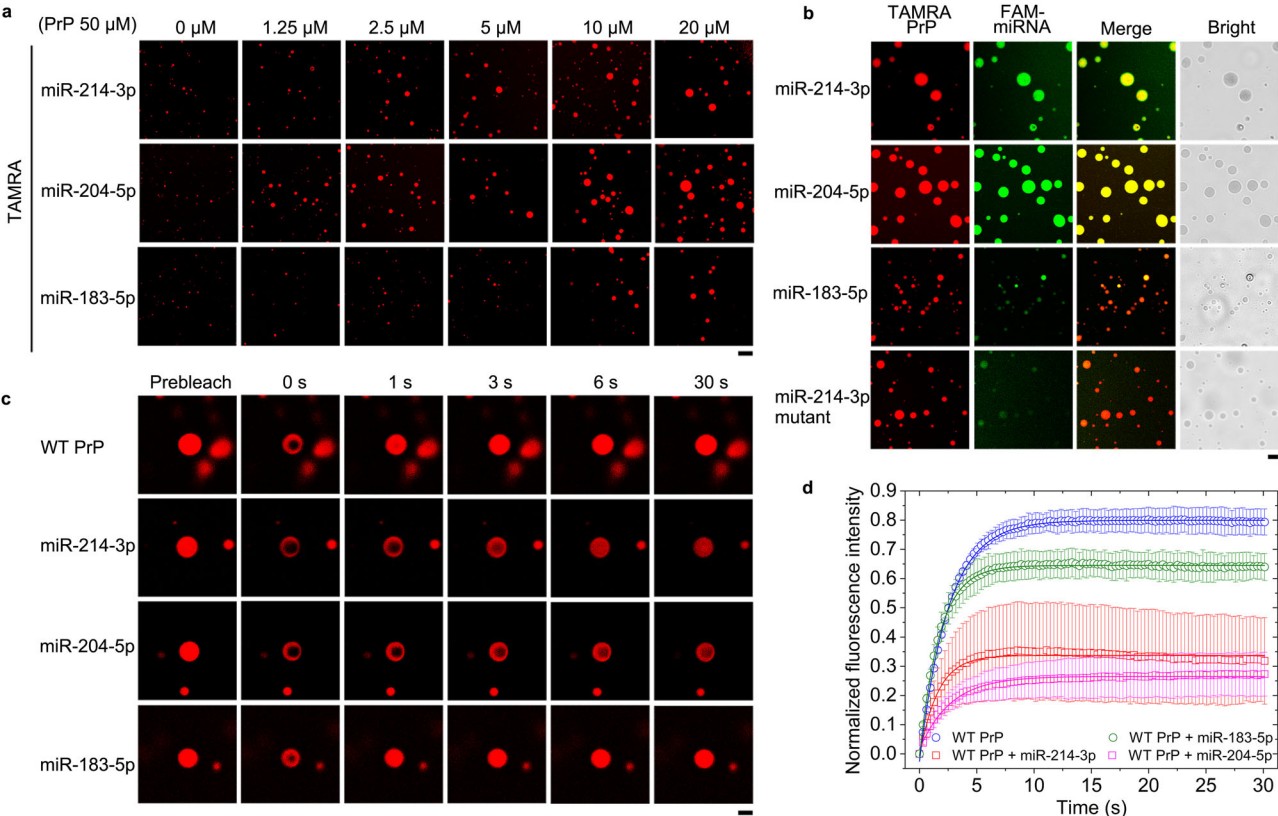

**Fig. 7 | PrP$^C$ selectively recruits a subset of miRNAs into its phase-separated condensate, which in turn enhances in vitro LLPS of PrP$^C$. a** Regulation of PrP$^C$ LLPS by three miRNAs. Fluorescence images of 50 μM recombinant wild-type mouse PrP$^C$ (WT PrP$^C$) labeled by TAMRA (red) and incubated with 1 × PBS (pH 7.4) containing 0, 1.25, 2.5, 5, 10, or 20 μM miRNA on ice for 5 min. Scale bar, 7.5 nm. **b** PrP$^C$ selectively recruits miR-214-3p and miR-204-5p into its phase-separated condensate. Fluorescence images of in vitro phase-separated droplets (red; Merge: yellow) of 50 μM TAMRA-labeled WT PrP$^C$ incubated with 1 × PBS containing 10 μM FAM-labeled miRNA (green) on ice for 5 min. Scale bar, 2.5 nm. **c** FRAP analysis on

the selected liquid droplets of 50 μM TAMRA-labeled WT PrP$^C$ before (prebleach), during (0 s), and after photobleaching (1, 3, 6, and 30 s, respectively). The internal photobleaching is marked by a black square. Scale bar, 2.5 nm. **d** Normalized kinetics of fluorescence recovery data of WT PrP$^C$ (blue circle), WT PrP$^C$ + miR-214-3p (red square), WT PrP$^C$ + miR-204-5p (magenta square), and WT PrP$^C$ + miR-183-5p (olive circle) obtained from FRAP intensity. The normalized fluorescence intensity is expressed as the mean ± SD (with error bars) of values obtained in $n = 3$ independent experiments. The solid lines show the best single exponential fit for the fluorescence intensity-time curves. Source data are provided as a Source Data file.

We then investigated and evaluated the dynamics of in vivo phase-separated droplets of PrP$^C$ with/without miRNA by FRAP (Fig. 8g–j). PrP$^C$ and miR-214-3p puncta exhibited features characteristic of liquid-like condensates (Fig. 8g, h and Supplementary Movies 6 and 7). FRAP of phase-separated PrP$^C$ droplets without miRNA revealed fluorescence recovery of (24.8 ± 1.8)% within 125 s (Fig. 8j). In sharp contrast, FRAP of phase-separated PrP$^C$ droplets coacervated with 10 μM miR-214-3p revealed much lower fluorescence recovery, (8.0 ± 1.9)%, within 125 s (Fig. 8j). The aforementioned experiments help drive the narrative that transfection of 10 μM miR-214-3p decreases fluorescence recovery and modulates the liquid nature of PrP$^C$ droplets in vivo. These data once again suggest that the interaction between PrP$^C$ and miR-214-3p controls liquidity and that miR-214-3p reduces PrP$^C$ mobility via specific interaction with PrP$^C$.

1,6-hexanediol is an aliphatic alcohol that disturbs weak hydrophobic interactions involved in phase separation[50,52,56]. To test whether hydrophobic interactions play a role in our observed puncta formation (Fig. 8c–f), we treated C2C12 cells expressing the mCherry-Cry2-WT PrP$^C$ construct with 1,6-hexanediol and took time-lapse images (0–300 s) of the living cells (Fig. 8k). We observed that the LLPS of PrP$^C$ (red puncta) in living C2C12 cells was dampened by treatment of 2.5% 1,6-hexanediol (Fig. 8k), suggesting that hydrophobic interactions play an important role in PrP$^C$ liquid-phase condensation. To test the second half of this hypothesis and the functional relevance of the observations that we made in vitro and in vivo with PrP$^C$ LLPS, we used C2C12

cells stably expressing mCherry-Cry2-WT PrP$^C$, which were cultured until their confluence reached 80% (Fig. 8l), transfected with or without 10 μM miR-214-3p for 30 min, and incubated with or without 2.5% 1,6-hexanediol for 5 min after activation by 488-nm laser for 10 min, using C2C12 myoblasts stably expressing mCherry-Cry2 as a control. To gain a quantitative understanding of how miR-214-3p regulates the LLPS of PrP$^C$ in living myoblasts and thus regulates skeletal muscle cell autophagy, we performed western blot analysis for the autophagy markers ATG5 and LC3B in the aforementioned cell lines (Fig. 8l). Incubation of C2C12 cells stably expressing mCherry-Cry2-WT PrP$^C$ with 1,6-hexanediol inhibited the phase separation ability of PrP$^C$ in cells (Fig. 8k) and thus caused a significant increase in ATG5 and LC3B levels as proxies for autophagy (Fig. 8m, n). Transfection of C2C12 myoblasts stably expressing mCherry-Cry2-WT PrP$^C$ with miR-214-3p enhanced the phase separation ability of PrP$^C$ in cells (Fig. 8j), thus inhibiting autophagy (Fig. 8m, n). Therefore, the recruitment of miR-214-3p into PrP$^C$ condensates enhances the LLPS of PrP$^C$ in skeletal muscle cells and results in the inhibition of autophagy.

### PrP$^C$ recruits miR-214-3p into its phase-separated condensate in living skeletal muscle cells, which in turn promotes pathological aggregation of PrP

Given that PrP$^C$ selectively recruits a subset of miRNAs into its phase-separated condensate, which in turn enhances not only in vitro LLPS of PrP$^C$ (Fig. 7) but also the LLPS of PrP$^C$ in skeletal muscle cells (Fig. 8), we

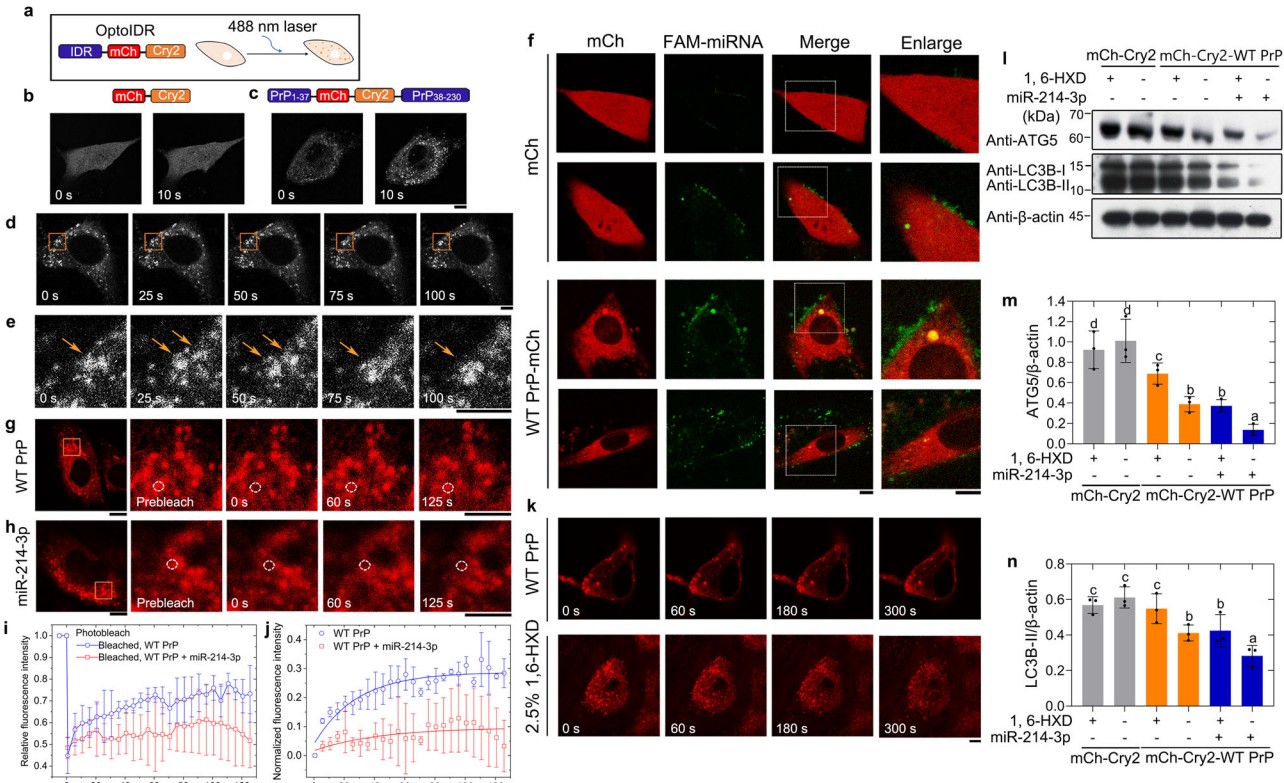

**Fig. 8 | PrP$^C$ recruits miR-214-3p into its phase-separated condensate in living myoblasts, which results in the inhibition of autophagy. a** Schematic of the optoIDR assay, depicting recombinant protein with an IDR (blue), mCherry (red), and Cry2 (orange) expressed in cells exposed to 488-nm laser. **b–e** Time-lapse images of living C2C12 cells expressing mCherry-Cry2-WT PrP$^C$ construct containing PrP$_{1-37}$ IDR (blue) linked to mCherry (red), Cry2 (orange), and PrP$_{38-230}$ IDR (blue) (**c–e**), and mCherry-Cry2 fusion alone as a control (**b**). A droplet fusion event occurs in the region (orange box, (**d**)); and orange arrows, (**e**). Cells were subjected to laser excitation every 2 s for the indicated time. **f** Representative live-cell images of PrP$^C$-miRNA speck formation when FAM-labeled miRNA (green) was transfected into C2C12 cells expressing mCherry or WT PrP$^C$-mCherry (red), among ≥ 10 cells. FRAP analysis on the selected liquid droplets of PrP$^C$ (red) in living C2C12 cells expressing mCherry-Cry2-WT PrP$^C$ transfected without (**g**) or with (**h**) miRNA before (prebleach), during (0 s), and after photobleaching (60 and 125 s). Dashed

white circle highlights the punctum undergoing targeted bleaching. Scale bars, 7.5 μm (**b–h**). Quantification of FRAP data (**i**) and normalized fluorescence intensity (**j**) of WT PrP$^C$ puncta (blue circle) and WT PrP$^C$ puncta + miR-214-3p (red square) were expressed as the mean ± SD (with error bars) of values (n = 3). The solid lines show the best single exponential fit (**j**). **k** Time-lapse images of living C2C12 cells expressing mCherry-Cry2-WT PrP$^C$ treated with 2.5% 1,6-hexanediol. Scale bar, 5 μm. **l** Western blot for ATG5 and LC3B in C2C12 cells expressing mCherry-Cry2-WT PrP$^C$ transfected without (-) or with (+) miRNA and then treated with (+) or without (-) 1,6-hexanediol. The relative amount of ATG5 (**m**) and LC3B-II (**n**) in the aforementioned cells (solid black circles shown in scatter plots) was expressed as the mean ± SD (with error bars) of values obtained in three independent experiments. One-way two-sided ANOVA and multiple comparisons with no adjustments were performed by SPSS 19.0, and different letters indicate significant differences at the level of p < 0.05.

predicted that miR-214-3p, an example of these PrP$^C$-associated miR-NAs, might regulate the subsequent PrP aggregation in skeletal muscle cells. We next employed western blotting and immunogold electron microscopy[79–81] to test this hypothesis. To further test the functional relevance of the observations that we made in vitro and in vivo with PrP$^C$ LLPS, we used C2C12 cells stably expressing mCherry-Cry2-WT PrP$^C$, which were cultured until their confluence reached 85% (Fig. 9a, b) or 80% (Fig. 9c), transfected with or without 10 μM miR-214-3p for 30 min, and cultured for 12 h after activation by 488-nm laser for 10 min, using C2C12 myoblasts stably expressing mCherry-Cry2 as a control. The sarkosyl-insoluble pellets from the above cells were probed with anti-PrP antibody, and the corresponding cell lysates were probed using anti-PrP antibody and anti-β-actin antibody (Fig. 9a). The above cells were also detected by a 3-(4,5-dimethylthiazol-2-yl)-2,5-diphenyltetrazolium bromide (MTT) reduction assay (Fig. 9c). Transfection of 10 μM miR-214-3p significantly promoted light-activated pathological aggregation of PrP and significantly increased PrP toxicity in C2C12 myoblasts stably expressing WT PrP$^C$ (Fig. 9a–c).

To ascertain the nature of light-activated pathological aggregates of PrP in skeletal muscle cells, we conducted immunogold electron microscopy. C2C12 cells stably expressing mCherry-Cry2-WT PrP$^C$

were cultured until their confluence reached 85%, transfected without (Fig. 9d) or with (Fig. 9e) 10 μM miR-214-3p for 30 min, cultured for 12 h after activation by 488-nm laser for 10 min, and labeled by gold particles conjugated with anti-PrP antibody. The amyloid fibrils in the above cell samples were recognized by anti-PrP antibody and decorated with 10-nm gold labels, and the skeletal muscle cells transfected with 10 μM miR-214-3p produced much more amyloid fibrils than those transfected without miRNA (Fig. 9d, e).

The PrP aggregates in C2C12 cells were demonstrated by the amount of PrP in sarkosyl-insoluble pellets from cells overexpressing the chimeric mCherry-Cry2-WT PrP$^C$ (Fig. 9a), and were further proved by the amount of PrP in sarkosyl-insoluble pellets from cells overexpressing WT PrP$^C$ (Fig. 9f). C2C12 myoblasts stably overexpressing full-length WT mouse PrP$^C$ were cultured until their confluence reached 85% and then incubated with differentiation medium for 2 days, transfected without or with miR-214-3p, and cultured for 48 h (Fig. 9f, g). The sarkosyl-insoluble pellets from the above cells were probed with anti-PrP antibody, and the corresponding cell lysates were probed using anti-PrP antibody and anti-β-actin antibody (Fig. 9f). In C2C12 cells stably expressing WT PrP$^C$ incubated with differentiation medium for 4 days, miR-214-3p specifically interacted with PrP$^C$ in the

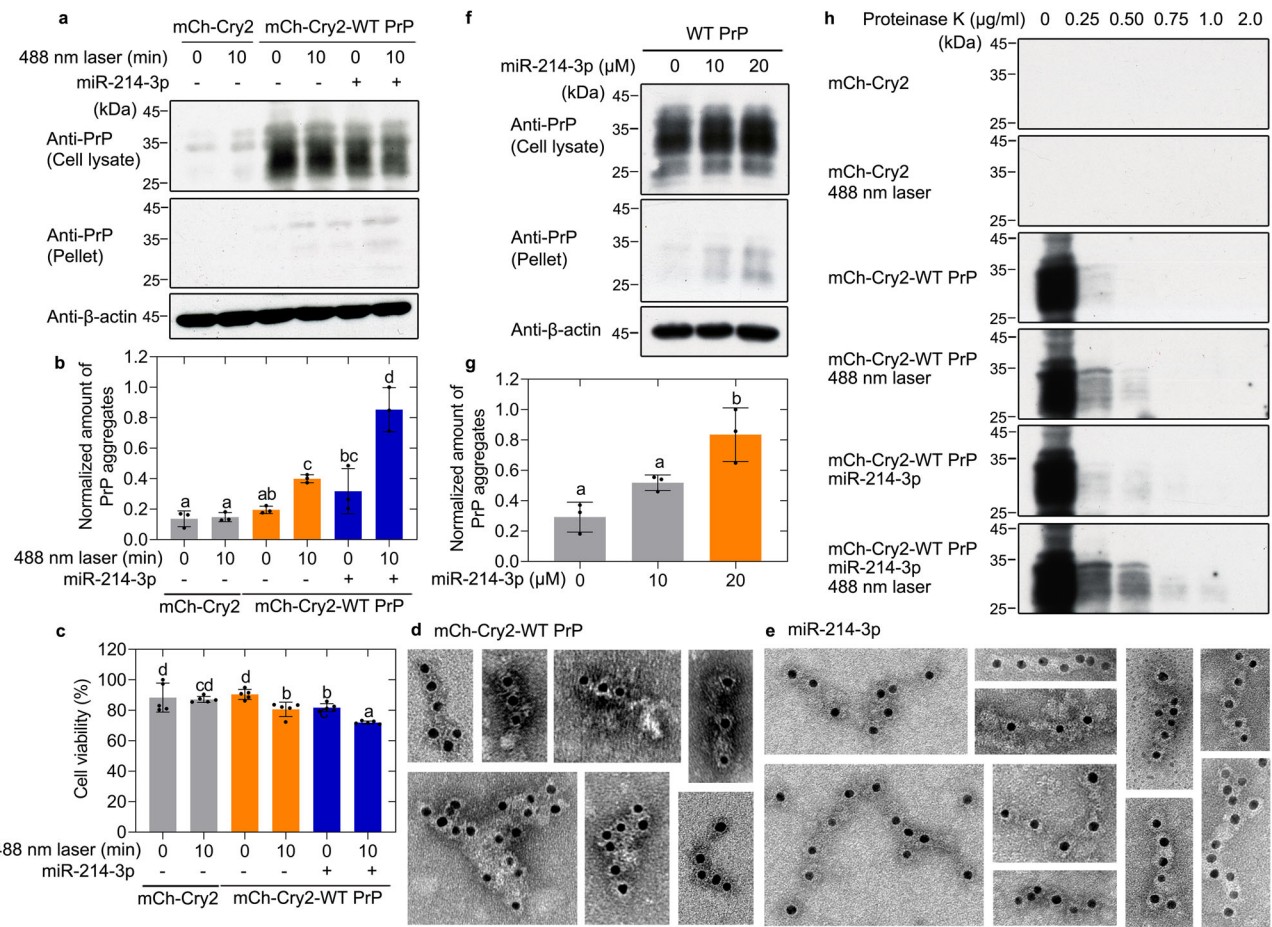

**Fig. 9 | PrP$^C$ recruits miR-214-3p into its phase-separated condensate in living skeletal muscle cells, which in turn promotes pathological aggregation of PrP.** **a** Western blot for PrP in the sarkosyl-insoluble pellets and the corresponding cell lysates from C2C12 cells stably overexpressing mCherry-Cry2-WT PrP$^C$ transfected without (-) or with (+) 10 μM miR-214-3p and cultured for 12 h after activation by 488-nm laser for 10 min. β-actin served as the protein loading control. **b** The normalized amount of insoluble PrP aggregates in the aforementioned cells (solid black circles shown in scatter plots) was expressed as the mean ± SD (with error bars) of values obtained in three independent experiments. **c** The cell viability (%) (solid black circles shown in scatter plots) was measured by MTT reduction assay and expressed as the mean ± SD (with error bars) of values obtained in five independent experiments. C2C12 cells expressing mCherry-Cry2 (gray) or expressing mCherry-Cry2-WT PrP$^C$ transfected without (-) (orange) or with (+) (blue) 10 μM miR-214-3p (**b**, **c**). Immunogold electron microscopy of PrP fibrils purified from the same C2C12 myoblasts stably expressing mCherry-Cry2-WT PrP$^C$ transfected without (**d**) or with (**e**) 10 μM miR-214-3p as in (**a**), and labeled by gold particles conjugated with anti-PrP antibody. Scale bars, 100 nm. **f** Western blot for PrP in the sarkosyl-insoluble pellets and the corresponding cell lysates from C2C12 myoblasts stably overexpressing WT PrP$^C$ incubated with differentiation medium for 2 days, transfected without or with 10 or 20 μM miR-214-3p and cultured for 48 h. β-actin served as the protein loading control. **g** the same as in (**b**). C2C12 cells expressing WT PrP$^C$ transfected without or with 10 (gray) or 20 μM (orange) miR-214-3p. **b**, **c**, **g** One-way two-sided ANOVA and multiple comparisons with no adjustments were performed by SPSS 19.0, and different letters indicate significant differences at the level of $p < 0.05$. **h** Western blot for PrP in the cell lysates from the same C2C12 cells stably overexpressing mCherry-Cry2-WT PrP$^C$ transfected without or with 10 μM miR-214-3p as in (**a**), and digested with 0.25, 0.50, 0.75, 1.0, and 2.0 μg/ml proteinase K.

cytoplasm and transfection of 20 μM miR-214-3p significantly promoted pathological aggregation of PrP (Fig. 9g).

Because PrP aggregates are more resistant (than normal PrP$^C$) to digestion with modest amounts of proteases (such as proteinase K)[7,8], we wanted to determine whether the PrP aggregates in C2C12 cells show some level of protease resistance. C2C12 cells stably expressing mCherry-Cry2-WT PrP$^C$ were cultured until their confluence reached 85%, transfected without or with 10 μM miR-214-3p for 30 min, and cultured for 12 h after activation by 488-nm laser for 10 min, and mCherry-Cry2 fusion alone was used as a control (Fig. 9h). The aforementioned cells were digested with various concentrations of proteinase K and probed with anti-PrP antibody. We found that PrP aggregates in the cell lysates from cells transfected without miRNA were completely digested with a proteinase K concentration as low as 0.75 μg/ml. In sharp contrast, PrP aggregates in the cell lysates from cells transfected with 10 μM miR-214-3p became resistant to 1.0 μg/ml

proteinase K, producing a fragment that migrated at 27–30 kDa (Fig. 9h). Therefore, WT PrP$^C$ can be induced by light to form liquid condensates in C2C12 cells; such PrP$^C$ condensates recruit miR-214-3p, which in turn promotes the LLPS of PrP$^C$ in C2C12 cells to evolve into PrP aggregates with some level of protease resistance.

Altogether these data demonstrate that PrP$^C$ recruits miR-214-3p into its phase-separated condensate in living skeletal muscle cells, which in turn promotes pathological aggregation of PrP to form protease-resistant aggregates.

## Discussion

Because the cellular prion protein PrP$^C$ has important functions in normal cellular physiology, including muscle cell differentiation and regeneration[15,16,18], it has generally been thought that PrP$^C$ dysfunction might be responsible for skeletal muscle cell death in patients with inclusion-body myositis, dermatomyositis, and

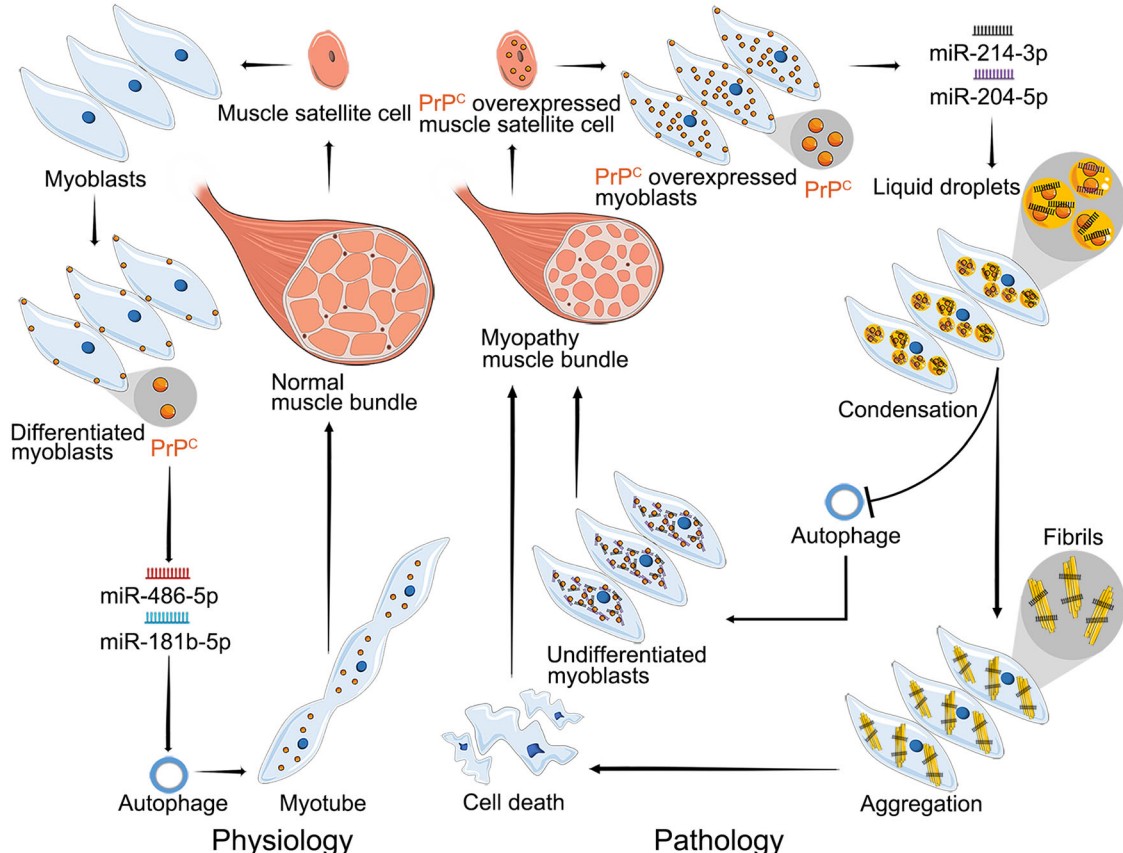

**Fig. 10 | A hypothetical model shows how excess PrP^C inhibits muscle cell differentiation via miRNA-enhanced LLPS of PrP^C implicated in myopathy.** Under pathological conditions (right), PrP^C (orange) is overexpressed in muscle satellite cells (orange) and myoblasts (pale blue). Then, PrP^C selectively recruits miR-214-3p (black) and miR-204-5p (violet) into phase-separated condensates (gold balls) in living myoblasts to block autophagy and preserve myoblasts undifferentiated. This recruitment in turn greatly enhances PrP^C liquid phase condensation and the subsequent aggregation to produce PrP fibrils (gold bars), resulting in skeletal muscle cell death and muscle bundle formation in myopathy patients. Under physiological conditions (left), PrP^C (orange) is expressed in myoblasts (pale blue) and enhances muscle cell differentiation by selectively interacting with miR-486a-5p (red) and miR-181b-5 (blue) to significantly promote autophagy, and finally, muscle bundles are assembled to create the whole muscle.

other myopathies[19,20,22]. In this study, we detected a much higher level of PrP^C in skeletal muscles from six myopathy patients with dermatomyositis, neurogenic myopathy, or muscular atrophy than in skeletal muscles from one healthy control, two controls with lipid storage myopathy, and one control with glycogen storage disease. Accumulating pieces of evidence point to a crucial role of autophagy in myoblast differentiation[25–27], whereas impaired autophagy is observed in aged muscle satellite cells[28]. We showed that in cell culture both overexpression and knockout of PrP^C impaired myoblast differentiation. To decode the mechanism, we examined autophagy in cells and found autophagy to be inhibited and enhanced by overexpression and knockout of PrP^C, respectively. Interestingly, PrP^C controls the distribution of caveolin 1 between lipid raft domains on the cell membrane and the cytoplasm, where caveolin one can function to impair the ATG12-ATG5 complex and thus inhibit autophagy progression[82].

Several in vitro studies have described physical interactions between PrP and different classes of molecules, among which nucleic acids are highlighted as potential PrP molecular partners[47,50,52,53,55,56,83,84]. Given that miRNAs play important roles in regulating differentiation, atrophy, and regeneration of skeletal muscle via interaction with specific proteins[35–38], we predicted that PrP^C might physically interact with miRNAs to modulate myoblast differentiation. In this study, we found subsets of miRNAs associating with endogenous or overexpressed PrP^C, and then focused on a couple of miRNAs (miR-214-3p and miR-204-5p) specifically associated with overexpressed PrP^C. We showed that a higher level of PrP^C colocalized with these miRNAs in myopathy muscle samples, and that overexpressed PrP^C colocalized with these miRNAs in skeletal muscle cells. PrP^C increases the stability of the mature forms of these miRNAs and enhances miRNA repression of their downstream mRNA targets to significantly inhibit autophagy and myoblast differentiation. Therefore, PrP^C regulates autophagy and differentiation of skeletal muscle cells via multiple mechanisms, the first being physical interaction with these specific miRNAs.

In this work, we report that PrP^C, a glycoprotein existing in cytoplasmic form during myoblast differentiation, exhibits disparate propensities to phase separate with miRNA. We show that PrP^C undergoes LLPS in vitro and within cells. PrP^C condensates selectively recruit a couple of miRNAs (miR-214-3p and miR-204-5p) in vitro and within skeletal muscle cells, which in turn strongly promotes the LLPS of PrP^C under both conditions. PrP^C recruits and concentrates miR-214-3p into phase-separated condensates (puncta) in the cytosol of skeletal muscle cells, which in turn mediates enhanced PrP^C condensation, rendering the resulting liquid droplets fibril-like. Mutations of the GCAG sequences in miR-214-3p that block the specific interaction of PrP^C with miR-214-3p impair the incorporation of miR-214-3p into PrP^C condensates. Overall, our results show that PrP^C, a potential RNA-binding protein, undergoes miRNA-mediated LLPS in vitro and within cells. The

recruitment of miR-214-3p into PrP$^C$ condensates enhances the LLPS of PrP$^C$ in skeletal muscle cells, results in the inhibition of autophagy, and promotes pathological aggregation of PrP. Intriguingly, the RNA-binding protein YBX1 undergoes LLPS in vitro and within cells, and YBX1 condensates selectively recruit miR-223 in vitro and into exosomes secreted by cultured cells[85]. In vitro LLPS of PrP$^C$ is modulated by three types of RNA molecules (polyU RNA, crude tRNA, and yeast total RNA)[52,55,56]. It should be mentioned that self-complementary RNA structures also play a role in LLPS by imparting an "identity" to biomolecular condensates. This identity prevents the merging of biomolecular condensates containing different or dissimilar RNAs[86]. Thus, the specific properties of miRNAs included in PrP$^C$ condensates influence the liquidness and organization of PrP$^C$ condensates.

In summary, our results describe a model to underpin molecular hypotheses of how excess PrP$^C$ inhibits muscle cell differentiation via miRNA-enhanced LLPS of PrP$^C$ implicated in myopathy (Fig. 10). We report interesting findings on the roles of PrP$^C$, LLPS composed of PrP$^C$ and miRNAs, and certain miRNAs on myopathies associated with accumulation or overexpression of PrP$^C$. Importantly, we show that under pathological conditions, PrP$^C$, an important protein for myoblast differentiation, is overexpressed in muscle satellite cells and myoblasts and selectively interacts with a couple of miRNAs, miR-214-3p and miR-204-5p. Then, PrP$^C$ selectively recruits these miRNAs into its phase-separated condensate in living myoblasts to significantly inhibit autophagy and preserve myoblasts undifferentiated. This recruitment in turn enhances the LLPS of PrP$^C$, resulting in the abnormal aggregation of the protein, skeletal muscle cell death, and the subsequent muscle bundle formation in myopathy patients characterized by incomplete muscle regeneration (Fig. 10). Therefore, accumulation or overexpression of PrP$^C$ is toxic to myoblasts via impairing autophagy and differentiation, and that miR-214-3p and miR-204-5p enhance the PrP toxicity. Under physiological conditions, when myotubes are damaged, muscle satellite cells differentiate into myoblasts, PrP$^C$ is expressed in myoblasts and enhances muscle cell differentiation by selectively interacting with miR-486a-5p and miR-181b-5 to significantly promote autophagy, and finally, muscle bundles are assembled to create the whole muscle (Fig. 10). We characterized the molecular basis of PrP$^C$ pathogenesis in myopathies based on its selective inclusion of miR-214-3p into phase-separated condensates, resulting in enhanced LLPS of PrP$^C$ and PrP aggregation, as well as inhibition of autophagy. These results provide insights on the possible cytotoxic mechanisms of accumulation or overexpression of PrP$^C$ in skeletal muscles. The observation of many colocalized dots of PrP$^C$ and miR-214-3p in the cytoplasm of a group of myopathy patients provides clinical relevance. The selective interaction of miRNAs with PrP$^C$ during cell differentiation will be valuable to understanding the functional basis underlying LLPS of proteins and inspiring future research on protein condensation diseases caused by abnormal liquid-like or solid-like states of proteins[87] and regulated by RNA[43–46].

## Methods

### Ethics statement
The study complies with all relevant ethical regulations. The study is based on analyses of skeletal muscle samples from two patients with dermatomyositis, two patients with neurogenic myopathy, and two patients with muscular dystrophy, and from one healthy individual, two patients with lipid storage myopathy, and one patient with glycogen storage disease (controls). The patient characteristics are described in Supplementary Table 1. Tissue materials were collected at the Department of Neurology, Renmin Hospital of Wuhan University after obtaining informed consent from the patients or their relatives, who did not receive any compensation. We have obtained consent to publish information that identifies individuals (including three or more indirect identifiers such as exact age, sex, and medical centre the study participants attended or rare diagnosis). All relevant regulations and

legal requirements, including ethical approval from relevant authorities at Wuhan University, were observed during material collection. The biochemical work at Wuhan University was conducted based on a permission from the Wuhan University Ethics Committee (WAEF-2022-0073).

### Pathological samples miRNA FISH and immunocytochemistry
The skeletal muscles of patients with myopathy such as dermatomyositis, neurogenic myopathy, and muscular dystrophy were collected at the Department of Neurology, Renmin Hospital of Wuhan University. The skeletal muscle samples were fixed with isopentane for 2-3 min, then frozen in liquid nitrogen, and sliced with a cryostat sectioning. Frozen skeletal muscle sections from six myopathy patients with dermatomyositis, neurogenic myopathy, or muscular atrophy, one healthy control, two controls with lipid storage myopathy, and one control with glycogen storage disease (Supplementary Table 1) were marked with hydrophobic circles using an immunohistochemical pen. H&E staining of the frozen skeletal muscle sections was conducted by following the manufacturer's instructions (Beyotime, C0105S). For small-RNA FISH, slices were permeabilized with cold 0.5% Triton X-100 in PBS at room temperature for 5 min. After being twice washed with PBS for 5 min, the fixed cells were incubated with the prehybridization buffer for 1 h at 55 °C. Prehybridized coverslips were incubated with a hybridization buffer (10% dextran sulfate in prehybridization buffer plus 10 ng/μl FAM-labeled miRNA probe) and covered with siliconized coverslips in a humidified chamber overnight at 55 °C. Coverslips were washed with buffer (2 × SSC, 30% formamide) for 10 min at 37 °C and then with 2 × SSC, 1 × SSC, 0.5 × SSC and 1 × PBS. Immunocytochemistry staining was then performed to achieve the purpose of double staining.

For immunostaining, slices blocked with 10% goat serum for 1 h at 37 °C. Primary antibody diluted in PBS containing 3% goat serum was applied to the slices and incubated in a humidified chamber overnight at 4 °C. After three times washes with PBS for 5 min, fluorescence-conjugated secondary antibody was applied to the cover slip and incubated in a dark room for 1 h. DAPI was then applied at the proper dilution; after being washed three times with PBS for 5 min and mounted with antifade mounting medium (Beyotime), cells were subjected to a Leica TCS SP8 laser scanning confocal microscope (Wetzlar, Germany). The following primary antibodies were used: mouse anti-PrP monoclonal antibody 8H4 (Abcam, ab61409, 1:200) and Alexa Fluor 555-labeled donkey anti-mouse IgG (H + L) (Beyotime, A0460, 1:500).

### Immunohistochemistry
Slices were permeabilized with cold 0.5% Triton X-100 in PBS at room temperature for 5 min. After being twice washed with PBS for 5 min, the slices blocking with 10% normal goat serum for 1 h at 37 °C. Primary antibody diluted in PBS containing 3% goat serum was applied to the slices and incubated in a humidified chamber at 4 °C. After three times washes with PBS for 5 min, antibody detection was carried out using biotinylated horse anti-mouse IgG, SABC-HRP Kit (Beyotime, P0603), DAB Horseradish Peroxidase Color Development Kit (Beyotime, P0202), and Hematoxylin Staining Solution (Beyotime, C0107). The slices were dehydrated with ethanol (80%, 10 s; 90%, 10 s; and 100%, 10 s), then transparent with xylene for 5 min, repeated twice, mounted with DPX mountant, and observed under a Leica TCS SP8 laser scanning confocal microscope (Wetzlar, Germany). The following antibodies were used: mouse monoclonal NCAM antibody (Santa Cruz Biotechnology, sc-106, 1:50) and Biotin-labeled Goat Anti-mouse IgG (H + L) with high molar ratio (Beyotime, A0288, 1:100).

### Cell culture and myogenic differentiation
Murine-derived C2C12 myoblast cells (catalog number GDC0175) and HEK-293T cells (catalog number GDC0187) were obtained from China

Center for Type Culture Collection (CCTCC, Wuhan, China). C2C12 myoblast cells were cultured in minimum essential media and in Dulbecco's modified Eagle's medium (Gibco, Invitrogen), supplemented with 20% (v/v) fetal bovine serum (Gibco) and 1% penicillin-streptomycin in 5% $CO_2$ at 37 °C. For myoblast differentiation, C2C12 myoblasts at 90% confluency were switched to a differentiation medium, supplemented with 2% horse serum (Gibco) and 1% penicillin-streptomycin.

## Plasmids and transfection

Total RNA was extracted from C2C12 myoblasts using TRIzol reagent from Beyotime (Nantong, China) according to the instructions, and the total RNA was reverse transcribed into a cDNA library using First Strand cDNA Synthesis Kit (Beyotime). The open reading frame of mouse PrP was obtained by amplifying the cDNA library using PCR, and cloned into the pBABE vector (pBABE-PrP$^C$) and the pET-28a (Pet-28a-PrP$^C$). Genetic prion disease–related mutation F198S (pBABE-F198S) was constructed by site-directed mutagenesis using pBABE-PrP$^C$ template. The target DNA fragments were inserted into the retroviral vector, and the plasmids containing target DNA, pUMVC3 gag-pol, and pCMV-VSV-G were packaged in HEK-293T cells with TransIT-X2 (Mirus) according to the manufacturer's protocols. After 36 h of transfection, the viruses were harvested and filtered, and then C2C12 cells were infected with the packaged lentivirus twice for 12 h each with a 12-h interval. To establish the stable cell lines, puromycin was used to screen overexpressed cells. The guide RNAs (gRNAs) for mouse PrP were selected from the mouse GeCKO CRISPR library and cloned into pLenti-CRISPR-V2 vector (pLenti-CRISPR-V2-PrP$^C$) according to the instruction described[88]. The gRNA oligo and PCR primers were listed in Supplementary Data 2. The 3'UTRs of *ATG5* and *LC3B* were obtained by PCR amplification of cDNA and cloned into psiCHECK2 vector (luciferase reporter). mCherry-Cry2 on pHR-mCherry-Cry2 was cloned with the "GSGSGSGS" linker into position 37 of PrP$^C$ on the pBABE-PrP$^C$ vector (pBABE-mCherry-Cry2-PrP$^C$). mCherry on pHR-mCherry-Cry2 was cloned with the "GSGSGSGS" linker into position 37 of PrP$^C$ on the pBABE-PrP$^C$ vector (pBABE-mCherry-PrP$^C$). Plasmids and miRNA mimics or inhibitors were transfected into the cells with TransIT-X2 (Mirus) according to the manufacturer's protocols. Individual RNA sequences are listed in Supplementary Data 2.

## Western blotting

For analysis by western blotting and IP, C2C12 cells grown in a 6-well plate were washed twice with ice-cold PBS and lysed in 300 μl (per well) cell lysis buffer containing 1× protease inhibitor cocktail (Target Mol). For analysis by western blotting treated with trehalose, C2C12 cells were grown in a 6-well plate and cultured in a differentiation medium of 2% equine serum supplemented with 50 or 100 mM D-(+)-trehalose (MedChem Express, HY-N1132) for 5 days, and the medium was changed daily. Cells were washed twice with ice-cold PBS and lysed in 300 μl (per well) cell lysis buffer containing 1× protease inhibitor cocktail (Target Mol). For analysis by western blotting treated with 1,6-hexanediol (Sigma, H11807), C2C12 cells expressing the mCherry-Cry2-WT PrP$^C$ construct were cultured until their confluence reached 85% in 6-well plate, then transfected with 10 μM miR-214-3p using TransIT-X2 (Mirus) for 30 min in a growing medium, activation by 488-nm laser for 10 min, and then treated with 2.5% 1,6-hexanediol for 5 min. Cells were washed twice with ice-cold PBS and lysed in 300 μl (per well) cell lysis buffer containing 1× protease inhibitor cocktail (Target Mol).

The amount of loaded protein was normalized using a BCA Protein Quantification kit (Beyotime). The cell lysates were boiled in SDS-PAGE loading buffer for 10 min and then subjected to SDS-PAGE and probed with the following specific antibodies: mouse anti-PrP monoclonal antibody 8H4 (Abcam, ab61409, 1:5000), mouse anti-

MyHC antibody MF-20 (Developmental Studies Hybridoma Bank, MAB4470-SP, 1:1,000), mouse anti-MyoG antibody F5D (Santa Cruz Biotechnology, sc-12732, 1:500), mouse anti-β-actin (Beyotime AA128, 1:1000), rabbit anti-ATG5 antibody (Sigma SAB5700062, 1:1000), rabbit anti-LC3B antibody (Sigma, SAB1306269, 1:1000), Alexa-conjugated fluorescent secondary antibodies (Beyotime, rabbit A0208, 1:1000; mouse A0216, 1:1000).

## Cell immunocytochemistry and miRNA FISH

For immunostaining, C2C12 cells were seeded in 12-well plates, washed twice with PBS for 5 min, fixed with 4% paraformaldehyde for 30 min at room temperature, washed twice with PBS for 5 min and then permeabilized for 4 min on ice with 0.25% Triton X-100 in PBS. Cells were washed three times with PBS for 5 min, and blocked with 3% bovine serum albumin (BSA) for 30 min at 37 °C. Primary antibody diluted in PBS containing 3% BSA was applied to the cells and incubated for 3 h at 37 °C. After three times washes with PBS for 5 min, fluorescence-conjugated secondary antibody was applied to the cover slip and incubated in a dark room for 45 min. DAPI (Beyotime) was then applied at the proper dilution; after being washed three times with PBS for 5 min and mounted with antifade mounting medium (Beyotime), cells were subjected to a Leica TCS SP8 laser scanning confocal microscope (Wetzlar, Germany). The following primary antibodies were used: mouse anti-PrP monoclonal antibody 8H4 (Abcam, ab61409, 1:200), mouse anti-MyHC antibody MF-20 (Developmental Studies Hybridoma Bank MAB4470-SP, 1:200), rabbit anti-ATG5 antibody (Sigma, SAB5700062, 1:200), rabbit anti-LC3B antibody (Sigma, SAB1306269, 1:200), Alexa Fluor 488-labeled goat anti-rabbit IgG (H + L) (Beyotime, A0423, 1:500), and Alexa Fluor 555-labeled donkey anti-mouse IgG (H + L) (Beyotime, A0460, 1:500).

Small-RNA FISH was performed as described previously[89] with minor modifications. C2C12 cells were seeded onto a sterile cover slip positioned in the bottom of a well in a 12-well dish and differentiated with 2% horse serum. Cells rinsed twice with PBS for 5 min, fixed in 4% paraformaldehyde in PBS for 30 min at room temperature, rinsed twice in PBS for 5 min and then permeabilized with cold 0.5% Triton X-100 in PBS at room temperature for 5 min. After being twice washed with PBS for 5 min, the fixed cells were incubated with a pre-hybridization buffer (2 × SSC, 1 × Denhardt's solution, 50% formamide, 10 mM EDTA, 100 μg/ml yeast tRNA, 0.01% Tween 20, and 2 U/μl RNase inhibitor, Beyotime) for 1 h at 37 °C. Prehybridized coverslips were incubated with a hybridization buffer (10% dextran sulfate in pre-hybridization buffer plus 40 ng/μl FAM-labeled miRNA probe) in a humidified chamber overnight at 37 °C. Coverslips were washed with 2 × SSC, 1 × SSC, 0.5 × SSC and 1 × PBS for 5 min. Immunocytochemistry staining was then performed to achieve the purpose of double staining as follows. C2C12 cells (control) and C2C12 cells stably expressing full-length wild-type mouse PrP$^C$ upon differentiation for 4 days, in which miRNA was detected by FISH (green) using FAM-labeled miRNA probe, were immunostained with the anti-PrP antibody 8H4 (red) and stained with DAPI (blue). Images of FAM-labeled miRNA (green) and PrP$^C$ (red) were captured using a Leica TCS SP8 laser scanning confocal microscope (Wetzlar, Germany).

## Immunoprecipitation (IP)

Primary antibodies or IgG were added into Protein A + G Agarose beads (Fast Flow for IP) (Beyotime) on a rotator for 3 h at 4 °C. C2C12 Cells were lysed in cell lysis buffer containing 1× protease inhibitor cocktail (Target Mol) for western and IP, and the lysates were centrifuged at 12,000 g for 10 min at 4 °C to remove cell debris. An input of collected supernatant was set aside for subsequent analysis. Cell lysate was added into the mixture on a rotator overnight at 4 °C. Finally, immunocomplexes eluted from beads were detected using indicated antibodies by western blotting.

## RNA immunoprecipitation (RIP)

C2C12 cells incubated with differentiation medium for 4 days and cultured in two 150-mm plates were washed twice with cold PBS and then crosslinking with 0.1% formaldehyde for 10 min at room temperature. Cells were added into 1.25 M glycine solution for 5 min and then washed 3 times with PBS for 5 min. Scrapped cells were harvested by centrifugation at 500 $g$ for 5 min at 4 °C. Crosslinked cells were lysed in 1 ml of RIP lysis buffer containing 25 mM Tris-HCl (pH 7.4), 150 mM NaCl, 1% NP-40, 1 mM EDTA, 5% glycerin, 400 U/ml RNasin ribonuclease inhibitor, 1 mM dithiothreitol (DTT), and 1× protease inhibitor cocktail (Target Mol) on ice for 20 min. Lysates were centrifuged at 12,000 $g$ at 4 °C for 10 min to remove cell debris. An input of collected supernatant was set aside for subsequent analysis. Each sample was then divided and incubated with 15 µg of either mouse anti-PrP monoclonal antibody 8H4 or mouse IgG on a rotator for 5 h at 4 °C. Complexes were pulled down by incubation with Protein A + G Agarose beads (Fast Flow for IP) (Beyotime) on a rotator for 3 h at 4 °C. Beads were washed twice with lysis buffer and washed twice with RIP wash buffer containing 350 mM NaCl. 150 µl of proteinase K buffer containing 1.2 mg/ml proteinase K, 10 mM Tris-HCl (pH 8.0), 5 mM EDTA, 0.5% SDS, 400 U/ml RNasin ribonuclease inhibitor, and 1 mM DTT for 5 min at 55 °C to reverse formaldehyde crosslinking. Co-precipitated RNAs were extracted from beads and used for construction of small-RNA libraries or analysis by RT−qPCR using specific primers.

## RIP-seq data analysis

Reads of small RNA-seq were trimmed using the Cutadapt program and then mapped to the mouse genome (mm10) using Bowtie v0.9.6 (ref. 90) with parameters "-n 0 -l 20 -k 1 −best". Read counts of mature miRNAs were calculated using the featureCounts program[91] with parameters '−fracOverlap 0.8−fracOverlapFeature 0.8 -s 1 -M'. The enrichment of miRNAs from RIP over input and pellet over total lysates were calculated with the edgeR package[92]. We defined the enriched miRNAs as CPM > 50, $\log_2$ fold change > 0.5 and $p$ < 0.05.

## RT−qPCR of miRNAs

The Qiagen miScript II RT Kit was used to quantify miRNAs. After TRIzol (Beyotime) extraction of total RNA, mature miRNAs were poly-adenylated by PolyA polymerase and reverse transcribed into cDNAs by using an oligo-dT primer provided in the kit. The oligo-dT primer contained a 3 degenerate anchor and a universal tag sequence at the 5′ end, allowing quantitative analysis of mature miRNA by real-time PCR using the universal primer and a miRNA-specific primer. Quantitative PCR was carried out with a 1:10 dilution of cDNA, 2 × SYBR Green PCR Mix, and 10 × miScript universal primers included in the kit in combination with 10 × miRNA-specific primers (listed in Supplementary Data 3). The U6 snRNA primer from Qiagen was used for normalization, and ΔCt was calculated to derive relative expression. C2C12 cells stably expressing full-length wild-type mouse PrP$^C$ and C2C12 cells KO for PrP$^C$ were incubated with a differentiation medium for 4 days and then treated with 20 mg/ml α-amanitin, an inhibitor of RNA polymerase II. The above cell lines were lysed every 3 h for up to 12 h. The relative stability of miRNAs in the above cell lines was determined as the relative enrichment (to U6 snRNA) of miRNAs by RT−qPCR.

## Small-RNA pulldown

For the small-RNA pulldown assay, 5′ biotinylated miRNA baits were commercially synthesized (listed in Supplementary Data 4). For the preparation of cell lysate, a 155-mm plate of confluent C2C12 cells incubated with differentiation medium for 4 days were harvested by centrifugation at 500 $g$ for 5 min at 4 °C. Crosslinked cells were lysed in 1 ml of pulldown lysis buffer containing 25 mM Tris-HCl (pH 8.0), 150 mM NaCl, 1% NP-40, 1 mM EDTA, 5% glycerin, 400 U ml$^{-1}$ RNasin ribonuclease inhibitors, 1 mM DTT on ice for 20 min. Insoluble material was removed by centrifugation at 20,000 $g$ for 30 min at 4 °C. Lysates

were centrifuged at 12,000 $g$ for 10 min at 4 °C to remove cell debris. An input of collected supernatant was set aside for subsequent analysis. For each pulldown assay, 40 µl of biotin-labeled small RNA (20 µM) was incubated with precleared lysate overnight at 4 °C with rotation. One-hundred microliters of magnetic streptavidin beads (Beyotime) were washed with pulldown lysis buffer twice and incubated with the RNA lysates for 3.5 h at 4 °C with rotation. The beads were washed with cold wash buffer I containing 300 mM NaCl, washed with cold wash buffer II containing 0.05% tween-20, washed with cold pulldown lysis buffer, and then boiled in 40 µl SDS loading buffer for analysis by western blotting.

## Luciferase reporter assay

For luciferase assays, C2C12 cells were seeded in 12-well plates and co-transfected with 1.25 µg of luciferase reporter plus 10 µM miRNA mimics or NC. After 36 h, cells were harvested for luciferase assays using the Dual Luciferase Reporter Gene Assay Kit (Yeasen, 11402ES60). A Cytation 3 Cell Imaging Multi-Mode Reader (BioTek) was used to collect light generated by *Renilla* or firefly.

## Protein purification

The open reading frame of mouse PrP was obtained by amplifying the cDNA library using PCR, and cloned into the pBABE vector (pBABE-PrP$^C$) and the pET-28a (Pet-28a-PrP$^C$). Pet-28a-PrP$^C$ plasmid was transformed into *E. coli*. Recombinant full-length wild-type mouse PrP was expressed in *E. coli* BL21 (DE3) cells (Novagen, Merck, Darmstadt, Germany) and purified by high-performance liquid chromatography on a C4 reverse-phase column (Shimadzu, Kyoto, Japan) as described by Bocharova et al.[93] and Zhou et al.[94] After purification, recombinant wild-type mouse PrP$^C$ was dialyzed against 1 × PBS (pH 7.4) for 24 h, concentrated, filtered, and stored at −80 °C. SDS-PAGE and mass spectrometry were used to confirm that the purified wild-type mouse PrP was single species with an intact disulfide bond. We used a Nano-Drop OneC Microvolume UV-Vis Spectrophotometer (Thermo Fisher Scientific) to determine the concentration of wild-type mouse PrP$^C$, using its absorbance at 280 nm and the molar extinction coefficient calculated from the composition of the protein (http://web.expasy.org/protparam/).

## Liquid-droplet formation

The freshly bacterial-purified wild-type mouse PrP$^C$ was incubated with TAMRA (red fluorescence, excitation at 561 nm) at a PrP$^C$: TAMRA molar ratio of 1:3 for 1 h. These labeled proteins were filtered, concentrated to 245 µM in a centrifugal filter (Millipore) and diluted in 1 × PBS (pH 7.4). In total, 25, 35, 45, 50, 65, and 80 µM wild-type mouse PrP$^C$ labeled by TAMRA were incubated with 1 × PBS (pH 7.4) on ice to induce LLPS[48] for 5 min. In total, 50 µM wild-type mouse PrP$^C$ labeled by TAMRA was incubated with 1 × PBS (pH 7.4) containing 0, 1.25, 2.5, 5, 10 or 20 µM miRNA on ice to induce LLPS for 5 min. The miRNAs include a subset of PrP$^C$-bound miRNAs, such as miR-214-3p and miR-204-5p, and nonspecific miR-183-5p (negative control). Liquid droplets of PrP$^C$ (protein condensates) formed in 1 × PBS containing 0−20 µM miRNA were observed by a Leica TCS SP8 laser scanning confocal microscope (Wetzlar, Germany) with excitation at 561 nm. In total, 50 µM wild-type mouse PrP$^C$ labeled by TAMRA (red fluorescence) was incubated with 1 × PBS containing 10 µM FAM-labeled miRNA (green fluorescence) on ice to induce LLPS for 5 min. The FAM-labeled miR-NAs, including FAM-labeled miR-214-3p, FAM-labeled miR-204-5p, FAM-labeled miR-83-5p, and FAM-labeled mutant miR-214-3p (listed in Supplementary Data 4), were synthesized. PrP$^C$ demixed droplets (red) fused with droplets of FAM-labeled miRNA (green) were observed by a Leica TCS SP8 laser scanning confocal microscope (Wetzlar, Germany), with excitation at 561 nm and 488 nm, respectively. All phase separation experiments were performed at least three times and were pretty reproducible.

## FRAP

In total, 50 µM wild-type mouse PrP[C] labeled by TAMRA was incubated with 1 × PBS (pH 7.4) or incubated with the same buffer further containing 10 µM miRNA on ice to induce LLPS for 10 min. Liquid droplets of wild-type PrP[C] were observed by a Leica TCS SP8 laser scanning confocal microscope with excitation at 561 nm. For each droplet, a square was bleached at 100% transmission for 320 ms, and post-bleaching time-lapse images were collected (100 frames, 320 ms per frame). Images were analyzed using Zen (LSM 880 confocal microscope manufacturer's software). All FRAP experiments were repeated three times and the results were reproducible.

## OptoIDR assays

We employed an optogenetic tool that uses blue light (488-nm laser) to activate IDR-mediated LLPS of proteins in living C2C12 cells[76–78]. C2C12 cells were transfected with optoIDR plasmids (pBABE-mCherry-PrP[C]). At 24 h post-transfection, cells were plated at 35-mm confocal dishes. After another 24 h, images were captured using a Zeiss LSM 880 with Airyscan confocal microscopy. Unless indicated otherwise, droplet formation was induced with 488-nm light pulses every 2 s for the duration of the imaging as indicated, and images were taken every 2 s. Fluorescence from mCherry was excited with 561-nm light.

## Live-cell imaging

Cells were grown on chambered cover 35 mm confocal dishes to an appropriate density. Cells were transfected with FAM-labeled miR-214-3p using TransIT-X2 (Mirus). Live-cell images of mCherry-PrP[C] (red)-FAM-labeled miR-214-3p (green) speck formation were captured after 30 min using a Zeiss LSM 880 with Airyscan confocal microscopy with a 63 × oil objective and with excitation at 561 nm and 488 nm, respectively. For 1,6-hexanediol treatment experiments, cells were also grown on chambered cover 35 mm confocal dishes to an appropriate density. Then, 105 µl of 50% 1,6-hexanediol was quickly added into the plates. Live-cell images of mCherry-Cry2-PrP[C] (red) were captured using a Zeiss LSM 880 with Airyscan confocal microscopy with a 63 × oil objective and with excitation at 561 nm. Images were analyzed using Zen (LSM 880 confocal microscope manufacturer's software).

## Cellular FRAP Assays

Cells were grown on chambered cover 35 mm confocal dishes for 24 h, transfected with 10 µM miR-214-3p using TransIT-X2 (Mirus) for 30 min in a growing medium, and subjected to laser excitation for 10 s. FRAP was then done on a Zeiss LSM 880 with Airyscan confocal microscopy with a 561 nm laser. Bleaching was undertaken over a -1 mm radius using 80% laser power, time 0 indicated the start of recovery after photobleaching, and images were collected every 5 s. Fluorescence intensity was measured using ZEN (LSM 880 confocal microscope manufacturer's software). All FRAP experiments were repeated three times and the results were reproducible.

## Sarkosyl-insoluble western blotting

Sarkosyl-insoluble western blotting was used to investigate light-activated pathological aggregation of PrP in skeletal muscle cells. C2C12 cells stably overexpressing mCherry-Cry2-wild-type PrP[C] were cultured until their confluence reached 85% in 60-mm plates, then transfected with 10 µM miR-214-3p using TransIT-X2 (Mirus) for 30 min in a growing medium, and cultured for 12 h after activation by 488-nm laser for 10 min, using C2C12 cells stably overexpressing mCherry-Cry2 and cells not activated by 488-nm laser as controls. Sarkosyl-insoluble western blotting was also used to investigate pathological aggregation of PrP in skeletal muscle cells by PrP specific binding to miRNA. C2C12 myoblasts stably expressing WT mouse PrP[C] were cultured until their confluence reached 85% in 60-mm plates and then incubated with differentiation medium for 2 days, transfected without or with miR-214-3p using TransIT-X2 (Mirus) and cultured for 48 h.

Cell lysates from the above C2C12 stable cells were centrifuged at 17,000 g for 30 min at 4 °C to remove the cell debris. Sixty percent of the supernatant was incubated with 1% sarkosyl for 30 min at 25 °C. The mixture was then ultracentrifuged at 150,000 g for 30 min, and the supernatant was carefully removed. The sarkosyl-insoluble pellets were boiled in the SDS-PAGE loading buffer for 10 min. The forty percent of the supernatant, which served as the total protein sample, was also boiled in the SDS-PAGE loading buffer for 10 min. The samples were separated by 12.5% SDS-PAGE and then western blotted as follows. The samples were transferred to polyvinylidene difluoride membranes (Millipore). The membranes were blocked with 5% fat-free milk in 25 mM Tris-buffered saline buffer containing 0.047% Tween 20 (TBST). Then the sarkosyl-insoluble pellets from the above cells were probed with the anti-PrP antibody 8H4, and the corresponding cell lysates were probed using 8H4 and anti-β-actin antibodies. The amount of loaded protein was normalized using a BCA Protein Quantification kit (Beyotime). For calculating the amounts of sarkosyl-insoluble PrP, the ImageJ software (NIH) was used to assess the densitometry of PrP bands. The normalized amount of insoluble PrP aggregates in C2C12 cells stably expressing mCherry-Cry2-wild-type PrP[C] was determined as a ratio of the density of insoluble PrP aggregate bands over that of the total PrP bands in cell lysates.

## Cell viability assays

C2C12 myoblast cells were cultured in minimum essential media and in Dulbecco's modified Eagle's medium (Gibco, Invitrogen), supplemented with 20% (v/v) fetal bovine serum (Gibco) and 1% penicillin-streptomycin in 5% $CO_2$ at 37 °C. C2C12 cells stably overexpressing mCherry-Cry2-wild-type PrP[C] were cultured until their confluence reached 80%, then transfected with 10 µM miR-214-3p using TransIT-X2 (Mirus) for 30 min in a growing medium, and cultured for 12 h after activation by 488-nm laser for 10 min, using C2C12 cells stably overexpressing mCherry-Cry2 and cells not activated by 488-nm laser as controls. C2C12 cells were plated in 96-well plates in minimum essential medium. The MTT stock solution (5 mg/ml) was diluted with 1 × PBS and added into the well for 4 h until formazan was formed in cells. The final concentration of MTT was 0.5 mg/ml. Finally, the dark blue formazan crystal was dissolved with dimethyl sulfoxide, followed by measuring its absorbance at 492 nm using a Thermo Multiskan MK3 microplate reader (Thermo Fisher Scientific). Cell viability was expressed as the percentage ratio of the absorbance of wells containing the samples treated with miRNA to that of wells containing cells treated without miRNA.

## Immunogold electron microscopy of PrP fibrils

C2C12 cells stably overexpressing mCherry-Cry2-wild-type PrP[C] were cultured until their confluence reached 85%, then transfected with 10 µM miR-214-3p using TransIT-X2 (Mirus) for 30 min in a growing medium, and cultured for 12 h after activation by 488-nm laser for 10 min, using C2C12 cells stably overexpressing mCherry-Cry2 transfected without miRNA as a control. The above C2C12 stable cells were washed twice with cold PBS, digested with 2 ml (per dish) of pancreatic enzyme (Beyotime), and resuspended with 1 ml PBS and centrifuged at 500 g for 5 min at 4 °C to collect the cells. The cells were then resuspended with 450 µl of 10% sucrose solution in PBS (pH 7.4) containing 1× protease inhibitor cocktail (Target Mol). The mixtures were sonicated 10 times on ice at 50 W and 3 s/3 s, and then centrifuged at 17,000 g for 30 min at 4 °C to remove the cell debris. Sixty percent of the supernatant was incubated with 1% sarkosyl for 30 min at 25 °C. The mixture was then ultracentrifuged at 150,000 g for 30 min, and the supernatant was carefully removed. The sarkosyl-insoluble pellets were resuspended in PBS (50 µl). Sample aliquots of 10 µl were absorbed onto nickel grids for 1 min, and then wash twice with water for 30 s. After blocked with 0.1% BSA of 10 µl, samples on grids were incubated with 1/100 anti-PrP antibody 8H4 of 10 µl. After wash twice

with water for 30 s, 1/200 10-nm gold-labeled homologous secondary antibodies of 10 μl were used to incubate the grids for 20 min at 37 °C. Unbound gold-labeled homologous secondary antibodies were removed by washing with 200 μl of water drop by drop. Samples on grids were then stained with 2% (w/v) uranyl acetate for 1 min. The stained samples were examined using a JEM-1400 Plus transmission electron microscope (JEOL) operating at 100 kV.

## Proteinase K digestion assay

C2C12 cells stably expressing mCherry-Cry2-WT PrP$^C$ were cultured until their confluence reached 85% in 60-mm plates, then transfected without or with 10 μM miR-214-3p using TransIT-X2 (Mirus) for 30 min in a growing medium, and cultured for 12 h after activation by 488-nm laser for 10 min, using C2C12 cells stably overexpressing mCherry-Cry2 and cells not activated by 488-nm laser as controls. The cells were harvested and lysed in mild lysis buffer containing 50 mM Tris, 0.5% sodium deoxycholate, and 0.5% Triton X-100 (pH 7.4), and then digested with 0.25, 0.50, 0.75, 1.0, and 2.0 μg/ml proteinase K for 30 min at 37 °C. Digestion was terminated by the addition of 1.5 μl protease inhibitor cocktail (Target Mol) and incubation in boiling water for 10 min. The cell lysates were boiled in SDS-PAGE loading buffer, subjected to 12.5% SDS-PAGE and transferred to polyvinylidene difluoride membranes (Millipore). The membranes were blocked with 5% fat-free milk in 25 mM Tris-buffered saline buffer containing 0.047% Tween 20 (TBST). The membranes were incubated with the mouse monoclonal anti-PrP antibody 8H4 (1:5000) overnight at 4 °C followed by incubation with a homologous horseradish peroxidase-conjugated secondary antibody at a dilution of 1:5000 for 1 h at room temperature.

## Statistical analysis and reproducibility

The data shown for each experiment were based on at least three technical replicates, as indicated in individual figure legends. Data are presented as mean ± SD, and $p$-values were determined using a two-sided Student's $t$-test in Fig. 6a–e. One-way two-sided ANOVA and multiple comparisons with no adjustments were performed by SPSS 19.0 and different letters indicate significant differences at the level of $p < 0.05$ in Figs. 3b–d, 4d, e, 6h–k, 8m, n, and 9b, c, g. All experiments were further confirmed by biological repeats. The experiments in Fig. 1a–c, 2a–d, 3a, e, f, 4a–c, 5b, j, 6f, g, 7a–c, 8g, h, l, and 9a, d–f, h were repeated three times independently with similar results.

## Reporting summary

Further information on research design is available in the Nature Portfolio Reporting Summary linked to this article.

# Data availability

The RIP-seq data generated in this study have been deposited in the Gene Expression Omnibus (GEO) database under accession code GSE203419. The mouse genome (mm10, Genome Reference Consortium Mouse Build 38, GCA_000001635.2) was produced by the Mouse Genome Sequencing Consortium, and the National Center for Biotechnology Information (NCBI). The supplementary data generated in this study, including Supplementary Table 1 and Supplementary Figs. 1–3, are provided in the Supplementary Information file. Biological materials are available upon request. The source data are provided with this paper, including the statistical source data for Fig. 3b–d, Fig. 4d, e, Fig. 5f, g, Fig. 6a, b, d, e, g–k, Fig. 7d, Fig. 8m, n and Fig. 9b, d, e and the uncropped gel images for Fig. 3a, Fig. 4c, Fig. 5b, j, Fig. 6c, f, Fig. 8l, and Fig. 9a, c, h. Source data are provided with this paper.

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

## Acknowledgements

Y.Liang acknowledges fundings from the National Natural Science Foundation of China (nos. 32271326, 32071212, and 31770833). Y.Liang also acknowledges financial support from the Key Project of Basic Research, Science and Technology R&D Fund of Shenzhen (no. JCYJ20200109144418639) and the Translational Medicine and Interdisciplinary Research Joint Fund of Zhongnan Hospital of Wuhan University (no. ZNJC201934). L.X. was supported by the National Natural Science Foundation of China (nos. 81900797 and 81871036) and Guangdong Basic and Applied Basic Research Foundation (no. 2020B1515020046). Y.Zeng acknowledges financial support from the Scientific Research Project of Hubei Health Commission (no. WJ2023M079). L.-Q.W. acknowledges financial support from the National Natural Science Foundation of China (no. 32201040), China Postdoctoral Science Foundation (nos. 2021TQ0252 and 2021M700103), and the Fundamental Research Funds for the Central Universities (no. 2042022kf1047). Y.Zhou acknowledges financial support from the National Natural Science Foundation of China (no. 31922039) and the Natural Science Foundation of Hubei Province (no. 2020CFA057). We thank X. Ji (Peking University) for the gift of the pHR-mCherry-cry2 vector and Y.Liu (Wuhan University) for helpful suggestions.

## Author contributions

Y.Liang supervised the project. J.T. and Y.Liang designed the experiments. J.T., L.-Q.W., and J.C. purified the mouse PrPC and cultured the cells. J.T. and B.D. performed in vitro phase separation and droplet fusion experiments. J.T. performed western blotting, immunoblotting and immunohistochemistry analyses, RIP-seq, small RNA-seq, and OptoIDR assays. J.T., Y.Zhou, and Y.Liang performed RIP-seq and small RNA-seq analyses. Y.Zeng, Y.Liu, and Z.L. provided the skeletal muscle samples. Y.Zeng and Y.Liu performed H&E staining of frozen skeletal muscle sections. X.P. and L.X. provided transfection methods and plasmids for myoblasts. J.T. and Y.Liang wrote the manuscript. All authors proofread and approved the manuscript.

## Competing interests

The authors declare no competing interests.
