## [Peer Review File · Nature Communications]

REVIEWER COMMENTS

Reviewer #1 (Remarks to the Author):

The cellular prion protein PrP has essential functions in normal cellular physiology, including muscle cell differentiation and regeneration. In this study the authors examined muscle samples from 6 human myopathy patients and found PrP to be overexpressed. They showed that in cell culture both overexpression and knockout of PrP impaired myoblast differentiation (which is not new as acknowledged by the authors themselves in Introduction). In trying to find a mechanism for these observations, they examined autophagy in cells, and found autophagy to be inhibited by overexpression but not knockdown of PrP. However, the effect of PrP on autophagy and differentiation is correlative at best, and no evidence is provided for a causal relationship. The authors then turned to microRNA association with PrP, the rationale of which is not clear. They found subsets of miRNAs associating with endogenous or overexpressed PrP, and then focused on a couple of miRs specifically associated with overexpressed PrP (miR-214-3p and miR-204-3p). They showed that these miRs colocalized with PrP in myopathy muscle samples, and that overexpression of miR mimics or inhibitors perturbed myoblast differentiation and autophagy. In another turn of direction that was not well rationalized, the authors then examined the involvement of those miRs in PrP liquid phase separation.

Overall, this study utilized some nice technical methods, and led to some interesting observations. But the experiments seemed somewhat scattered and the experimental design was not clearly justified. The authors over-interpreted their data and drew conclusions that were not substantiated by evidence provided.

The manuscript is difficult to read. The Results section and figure legends are excessively lengthy and repetitive, making it difficult for this reviewer to extract key information in order to comprehend the results. Figure legends should be just clear description of the experiments, not repeating the data interpretation and conclusion that are already in Results.

Specific comments pertaining to each figure:

Figure 1:

1. In 1a and 1b, the control samples should have normal levels of PrP but it is not visible. What was done to the imaging to ensure fair comparison between normal and myopathy samples?
2. What led to the authors' conclusion that the myopathy muscles underwent "incomplete regeneration"? What is the evidence that these muscles were regenerating at all? Most of these samples do not seem to contain any centrally nucleated fibers (indicative of regeneration).

3. “ We have shown that PrPC overexpression does impair its function in regulating myoblast differentiation (Fig. 1a–d).” Figure 1 has no myoblast differentiation data.

Figure 2:

1. Western blotting data should not be compared across different blots to reach the conclusion about the effect of PrP overexpression on differentiation. For instance, MyoG blots appear to show constitutive expression upon WT-PrP overexpression, even though the band intensity was overall low (which could simply be exposure time). To make direct comparisons, different samples need to be run on the same blot.

2. 2f: It appears that WT PrP-expressing cells had some MHC signal on Day 3 which was gone on Day 5. What is an interpretation of this observation?

Figure 3:

1. The statement “localization of endogenous PrPC, which is mainly attached to the plasma membrane, with the autophagy marker LC3B” is confusing. LC3B should not be in the PM. The image in 3a shows a strong patch of LC3B near the PM which overlaps with PrP, but is this a normal pattern for LC3B?

2. What does “endogenous PrPc did form abundant LC3B puncta” mean? Do they mean PrPc colocalizes with LC3B puncta?

3. No experiment is performed to probe whether autophagy is RESPONSIBLE for the effect of PrP overexpression on differentiation. In fact, PrP KO inhibited differentiation but not autophagy, implying that the two processes can be uncoupled.

Figure 4:

1. 4f and 4g: assuming each orange bar is paired with a gray bar as control (IgG), the last one on each graph seems to be missing the control.

2. The overlap of miRNA pulled down between the two sets of RIP seems very small. Since PrP overexpression inhibits differentiation, could the difference in miRNA pulled down mainly due to the differentiation vs undifferentiation state?

3. The selective pulldown of specific miRNAs seems to be correlated with their abundance, i.e., the upregulated miRNAs under each condition got pulled down. This raises the question whether PrP pulldown of miRNA is really specific.

Figure 5:

1. 5c, d, e, f, g, h should be labeled as miR inhibitor rather than miR to avoid causing confusion.

2. The effect of miR inhibitor is there with either KO or PrP overexpression. Would it suggest that the miRNA actions are independent of their binding to PrP?
3. 5c, f, d, g: "ATG5-dependent autophagy" and "LC3B-dependent autophagy" are misleading; the authors were simply measuring ATG5 and LC3B levels as proxy for autophagy.
4. 5d, 5e: the conclusion "...after transfection with miR-214-3p inhibitor, excess PrPC significantly inhibited both ATG5-dependent autophagy and myoblast differentiation..." cannot be derived from this data because comparing band intensities on western blots from two different cell lines would not be reliable. For this reason, the conclusion "These results demonstrate that miR-214-3p significantly inhibits ATG5-dependent autophagy and myoblast differentiation via specific interaction with PrPC..." is not substantiated by data.

Figure 6 and Figure 7

1. It is not at all clear how LLPS came into the picture after the authors found effects of PrP-associated miRNAs on autophagy.
2. What is the FUNCTIONAL relevance of all the observations on LLPS?

Reviewer #2 (Remarks to the Author):

The authors report very interesting findings on the roles of PrP, LLPS composed of PrP and miRNAs, and certain miRNAs on myopathies associated with PrP accumulation/overexpression. They first detected much higher levels of PrP in the cytoplasm of skeletal muscles from 6 myopathy patients with dermatomyositis, neurogenic myopathy or muscular atrophy, and showed that there were numerous "dots" where PrP and miRNA-214-3p co-localize. Similar co-localization of PrP and miRNA-214-3p was also observed in "dots" in the cytoplasm of differentiating C2C12 cells overexpressing wild type mouse PrP. They went on to demonstrate the following with a number of techniques:

1. In mouse C2C12 myoblast cells, PrP knockout prevents myoblast differentiation.
2. Overexpression of wild type or mutant PrP also prevents myoblast differentiation and most of the PrP is in the cytoplasm.
3. In C2C12 cells, overexpression of wild type PrP strongly inhibits autophagy related proteins (LC3B and ATG5).
4. PrPC selectively binds to a subset of miRNAs, especially miRNA-214-3p and miRNA-204-5p in differentiating myoblasts (C2C12)
5. Inhibition of miRNA-214-3p significantly enhances both autophagy (ATG5) and myoblast differentiation in C2C12 cells (even when PrP is knocked out), and such enhancement is reduced when PrP is overexpressed.

6. In PrP-KO C2C12 cells, inhibition of miRNA-205-5p significantly promotes autophagy (LC3B-II), but had no effect on myoblast differentiation (no differentiation); in C2C12 cells overexpressing PrP, inhibition of miRNA-205-5p enhances both autophagy (LC3B-II) and myoblast differentiation.
7. PrP overexpression enhanced stability of miRNA-204-5p and to a lesser degree for miRNA-214-3p.
8. Recombinant PrP selectively recruits miRNA204-5p and miRNA-214-3p into its phase-separated condensate, which stimulates LLPS of PrP and the fusion of PrP droplets in vitro.
9. A chimeric PrP-mCh-Cry protein that can be induced by light to form liquid condensate shows that PrP LLPS can form in C2C12 cells; such LLPS recruits miRNA-214-3p, which in turn promotes PrP to form LLPS and evolve into PrP aggregates.

These data demonstrate that overexpression of wild type PrP (or mutant PrP) is toxic to myoblasts via impairing autophagy and differentiation, that miRNA-214-3p and miRNA-204-5p enhance such PrP toxicity, that PrP selectively binds to miRNA-214-3p and miRNA-204-5p and recruits them into PrP liquid droplets, and that miRNA-214-3p enhances LLPS of PrP and the fusion of PrP droplets in vitro and the formation of PrP aggregates in C2C12 cells overexpressing PrP. These results are consistent with the authors' conclusion that "PrPC located in the cytoplasm blocks muscle cell differentiation via selectively recruiting a subset of miRNAs including miR-214-3p into phase-separated condensates in living myoblasts, which in turn greatly enhances the LLPS of PrPC and the subsequent PrP aggregation in vivo and significantly inhibits ATG5-dependent autophagy", providing insights on the possible cytotoxic mechanisms of PrP overexpression in skeletal muscles. The observation of many co-localized dots of PrP and miRNA-214-3p in the cytoplasm of a group of myopathy patients provide clinical relevance. The design is good and the data of high quality overall. The findings are very significant and meaningful.

However, there are some minor to modest caveats.

First, the PrP aggregates in C2C12 cells were demonstrated by the amount of PrP in sarkosyl-insoluble pellets of cell lysate from cells overexpressing the chimeric PrP-mCh-Cry2 (Fig. 7k). It would be more authentic if the same can be proven with cells expressing wild type mouse PrP. In addition, aggregated PrP is more resistant (than normal PrPC) to digestion with modest amount of proteases (such as proteinase K, 5-100ug/ml), so it would be helpful to show some level of protease resistance.

Second, the age-matched controls (age 32, 70, 63, 64) do not match well with the myopathy patients (age 43-52). Moreover, only one control (age 70) is a healthy control and two of the controls are lipid storage myopathy patients. So it is not myopathy patients vs. non-myopathy patients, rather it is the myopathy patients with excessive PrP in skeletal muscles vs others. It would help to make this point explicit.

Third, the figure legends are too long and contain some redundancy.

Fourth, in Fig.5 c-k, the data labeled miR-214-3p or miRNA-204-5p actually meant “inhibitor” of miR-214-3p or miRNA-204-5p. It would be better to label as such to avoid confusion.

Fifth, many pictures are too small and of low resolutions that makes it hard to discern key details and structures (such as membrane vs cytoplasmic PrP distribution in Fig. 2c).

Sixth, some phrases/words are less than optimal or incorrect in the context. It would help to have a qualified scientific proofreader to edit the text. Below are some examples:

- Lines 141-142, “After 5 days of differentiation, however, excess WT PrPC is completely located in the cytoplasm” (Fig. 2e) is an overstatement. It is nearly impossible to be sure that PrP is not on the cell membrane when there are plenty of PrP in the cytoplasm and examined by confocal microscopy alone.
- Line 145, “maintenance of PrPC homeostasis is essential for myoblast differentiation” is an overstatement. It is well known that PrP-KO mice are basically normal, including their skeletal muscles, under normal conditions. So “essential” should be replaced with something like “important”.
- Line 47, “conservative” should be “conserved”.
- Line 119, “until confluence was reached 90%” should be “until confluence reached 90%”.
- Line 123, “more than 90% sequence identity” should be “more than 90% sequence homology”
- Line 98, “PrPC (red) was overexpressed” is inaccurate since no study was done to demonstrate overexpression of the PrP gene in this study. It is more accurate to state “a higher level of PrP was accumulated”.

Reviewer #1

Remarks to the Author:

Summary:

The cellular prion protein PrP has essential functions in normal cellular physiology, including muscle cell differentiation and regeneration. In this study the authors examined muscle samples from 6 human myopathy patients and found PrP to be overexpressed. They showed that in cell culture both overexpression and knockout of PrP impaired myoblast differentiation (which is not new as acknowledged by the authors themselves in Introduction). In trying to find a mechanism for these observations, they examined autophagy in cells, and found autophagy to be inhibited by overexpression but not knockdown of PrP. However, the effect of PrP on autophagy and differentiation is correlative at best, and no evidence is provided for a causal relationship. The authors then turned to microRNA association with PrP, the rationale of which is not clear. They found subsets of miRNAs associating with endogenous or overexpressed PrP, and then focused on a couple of miRs specifically associated with overexpressed PrP (miR-214-3p and miR-204-3p). They showed that these miRs colocalized with PrP in myopathy muscle samples, and that overexpression of miR mimics or inhibitors perturbed myoblast differentiation and autophagy. In another turn of direction that was not well rationalized, the authors then examined the involvement of those miRs in PrP liquid phase separation.

Overall, this study utilized some nice technical methods, and led to some interesting observations. But the experiments seemed somewhat scattered and the experimental design was not clearly justified. The authors over-interpreted their data and drew conclusions that were not substantiated by evidence provided.

The manuscript is difficult to read. The Results section and figure legends are excessively lengthy and repetitive, making it difficult for this reviewer to extract key information in order to comprehend the results. Figure legends should be just clear description of the experiments, not repeating the data interpretation and conclusion that are already in Results.

We sincerely thank the reviewer for recognizing the significance of our work. The reviewer's suggestion is very valuable for us to improve our manuscript. Right now, we have revised and added the following sentences into the revised manuscript, as followed the advice of reviewer #1. *Because the cellular prion protein PrP^C has important functions in normal cellular physiology, including*

muscle cell differentiation and regeneration^{15,16,18}, ... (Lines 563-564). We showed that in cell culture both overexpression and knockout of PrP^C impaired myoblast differentiation. To decode the mechanism, we examined autophagy in cells and found autophagy to be inhibited and enhanced by overexpression and knockout of PrP^C, respectively (Lines 572-576). In this study, we found subsets of miRNAs associating with endogenous or overexpressed PrP^C, and then focused on a couple of miRNAs (miR-214-3p and miR-204-5p) specifically associated with overexpressed PrP^C. We showed that a higher level of PrP^C colocalized with these miRNAs in myopathy muscle samples, and that overexpressed PrP^C colocalized with these miRNAs in skeletal muscle cells (Lines 585-590).

Thanks for the comments. The reviewer is correct. We have now revised the manuscript by addressing all methodological concerns and the request for more evidence to substantiate our conclusions, as followed the advice of review #1.

We sincerely thank the reviewer for these important suggestions to improve our manuscript. We totally agree that the Results section and figure legends of the previous manuscript are excessively lengthy and repetitive. We apologize for such extraordinary repetition in the previous manuscript. According to the advice of the reviewers #1 and 2, we have now made the Results section and figure legends more concise. We hope that it would be much easier for the reviewers to extract key information in order to comprehend the results. For instance, “We first took confocal images of frozen skeletal muscle sections from six myopathy patients and four age-matched controls (Fig. 1a,b). These myopathy patients include two patients with dermatomyositis, two patients with neurogenic myopathy, and two patients with muscular dystrophy, and those controls include one healthy individual, two patients with lipid storage myopathy, and one patient with glycogen storage disease (Table 1 and Fig. 1a–d)” **in the previous version have been shortened into** “*We first took confocal images of frozen skeletal muscle sections from six myopathy patients with dermatomyositis, neurogenic myopathy or muscular atrophy, one healthy control, two controls with lipid storage myopathy, and one control with glycogen storage disease (Table 1 and Figs. 1a and 2a)*” **in the revised manuscript (Lines 88-92)**. “The co-localization of endogenous PrP^C, which is mainly attached to the plasma membrane, with the autophagy marker LC3B (yellow dots in the merged image, Fig. 3a) was observed in C2C12 cells (control) upon

differentiation for 3 days. The co-localization of endogenous PrP with LC3B (yellow dots in the merged image, Fig. 3b) was also observed in the elongated myotubes differentiated from C2C12 myoblasts after 5 days of differentiation. After 3 days of differentiation, endogenous PrP^C did form abundant LC3B-positive puncta (green dots, Fig. 3a) in C2C12 cells, and after 5 days of differentiation, endogenous PrP^C produced more abundant LC3B-positive puncta (green dots, Fig. 3b) in C2C12 cells, as detected by immunofluorescence using anti-LC3B antibody (green) and the anti-PrP antibody 8H4 (red). Importantly, excess PrP^C strongly inhibited skeletal muscle cell autophagy and blocked myoblast differentiation, producing much fewer LC3B-positive puncta (green dots, Fig. 3a,b) when C2C12 cells stably expressing WT PrP^C were incubated with the differentiation medium for 3 and 5 days. After 3 days of differentiation, excess PrP^C is partly located on the plasma membrane and partly located in the cytoplasm (Fig. 3a); after 5 days of differentiation, however, excess PrP^C is completely located in the cytoplasm (Fig. 3b); under both conditions, the co-localization of PrP^C and LC3B (the merged images, Fig. 3a,b) were not observed. Moreover, PrP^C deficiency completely blocked muscle cell differentiation but did not inhibit skeletal muscle cell autophagy (green dots, Fig. 3a,b) when C2C12 myoblasts KO for PrP^C were incubated with this medium for 3 and 5 days” **in the previous version have been shortened into** *“Endogenous PrP^C (red) was mainly attached to the plasma membrane and did not colocalize with the autophagy marker LC3B (green) in C2C12 cells (control) upon differentiation for 3 days (Fig. 4a). However, the colocalization of endogenous PrP^C and LC3B puncta (green dots, Fig. 4b) was observed in the elongated myotubes differentiated from C2C12 myoblasts after 5 days of differentiation (yellow dots in the merged image, Fig. 4b). Importantly, excess PrP^C strongly inhibited skeletal muscle cell autophagy and blocked myoblast differentiation, producing much fewer LC3B-positive puncta (green dots, Fig. 4a,b) when C2C12 cells stably expressing WT PrP^C were incubated with the differentiation medium for 3 and 5 days. After 3 days of differentiation, excess PrP^C was partly located on the plasma membrane and partly located in the cytoplasm (Fig. 4a); after 5 days of differentiation, however, excess PrP^C was primarily located in the cytoplasm (Fig. 4b); under both conditions, the colocalization of PrP^C and LC3B (the merged images, Fig. 4a,b) was not observed. Moreover, PrP^C deficiency resulted in increased autophagic activity in skeletal muscle cells (green dots, Fig. 4a,b) and blocked muscle cell differentiation when C2C12 myoblasts KO*

for PrP^C were incubated with this medium for 3 and 5 days” in the revised manuscript (Lines 161-176). “To gain a quantitative understanding of how miR-214-3p regulates skeletal muscle cell autophagy, we performed Western blot analysis for ATG5 and the myogenic differentiation marker MyHC after transfection with NC or miR-214-3p inhibitor at 10 μM in the above cell lines (Fig. 5c). Upon differentiation for 4 days, the relative amounts of ATG5 (Fig. 5d) and MyHC (Fig. 5e) in the cell lysates from C2C12 myoblasts stably expressing WT PrP^C or KO for PrP^C transfected with 10 μM miR-214-3p inhibitor were significantly higher than those in the cell lysates from the above cell lines transfected with NC. Thus, compared with transfection of NC in C2C12 myoblasts KO for PrP^C and C2C12 myoblasts stably expressing WT PrP^C, transfection of 10 μM miR-214-3p inhibitor significantly promoted ATG5-dependent autophagy and myoblast differentiation in the above cell lines (Fig. 5c–e), suggesting that miR-214-3p significantly inhibits both ATG5-dependent autophagy and myoblast differentiation via specific interaction with PrP^C. Importantly, after transfection with miR-214-3p inhibitor, excess PrP^C significantly inhibited both ATG5-dependent autophagy and myoblast differentiation in differentiating C2C12 cells stably expressing WT PrP^C, compared to those in differentiating C2C12 myoblasts KO for PrP^C (Fig. 5d,e). To gain a quantitative understanding of how miR-204-5p regulates skeletal muscle cell autophagy, we performed Western blot analysis for LC3B and MyHC after transfection with NC or miR-204-5p inhibitor at 10 μM in the above cell lines (Fig. 5f). Upon differentiation for 4 days, transfection of 10 μM miR-204-5p inhibitor significantly promoted LC3B-dependent autophagy in C2C12 myoblasts KO for PrP^C and C2C12 myoblasts stably expressing WT PrP^C, compared to transfection of NC in the above cell lines (Fig. 5g). Importantly, PrP^C deficiency completely blocked muscle cell differentiation after transfection with NC or miR-204-5p inhibitor at 10 μM in C2C12 myoblasts KO for PrP^C upon differentiation for 4 days (Fig. 5h). Compared with transfection of NC in C2C12 myoblasts stably expressing WT PrP^C, however, transfection of miR-204-5p inhibitor significantly promoted myoblast differentiation in the above cell line (Fig. 5h), suggesting that miR-204-5p, another PrP^C-bound miRNA, significantly inhibits myoblast differentiation” in the previous version have been shortened into “To gain a quantitative understanding of how miR-214-3p and miR-204-5p regulate skeletal muscle cell autophagy, we measured ATG5 and LC3B levels as proxies for

autophagy and performed western blot analysis for ATG5, LC3B, and the myogenic differentiation marker MyHC after transfection with NC, miR-214-3p inhibitor or miR-204-5p inhibitor at 10 μ M in the above cell lines (Fig. 6f,g). Upon differentiation for 4 days, the relative amounts of ATG5 (Fig. 6h) and MyHC (Fig. 6i) in the cell lysates from C2C12 myoblasts stably expressing WT PrP^C or KO for PrP^C transfected with 10 μ M miR-214-3p inhibitor were significantly higher than those in the cell lysates from the above cell lines transfected with NC. Thus, compared with transfection of NC in C2C12 myoblasts KO for PrP^C and C2C12 myoblasts stably expressing WT PrP^C, transfection of 10 μ M miR-214-3p inhibitor significantly promoted ATG5-related and myoblast differentiation in the above cell lines (Fig. 6f,h,i), suggesting that miR-214-3p significantly inhibits autophagy and differentiation of skeletal muscle cells via specific interaction with PrP^C. Upon differentiation for 4 days, transfection of 10 μ M miR-204-5p inhibitor significantly promoted LC3B-related autophagy in C2C12 myoblasts KO for PrP^C and C2C12 myoblasts stably expressing WT PrP^C compared to transfection of NC in the above cell lines (Fig. 6j). Importantly, PrP^C deficiency completely blocked muscle cell differentiation after transfection with NC or miR-204-5p inhibitor at 10 μ M in C2C12 myoblasts KO for PrP^C upon differentiation for 4 days (Fig. 6k). Compared with transfection of NC in C2C12 myoblasts stably expressing WT PrP^C, however, transfection of miR-204-5p inhibitor significantly promoted myoblast differentiation in the above cell line (Fig. 6k), suggesting that miR-204-5p, another PrP^C-bound miRNA, significantly inhibits myoblast differentiation” in the revised manuscript (Lines 319-342). We have now deleted the data interpretation and conclusion that are already in Results section from figure legends as follows.

Fig. 1 | Accumulation of PrP^C and the colocalization of PrP^C and miR-214-3p were clearly observed in the skeletal muscle of six myopathy patients. a, Confocal images of frozen skeletal muscle sections from six myopathy patients. These myopathy patients included two patients with dermatomyositis (DM) (Cases 1 and 2), two with neurogenic myopathy (NM) (Cases 3 and 4), and two with muscular dystrophy (MD) (Cases 5 and 6). Shown are nuclei stained with DAPI (blue); signals were detected with the anti-PrP antibody 8H4 (red), and miR-214-3p was detected by FISH (green) using an FAM-labeled miR-214-3p probe (FAM-anti-miR). The enlarged regions (right) show 4-fold enlarged images from the merged images.

Arrows indicate colocalization of PrP^C and miR-214-3p in granules. Scale bars, 75 μ m. **b**, H&E staining of the frozen sections showed that the skeletal muscle of the six myopathy patients was characterized by incomplete muscle regeneration. Dashed loops indicate centrally nucleated fibers. Scale bar, 400 μ m. **c**, Immunohistochemical analysis of NCAM expression in the frozen skeletal muscle sections showed a positive signal (brown) in the muscle bundles. Scale bar, 100 μ m (**pages 40-41**).

Fig. 2 | Accumulation of PrP^C and the colocalization of PrP^C and miR-214-3p
PrP^C were not observed in the skeletal muscle of one healthy control (Control 1), two controls with lipid storage myopathy (LSM) (Controls 2 and 3), and one control with glycogen storage disease (GSD) (Control 4). **a**, Confocal images of frozen skeletal muscle sections from these four controls. We have replaced panel a with a correct version of these four controls, in which the control samples do have normal levels of PrP^C (red). **b,c**, H&E staining and immunohistochemical staining of the frozen sections showed that muscle regeneration was not observed in the skeletal muscle of these four controls. The experimental conditions are the same as those in Fig. 1a–c. **d**, Colocalization of PrP^C and miR-214-3p in C2C12 mouse myoblasts (control) and C2C12 myoblasts stably expressing WT PrP^C upon differentiation for 4 days. Shown are nuclei stained with DAPI (blue); signals were detected with the anti-PrP antibody 8H4 (red), and miR-214-3p was detected by FISH (green). Scale bar, 10 μ m (**pages 42-43**).

Fig. 3 | Maintenance of PrP^C homeostasis is important for myoblast differentiation. **a**, Western blot for PrP^C and the myogenic differentiation markers MyHC and MyoG during C2C12 mouse myoblasts (control), C2C12 myoblasts stably expressing full-length wild-type mouse PrP^C (WT PrP^C), C2C12 myoblasts stably expressing F198S PrP^C, and C2C12 myoblasts KO for PrP^C cultured until their confluence reached 90% and then incubated with differentiation medium for 0, 3, and 5 days, respectively. β -actin served as the protein loading control. **b–d**, The relative amount of PrP^C (**b**), MyHC (**c**) or MyoG (**d**) in the above cell lines (solid black circles shown in scatter plots) was expressed as the mean \pm S.D. (with error bars) of values obtained in three independent experiments. One-way ANOVA and multiple comparisons were performed by SPSS 19.0 and different letters indicate significant differences at the level of $p < 0.05$. **e,f**, Immunofluorescence imaging of the above four cell lines incubated with differentiation medium for 3 and 5 days, respectively,

using antibody against PrP^C (red) (e) or MyHC (red) (f) and staining with DAPI (blue). Scale bars, 50 (e) and 75 (f) μ m, respectively. We have replaced the second row in panel f with a correct version of WT PrP^C, in which WT PrP^C-expressing cells did not have any MyHC signal at day 3 (pages 44-45).

Fig. 4 | Excess PrP^C strongly inhibits skeletal muscle cell autophagy and blocks myoblast differentiation. a,b, Immunofluorescence imaging of C2C12 mouse myoblasts (control), C2C12 myoblasts stably overexpressing WT PrP^C, and C2C12 myoblasts KO for PrP^C cultured until their confluence reached 90% and then incubated with differentiation medium for 3 (a) and 5 (b) days, respectively, using antibodies against PrP^C (red) and LC3B (green) and staining with DAPI (blue). The enlarged regions in the lower left corner (a) or the lower right corner (b) of the merged images show 4-fold enlarged images from the same images. Scale bars, 7.5 (a) and 50 (b) μ m, respectively. We have replaced the first row in panel a with a correct version of control, in which LC3B did not overlap with PrP^C. c, Western blot for PrP^C and the autophagy markers ATG5 and LC3B during the above three cell lines incubated with differentiation medium for 3 and 5 days, respectively. β -actin served as the protein loading control. d,e, The relative amount of ATG5 (d) or LC3B-II (e) in the above cell lines (solid black circles shown in scatter plots) was expressed as the mean \pm S.D. (with error bars) of values obtained in three independent experiments. One-way ANOVA and multiple comparisons were performed by SPSS 19.0 and different letters indicate significant differences at the level of $p < 0.05$ (pages 46-47).

Fig. 5 | PrP^C selectively binds to a subset of miRNAs during myoblast differentiation. a, Profile of total miRNAs and PrP^C IP-enriched miRNAs at day 4 of differentiation in C2C12 myoblasts stably overexpressing WT PrP^C (WT PrP^C) and C2C12 myoblasts (control). Heat map shows the expression levels of miRNAs and fold-enrichment compared to Inputs in the above two cell lines. b, Western blotting analysis of PrP^C immunoprecipitation in C2C12 myoblasts at day 4 of differentiation by probing PrP^C and GAPDH. c, Volcano plot of 31 PrP^C-bound miRNAs identified by small RNA-seq in differentiating C2C12 cells (control). d, Volcano plot of 51 PrP^C-bound miRNAs identified by small RNA-seq in differentiating C2C12 cells stably expressing WT PrP^C. Representative PrP^C RIP-enriched miRNAs are highlighted in orange. e, Venn diagram of miRNAs enriched by PrP^C RIP in C2C12 cells stably expressing WT PrP^C versus miRNAs enriched by PrP^C RIP in C2C12

myoblasts (control). Group I ($n = 21$, Control RIP only), $n = 10$ (overlap), and group II ($n = 41$, WT PrP^C RIP only). **f,g**, RT-qPCR validation of depleted versus enriched miRNAs in differentiating C2C12 myoblasts (control) (**f**) and those stably expressing WT PrP^C (**g**) normalized to total input. An axis break was introduced in the ordinate so that each orange bar is clearly paired with a gray bar as control (IgG). U6 snRNA served as a negative control. Data are presented as means \pm S.D. ($n = 3$ biologically independent measurements). **h,i**, The top six biological processes (BP) of Gene Ontology (GO) enrichment in the control cells (**h**) and the top eight BPs of GO enrichment in C2C12 cells stably overexpressing WT PrP^C (**i**). **j**, Pulldown of endogenous PrP^C and excess PrP^C with WT and mutant biotin-labeled miR-214-3p or biotin-labeled miR-204-5p in the above two cell lines at day 4 of differentiation (**pages 48-49**).

Fig. 6 | PrP^C increases the stability of mature miR-214-3p to significantly inhibit autophagy and myoblast differentiation. **a,b**, Dual luciferase reporter assays, in which miR-214-3p (**a**) or miR-204-5p (**b**) represses the wild-type 3' end of untranslated region (WT 3'UTR) of the gene *ATG5* (**a**) or *LC3B* (**b**) associated with luciferase. Each pair of reporters was assayed in response to specific miRNA mimics at day 4 of differentiation in C2C12 myoblasts stably overexpressing WT PrP^C or KO for PrP^C. The relative luciferase activity of the reporters after transfection with NC (gray), miR-214-3p mimic (orange) or miR-204-5p mimic (blue) in the above cell lines (solid black circles shown in scatter plots) was expressed as the mean \pm S.D. (with error bars) of values obtained in three independent experiments. **c-e**, The relative stability of miR-214-3p (**c**), miR-204-5p (**d**), and nonspecific miR-183-5p (**e**) in C2C12 myoblasts stably overexpressing WT PrP^C (red) or KO for PrP^C (blue) incubated with differentiation medium for 4 days and then treated with 20 mg/ml α -amanitin was determined as the relative enrichment (to U6 snRNA) of miRNAs by RT-qPCR and expressed as the mean \pm S.D. (with error bars) of values obtained in three independent experiments. **f,g**, Western blot for the autophagy markers ATG5 (**f**) and LC3B (**g**) and the myogenic differentiation marker MyHC after transfection with NC, miR-214-3p inhibitor (**f**) or miR-204-5p inhibitor (**g**) at 10 μ M in the above cell lines upon differentiation for 4 days. β -actin served as the protein loading control. **h-k**, The relative amount of ATG5 (**h**), LC3B-II (**j**) or MyHC (**i,k**) in the above cell lines (solid black circles shown in scatter plots) was expressed as the mean \pm S.D.

(with error bars) of values obtained in three independent experiments. Transfection of NC, miR-214-3p inhibitor (**h,i**) or miR-204-5p inhibitor (**j,k**) in C2C12 myoblasts KO for PrP^C (gray) and C2C12 myoblasts stably expressing WT PrP^C (orange or blue). **f–k**, Labeled as miR-214-3p inhibitor or miR-204-5p inhibitor to avoid confusion. **a–e**, Statistical analyses were performed using a two-tailed unpaired *t* test. Values of *p* < 0.05 indicate statistically significant differences. The following notation is used throughout: **p* < 0.05; ***p* < 0.01; ****p* < 0.001; and *****p* < 0.0001 relative to controls. n.s., no significance. **h–k**, One-way ANOVA and multiple comparisons were performed by SPSS 19.0 and different letters indicate significant differences at the level of *p* < 0.05 (**pages 50-51**).

Fig. 7 | PrP^C selectively recruits a subset of miRNAs into its phase-separated condensate, which in turn greatly enhances in vitro LLPS of PrP^C. **a**, Regulation of PrP^C LLPS by three miRNAs. Fluorescence images of 50 μM recombinant wild-type mouse PrP^C (WT PrP^C) labeled by TAMRA (red) and incubated with 1 × PBS (pH 7.4) containing 0, 1.25, 2.5, 5, 10 or 20 μM miRNA on ice for 5 min. Scale bar, 7.5 nm. **b**, PrP^C selectively recruits miR-214-3p and miR-204-5p into its phase-separated condensate. Fluorescence images of in vitro phase-separated droplets (red; Merge: yellow) of 50 μM TAMRA-labeled WT PrP^C incubated with 1 × PBS containing 10 μM FAM-labeled miRNA (green) on ice for 5 min. Scale bar, 2.5 nm. **c**, FRAP analysis on the selected liquid droplets of 50 μM TAMRA-labeled WT PrP^C before (prebleach), during (0 s), and after photobleaching (1, 3, 6 and 30 s, respectively). The internal photobleaching is marked by a black square. Scale bar, 2.5 nm. **d**, Normalized kinetics of fluorescence recovery data of WT PrP^C (blue circle), WT PrP^C + miR-214-3p (red square), WT PrP^C + miR-204-5p (magenta square), and WT PrP^C + miR-183-5p (olive circle) obtained from FRAP intensity. The normalized fluorescence intensity is expressed as the mean ± S.D. (with error bars) of values obtained in three independent experiments. The solid lines show the best single exponential fit for the fluorescence intensity-time curves (**pages 52-53**).

Fig. 8 | PrP^C recruits miR-214-3p into its phase-separated condensate in living skeletal muscle cells, which results in the inhibition of autophagy. **a**, Schematic of the optoIDR assay, depicting recombinant protein with an IDR (blue), mCherry (red), and Cry2 (orange) expressed in cells exposed to 488-nm excitation laser. **b,c**, Time-lapse images of living C2C12 cells expressing mCherry-Cry2-WT PrP^C construct

containing PrP₁₋₃₇ IDR (residues 1–37) (blue) linked to mCherry (red) and Cry2 (orange) and then linked to PrP₃₈₋₂₃₀ IDR (residues 38–230) (blue) (c), and mCherry-Cry2 fusion alone was used as a control (b). d,e, Time-lapse images (0–100 s) of a C2C12 cell expressing the mCherry-Cry2-WT PrP^C construct. A droplet fusion event occurs in the region highlighted by the orange box (d). The droplet fusion event highlighted by using orange arrows (e). b–e, Cells were subjected to laser excitation every 2 s for the indicated time. Scale bars, 7.5 μm. f, Representative live-cell images of PrP^C-miRNA speck formation when 10 μM FAM-labeled miRNA (green) was transfected into C2C12 cells stably expressing mCherry (red) or WT PrP^C-mCherry (red), among ≥ 10 cells. Scale bars, 7.5 μm. g,h, FRAP analysis on the selected liquid droplets of PrP^C (red) in living C2C12 cells expressing mCherry-Cry2-WT PrP^C transfected without (g) or with (h) miRNA before (prebleach), during (0 s), and after photobleaching (60 and 125 s, respectively). The dashed white circle highlights the punctum undergoing targeted bleaching. Scale bars, 7.5 μm. i, Quantification of FRAP data for WT PrP^C puncta without or with miRNA (*n* = 3), as in g,h. Time 0 indicates the start of recovery after photobleaching. j, Normalized kinetics of fluorescence recovery data of WT PrP^C puncta (blue circle) and WT PrP^C puncta + miR-214-3p (red square). The normalized fluorescence intensity is expressed as the mean ± S.D. (with error bars) of values obtained in three independent experiments. The solid lines show the best single exponential fit for the fluorescence intensity-time curves. k, Time-lapse images (0–300 s) of living C2C12 cells expressing mCherry-Cry2-WT PrP^C treated with 2.5% 1,6-hexanediol. Scale bar, 5 μm. l, Western blot for the autophagy markers ATG5 and LC3B in C2C12 cells expressing mCherry-Cry2-WT PrP^C transfected without (-) or with (+) 10 μM miRNA and then treated with (+) or without (-) 1,6-hexanediol. β-actin served as the protein loading control. m,n, The relative amount of ATG5 (m) and LC3B-II (n) in the above cell lines (solid black circles shown in scatter plots) was expressed as the mean ± S.D. (with error bars) of values obtained in three independent experiments. One-way ANOVA and multiple comparisons were performed by SPSS 19.0 and different letters indicate significant differences at the level of *p* < 0.05 (pages 54-55).

Fig. 9 | PrP^C recruits miR-214-3p into its phase-separated condensate in living skeletal muscle cells, which in turn promotes pathological aggregation of PrP. a, Western blot for PrP in the sarkosyl-insoluble pellets and the corresponding cell

lysates from C2C12 cells stably overexpressing mCh-Cry2-WT PrP^C transfected without (-) or with (+) 10 μ M miR-214-3p and cultured for 12 h after activation by 488-nm laser for 10 min. β -actin served as the protein loading control. **b**, The normalized amount of insoluble PrP aggregates in the above cell lines (solid black circles shown in scatter plots) was expressed as the mean \pm S.D. (with error bars) of values obtained in three independent experiments. **c**, The cell viability (%) (solid black circles shown in scatter plots) was measured by MTT reduction assay and expressed as the mean \pm S.D. (with error bars) of values obtained in five independent experiments. **d,e**, Immunogold electron microscopy of PrP fibrils purified from the same C2C12 myoblasts stably expressing mCh-Cry2-WT PrP^C transfected without (**d**) or with (**e**) 10 μ M miR-214-3p as in (**a**), and labeled by gold particles conjugated with anti-PrP antibody. Scale bars, 100 nm. **f**, Western blot for PrP in the sarkosyl-insoluble pellets and the corresponding cell lysates from C2C12 myoblasts stably overexpressing WT PrP^C incubated with differentiation medium for 2 days, transfected without or with 10 or 20 μ M miR-214-3p and cultured for 48 h. β -actin served as the protein loading control. **g**, the same as in (**b**). **b,c,g**, One-way ANOVA and multiple comparisons were performed by SPSS 19.0 and different letters indicate significant differences at the level of $p < 0.05$. **h**, Western blot for PrP in the cell lysates from the same C2C12 cells stably overexpressing mCh-Cry2-WT PrP^C transfected without or with 10 μ M miR-214-3p as in (**a**), and digested with 0.25, 0.50, 0.75, 1.0, and 2.0 μ g/ml proteinase K (**pages 56-57**).

Fig. 10 | A hypothetical model shows how excess PrP^C inhibits muscle cell differentiation via miRNA-enhanced LLPS of PrP^C implicated in myopathy.

Under pathological conditions (right), PrP^C (orange) is overexpressed in muscle satellite cells (orange) and myoblasts (pale blue). Then, PrP^C selectively recruits miR-214-3p and miR-204-5p into phase-separated condensates (gold balls) in living myoblasts to block autophagy and preserve myoblasts undifferentiated. This recruitment in turn greatly enhances PrP^C liquid phase condensation and the subsequent aggregation to produce PrP fibrils (gold bars), resulting in skeletal muscle cell death and muscle bundle formation in myopathy patients. Under physiological conditions (left), PrP^C (orange) is expressed in myoblasts (pale blue) and enhances muscle cell differentiation by selectively interacting with miR-486a-5p and miR-

181b-5 to significantly promote autophagy, and finally, muscle bundles are assembled to create the whole muscle (page 58).

Specific comments pertaining to each figure:

Comment #1 • Figure 1:

1. In 1a and 1b, the control samples should have normal levels of PrP but it is not visible. What was done to the imaging to ensure fair comparison between normal and myopathy samples?

REPLY: We apologize for this mistake. We have now replaced panel a with a correct version of these four controls (Fig. 2, page 42), as followed reviewer #1's nice suggestions. We have replaced panel a with a correct version of these 4 controls, in which the control samples do have normal levels of PrP^C (red) (Lines 858-859, Legend of Fig. 2). We have revised and added the following sentence into the Results section of the revision. *In sharp contrast, accumulation of PrP^C was not observed in the skeletal muscle of these four controls and the control samples only had normal levels of PrP^C (red) (Fig. 2a and Fig. S1) (Lines 95-97). To ensure fair comparison between normal and myopathy samples, all confocal images in Figs. 1 and 2 as well as Fig. S1 were acquired using the same laser intensity and gain values and the same confocal microscope (Leica TCS SP8 laser scanning confocal microscope).*

Fig. 2 | Accumulation of PrP^C and the colocalization of PrP^C and miR-214-3p
PrP^C were not observed in the skeletal muscle of one healthy control (Control 1),
two controls with lipid storage myopathy (LSM) (Controls 2 and 3), and one
control with glycogen storage disease (GSD) (Control 4). a, Confocal images of
frozen skeletal muscle sections from these four controls. We have replaced panel a
with a correct version of these four controls, in which the control samples do have
normal levels of PrP^C (red). Scale bar, 10 μm.

Fig. S1 | Accumulation of PrP^C was not observed in the skeletal muscle of one healthy control (Control 1), two controls with lipid storage myopathy (LSM) (Controls 2 and 3), and one control with glycogen storage disease (GSD) (Control 4). The confocal images of frozen skeletal muscle sections from these four controls are observed from three views and show that the control samples have normal levels of PrP^C (red). Scale bar, 10 μ m.

2. What led to the authors' conclusion that the myopathy muscles underwent "incomplete regeneration"? What is the evidence that these muscles were regenerating at all? Most of these samples do not seem to contain any centrally nucleated fibers (indicative of regeneration).

REPLY: Thank the reviewer for this great point! Indeed, we cannot be sure that the myopathy muscles underwent incomplete regeneration based solely on H&E staining (previous Figure 1c). To address the questions from reviewer #1, we have now provided evidence that skeletal muscles of the six patients were regenerating (Fig. 1b,c, page 40) but those of these four controls did not have morphological and molecular features of regeneration (Fig. 2b,c, page 42). Figs. 1b and 2b are enlarged versions of the previous Fig. 1c,d. We have revised and

added the following sentences into the Results section of the revision. *The skeletal muscles were then analyzed by H&E staining and immunohistochemistry using an anti-NCAM antibody (brown) (Figs. 1b,c and 2b,c). H&E staining and immunohistochemical staining of the frozen skeletal muscle sections showed that the skeletal muscle of the six myopathy patients had morphological features of regeneration, such as internalized nuclei (in dashed loops, Fig. 1b), and expressed high levels of neural cell adhesion molecule (NCAM) (brown) (Fig. 1c), a marker of muscle regeneration⁵⁷. In the skeletal muscles of the six patients, many centrally nucleated fibers were observed in each sample (Fig. 1b). In sharp contrast, the skeletal muscle of these four controls did not have any morphological features of regeneration (Fig. 2b) and showed lack of expression of NCAM (Fig. 2c). Thus, the skeletal muscle of the six myopathy patients was characterized by incomplete muscle regeneration (cases 2 and 6 with obvious muscular atrophy) but muscle regeneration was not observed in the skeletal muscle of these four controls (Figs. 1b,c and 2b,c) (Lines 97-110). A subsection titled “Immunohistochemistry” has been added into the Methods section of the revision (pages 61-62). Accordingly, one related publication (Ref. 57) has been added into the revision.*

Fig. 1 | Accumulation of PrP^C and the colocalization of PrP^C and miR-214-3p were clearly observed in the skeletal muscle of six myopathy patients. b, H&E staining of the frozen sections showed that the skeletal muscle of the six myopathy patients was characterized by incomplete muscle regeneration. Dashed loops indicate centrally nucleated fibers. Scale bar, 400 μ m. **c,** Immunohistochemical analysis of NCAM expression in the frozen skeletal muscle sections showed a positive signal (brown) in the muscle bundles. Scale bar, 100 μ m.

Fig. 2 | Accumulation of PrP^C and the colocalization of PrP^C and miR-214-3p PrP^C were not observed in the skeletal muscle of one healthy control (Control 1), two controls with lipid storage myopathy (LSM) (Controls 2 and 3), and one control with glycogen storage disease (GSD) (Control 4). b,c, H&E staining and immunohistochemical staining of the frozen sections showed that muscle regeneration was not observed in the skeletal muscle of these four controls. The experimental conditions are the same as those in Fig. 1a–c.

3. “We have shown that PrP^C overexpression does impair its function in regulating myoblast differentiation (Fig. 1a–d).” Figure 1 has no myoblast differentiation data.

REPLY: We apologize for this confusion. Indeed, Figure 1 has no myoblast differentiation data. According to the advice of the reviewer, we have now deleted the incorrect statement “We have shown that PrP^C overexpression does impair its function in regulating myoblast differentiation (Fig. 1a–d)” from the manuscript (Line 116).

Comment #2 • Figure 2:

1. Western blotting data should not be compared across different blots to reach the conclusion about the effect of PrP overexpression on differentiation. For instance, MyoG blots appear to show constitutive expression upon WT-PrP overexpression, even though the band intensity was overall low (which could simply be exposure time). To make direct comparisons, different samples need to be run on the same blot.

REPLY: We sincerely thank the reviewer for his (her) expert suggestion that western blotting data should not be compared across different blots to reach the conclusion about the effect of PrP overexpression on differentiation (previous Figure 2a–d)!! According to the advice of reviewer #1, we have performed

western blotting experiments on the same blot and have made direct comparisons of different samples run on the same blot (Fig. 3a, page 44). We have revised and added the following sentence into the Results section of the revision. *To make direct comparisons, different samples were run on the same blot, and the cell lysates were probed with the anti-PrP monoclonal antibody 8H4, anti-MyHC antibody, anti-MyoG antibody, and anti- β -actin antibody (Fig. 3a) (Lines 129-132).*

Fig. 3 | Maintenance of PrP^C homeostasis is important for myoblast differentiation. a, Western blot for PrP^C and the myogenic differentiation markers MyHC and MyoG during C2C12 mouse myoblasts (control), C2C12 myoblasts stably expressing full-length wild-type mouse PrP^C (WT PrP^C), C2C12 myoblasts stably expressing F198S PrP^C, and C2C12 myoblasts KO for PrP^C cultured until their confluence reached 90% and then incubated with differentiation medium for 0, 3, and 5 days, respectively. β -actin served as the protein loading control.

2. 2f: It appears that WT PrP-expressing cells had some MHC signal on Day 3 which was gone on Day 5. What is an interpretation of this observation?

REPLY: We apologize for this mistake. We have now replaced the second row in panel f with a correct version of WT PrP^C (Fig. 3f, page 44), as followed reviewer #1's nice suggestion. We have replaced the second row in panel f with a correct version of WT PrP^C, in which WT PrP^C-expressing cells did not have any MyHC signal at day 3 (Lines 879-880, Legend of Fig. 3).

Fig. 3 | Maintenance of PrP^C homeostasis is important for myoblast differentiation. **f**, Immunofluorescence imaging of the above four cell lines incubated with differentiation medium for 3 and 5 days, respectively, using antibody against MyHC (red) and staining with DAPI (blue). Scale bar, 75 μ m. We have replaced the second row in panel f with a correct version of WT PrP^C, in which WT PrP^C-expressing cells did not have any MyHC signal at day 3.

Comment #3 • Figure 3:

1. The statement “localization of endogenous PrP^C, which is mainly attached to the plasma membrane, with the autophagy marker LC3B” is confusing. LC3B should not be in the PM. The image in 3a shows a strong patch of LC3B near the PM which overlaps with PrP, but is this a normal pattern for LC3B?
2. What does “endogenous PrP^C did form abundant LC3B puncta” mean? Do they mean PrP^C colocalizes with LC3B puncta?

REPLY: We apologize for this confusion. We totally agree that the statement “localization of endogenous PrP^C, which is mainly attached to the plasma membrane, with the autophagy marker LC3B” in the previous version is confusing and that LC3B should not be in the plasma membrane. We have now

replaced the first row in panel a with a correct version of control (Fig. 4a, page 32), as followed reviewer #1's nice suggestion. We have replaced the first row in panel a with a correct version of control, in which LC3B did not overlap with PrP^C (Lines 890-891, Legend of Fig. 4). To clarify and elaborate upon, we have now deleted the incorrect statement "After 3 days of differentiation, endogenous PrP^C did form abundant LC3B-positive puncta" from the manuscript (page 9, line 166) and have reworded and added the following sentences into the revised manuscript. *Endogenous PrP^C (red) was mainly attached to the plasma membrane and did not colocalize with the autophagy marker LC3B (green) in C2C12 cells (control) upon differentiation for 3 days (Fig. 4a). However, the colocalization of endogenous PrP^C and LC3B puncta (green dots, Fig. 4b) was observed in the elongated myotubes differentiated from C2C12 myoblasts after 5 days of differentiation (yellow dots in the merged image, Fig. 4b) (Lines 161-166).*

Fig. 4 | Excess PrP^C strongly inhibits skeletal muscle cell autophagy and blocks myoblast differentiation. a, Immunofluorescence imaging of C2C12 mouse myoblasts (control), C2C12 myoblasts stably overexpressing WT PrP^C, and C2C12 myoblasts KO for PrP^C cultured until their confluence reached 90% and then incubated with differentiation medium for 3 days, using antibodies against PrP^C (red) and LC3B (green) and staining with DAPI (blue). The enlarged regions in the lower left corner of the merged images show 4-fold enlarged images from the same images. Scale bar, 7.5 μ m. We have replaced the first row in panel a with a correct version of control, in which LC3B did not overlap with PrP^C.

3. No experiment is performed to probe whether autophagy is RESPONSIBLE for the effect of PrP overexpression on differentiation. In fact, PrP KO inhibited differentiation but not autophagy, implying that the two processes can be uncoupled.

REPLY: We sincerely thank reviewer #1 for this important suggestion that we should perform additional experiments to probe whether autophagy is responsible for the effect of PrP overexpression on differentiation!! According to the advice of reviewer #1, we have now provided autophagy data for the effect of PrP^C overexpression on differentiation (Extended Data Fig. 1, page 82). We have added the following paragraph into the Results section of the revision. *We performed additional experiments to investigate whether autophagy is responsible for the effect of overexpression of PrP^C on myoblast differentiation (Extended Data Fig. 1). Rapamycin, a widely used autophagy enhancer, was used to induce ATG5-dependent and LC3B-dependent autophagy in differentiating C2C12 cells stably expressing WT PrP^C (Extended Data Fig. 1a,d,e). Overexpression of PrP^C significantly inhibited skeletal muscle cell autophagy and blocked myoblast differentiation (Extended Data Fig. 1a–e). Compared with C2C12 myoblasts (control), incubation of C2C12 myoblasts stably expressing WT PrP^C with 5 nM rapamycin restored their autophagic activity and thus partially restored muscle cell differentiation (Extended Data Fig. 1a–e). Therefore, the inhibitory effect of excess PrP^C on autophagy in skeletal muscle partially results in an inhibitory effect of excess PrP^C on myoblast differentiation. Compared with C2C12 myoblasts (control), incubation of C2C12 myoblasts stably expressing WT PrP^C with 10 nM rapamycin significantly increased their autophagic activity and thus blocked muscle cell differentiation (Extended Data Fig. 1a–e). Together, the data showed that autophagy is partially responsible for the effect of overexpression of PrP^C on myoblast differentiation and that maintenance of autophagy homeostasis is also important for myoblast differentiation (Lines 194-210). We have added the following sentences into the Methods section of the revision. For analysis by western blotting treated with rapamycin, C2C12 cells were grown in a 6-well plate and cultured in a differentiation medium of 2% equine serum supplemented with 5 or 10 nM rapamycin (MedChem Express, AY22989) for 5 days, and the medium was changed daily. Cells were washed twice with ice-cold PBS and lysed in 300 μ l (per well) cell lysis buffer containing 1 \times protease inhibitor cocktail (Target*

Mol) (Lines 1138-1143). To address the concern from reviewer #1, we have now reworded and added the following sentence into the revised manuscript. Moreover, PrP^C deficiency resulted in increased autophagic activity in skeletal muscle cells (green dots, Fig. 4a,b) and blocked muscle cell differentiation when C2C12 myoblasts KO for PrP^C were incubated with this medium for 3 and 5 days (Lines 174-176), suggesting that the two processes (autophagy and differentiation) could be coupled.

Extended Data Fig. 1 | Autophagy is partially responsible for the effect of overexpression of PrP^C on myoblast differentiation. **a**, Western blot for PrP^C, the myogenic differentiation markers MyHC and MyoG, and the autophagy markers ATG5 and LC3B during C2C12 mouse myoblasts (control) and C2C12 myoblasts stably expressing full-length wild-type mouse PrP^C (WT PrP^C) cultured until their confluence reached 90% and then incubated with 5 or 10 nM rapamycin and differentiation medium for 5 days. β-actin served as the protein loading control. **b–e**, The relative amount of MyHC (**b**), MyoG (**c**), ATG5 (**d**) or LC3B-II (**e**) in the above cell lines (solid black circles shown in scatter plots) was expressed as the mean ± S.D. (with error bars) of values obtained in three independent experiments. One-way ANOVA and multiple comparisons were performed by SPSS 19.0 and different letters indicate significant differences at the level of $p < 0.05$.

Comment #4 • Figure 4:

1. 4f and 4g: assuming each orange bar is paired with a gray bar as control (IgG), the last one on each graph seems to be missing the control.

REPLY: We apologize for this confusion. Yes, each orange bar should be paired with a gray bar as control (IgG). To address the concern from reviewer #1, we have now introduced an axis break in the ordinate so that each orange bar is clearly paired with a gray bar as control (IgG) (Fig. 5f,g, page 48). An axis break was introduced in the ordinate so that each orange bar is clearly paired with a gray bar as control (IgG) (Lines 914-916, Legend of Fig. 5).

Fig. 5 | PrP^C selectively binds to a subset of miRNAs during myoblast differentiation. f,g, RT-qPCR validation of depleted versus enriched miRNAs in differentiating C2C12 myoblasts (control) (f) and those stably expressing WT PrP^C (g) normalized to total input. An axis break was introduced in the ordinate so that each orange bar is clearly paired with a gray bar as control (IgG). U6 snRNA served as a negative control. Data are presented as means \pm S.D. ($n = 3$ biologically independent measurements).

2. The overlap of miRNA pulled down between the two sets of RIP seems very small. Since PrP overexpression inhibits differentiation, could the difference in miRNA pulled down mainly due to the differentiation vs undifferentiation state?

REPLY: To address the question from reviewer #1, we have now added the following sentences into the Results section of the revision. *It should be mentioned that the overlap of miRNAs pulled down between the two sets of RIP is small (one third or one fifth of miRNAs identified for control RIP or WT PrP RIP). Since overexpression of PrP^C inhibited myoblast differentiation, it is possible that in two*

inputs the difference in miRNAs pulled down might be mainly due to the differentiation versus undifferentiation state (Lines 222-226).

3. The selective pulldown of specific miRNAs seems to be correlated with their abundance, i.e., the upregulated miRNAs under each condition got pulled down. This raises the question whether PrP pulldown of miRNA is really specific.

REPLY: To address the question from reviewer #1, we have now added the following sentences into the Results section of the revision. *The selective pulldown of specific miRNAs is not correlated with their abundance, because in our heat map, some miRNAs with low abundance (light blue) in two inputs were also strongly enriched in WT PrP^C RIP (deep red) (Fig. 5a). The above data suggest that PrP^C pulldown of miRNAs is specific (Lines 226-230).*

Fig. 5 | PrP^C selectively binds to a subset of miRNAs during myoblast differentiation. **a**, Profile of total miRNAs and PrP^C IP-enriched miRNAs at day 4 of differentiation in C2C12 myoblasts stably overexpressing WT PrP^C (WT PrP^C) and C2C12 myoblasts (control). Heat map shows the expression levels of miRNAs and fold-enrichment compared to Inputs in the above two cell lines.

Comment #5 • Figure 5:

1. 5c, d, e, f, g, h should be labeled as miR inhibitor rather than miR to avoid causing confusion.

REPLY: We apologize for this confusion. We totally agree that previous Figure 5c–h should be labeled as miR inhibitor rather than miR to avoid confusion. According to the advice of the reviewers #1 and 2, we have now labeled Fig. 6f–k as miR-214-3p inhibitor or miR-204-5p inhibitor to avoid confusion (Fig. 6f–k, page 50). *f–k*, Labeled as miR-214-3p inhibitor or miR-204-5p inhibitor to avoid confusion (Lines 947-948, Legend of Fig. 6).

Fig. 6 | PrP^C increases the stability of mature miR-214-3p to significantly inhibit autophagy and myoblast differentiation. **f, g**, Western blot for the autophagy markers ATG5 (**f**) and LC3B (**g**) and the myogenic differentiation marker MyHC after transfection with NC, miR-214-3p inhibitor (**f**) or miR-204-5p inhibitor (**g**) at 10 μ M in the above cell lines upon differentiation for 4 days. β -actin served as the protein loading control. **h–k**, The relative amount of ATG5 (**h**), LC3B-II (**j**) or MyHC (**i, k**) in

the above cell lines (solid black circles shown in scatter plots) was expressed as the mean \pm S.D. (with error bars) of values obtained in three independent experiments. Transfection of NC, miR-214-3p inhibitor (**h,i**) or miR-204-5p inhibitor (**j,k**) in C2C12 myoblasts KO for PrP^C (gray) and C2C12 myoblasts stably expressing WT PrP^C (orange or blue). **f–k**, Labeled as miR-214-3p inhibitor or miR-204-5p inhibitor to avoid confusion. **h–k**, One-way ANOVA and multiple comparisons were performed by SPSS 19.0 and different letters indicate significant differences at the level of $p < 0.05$.

2. The effect of miR inhibitor is there with either KO or PrP overexpression. Would it suggest that the miRNA actions are independent of their binding to PrP?

REPLY: To address the question from reviewer #1, we have now added the following sentence into the Results section of the revision. *Moreover, PrP^C overexpression enhanced the stability of miR-204-5p and to a lesser degree for miR-214-3p, suggesting that the miRNA actions are dependent on their binding to PrP^C (Lines 348-350).*

3. 5c, f, d, g: “ATG5-dependent autophagy” and “LC3B-dependent autophagy” are misleading; the authors were simply measuring ATG5 and LC3B levels as proxy for autophagy.

REPLY: Thank the reviewers #1’s comments regarding previous Figure 5c,f,d,g, in which “ATG5-dependent autophagy” and “LC3B-dependent autophagy” are misleading. The reviewer is correct. We have now revised and added the following sentences into the revised manuscript, as followed reviewer’s suggestion. *To gain a quantitative understanding of how miR-214-3p and miR-204-5p regulate skeletal muscle cell autophagy, we measured ATG5 and LC3B levels as proxies for autophagy and performed western blot analysis for ATG5, LC3B and the myogenic differentiation marker MyHC after transfection with NC, miR-214-3p inhibitor or miR-204-5p inhibitor at 10 μ M in the above cell lines (Fig. 6f,g). Thus, compared with transfection of NC in C2C12 myoblasts KO for PrP^C and C2C12 myoblasts stably expressing WT PrP^C, transfection of 10 μ M miR-214-3p inhibitor significantly promoted ATG5-related and myoblast differentiation in the above cell lines (Fig. 6f,h,i), suggesting that miR-214-3p significantly inhibits autophagy and differentiation of skeletal muscle cells via specific interaction with PrP^C. Upon differentiation for 4 days, transfection of 10 μ M miR-204-5p inhibitor*

significantly promoted LC3B-related autophagy in C2C12 myoblasts KO for PrP^C and C2C12 myoblasts stably expressing WT PrP^C compared to transfection of NC in the above cell lines (Fig. 6j). Therefore, inhibition of miR-214-3p significantly enhances both autophagy (ATG5) and myoblast differentiation in C2C12 cells (even when PrP^C is knocked out), and such enhancement is reduced when PrP^C is overexpressed. In PrP^C-KO C2C12 cells, inhibition of miR-204-5p significantly promotes autophagy (LC3B-II), but had no effect on myoblast differentiation (no differentiation); in C2C12 cells overexpressing PrP^C, inhibition of miR-204-5p enhances both autophagy (LC3B-II) and myoblast differentiation (Lines 319-348).

4. 5d, 5e: the conclusion "... after transfection with miR-214-3p inhibitor, excess PrP^C significantly inhibited both ATG5-dependent autophagy and myoblast differentiation..." cannot be derived from this data because comparing band intensities on western blots from two different cell lines would not be reliable. For this reason, the conclusion "These results demonstrate that miR-214-3p significantly inhibits ATG5-dependent autophagy and myoblast differentiation via specific interaction with PrP^C..." is not substantiated by data.

REPLY: We sincerely thank the reviewer for the comments. We totally agree that the conclusion "... after transfection with miR-214-3p inhibitor, excess PrP^C significantly inhibited both ATG5-dependent autophagy and myoblast differentiation..." cannot be derived from this data because comparing band intensities on western blots from two different cell lines would not be reliable. According to the reviewer's suggestion, we have now deleted the conclusion "... after transfection with miR-214-3p inhibitor, excess PrP^C significantly inhibited both ATG5-dependent autophagy and myoblast differentiation..." in the previous version from the manuscript (Line 332). We have revised the last sentence of this paragraph in the Results section as "*These results demonstrate that PrP^C increases the stability of mature miR-214-3p and miR-204-5p and enhances miRNA repression of their downstream mRNA targets to significantly inhibit autophagy and myoblast differentiation*" (Lines 351-353).

Comment #6 • Figure 6 and Figure 7

1. It is not at all clear how LLPS came into the picture after the authors found effects of PrP-associated miRNAs on autophagy.

REPLY: We sincerely thank reviewer #1 for this important suggestion that we should find the causal relationship between PrP^C LLPS and effects of PrP^C-associated miRNAs on autophagy!! We totally agree that in the previous version it is not clear how LLPS came into the picture after we found effects of PrP^C-associated miRNAs on autophagy. According to the advice of the reviewer, we have now provided evidence for causality between PrP^C LLPS and effects of PrP^C-associated miRNAs on autophagy (Fig. 8k–n, page 54) and have added the following sentences and paragraphs into the Results section of the revision. *PrP^C recruits miR-214-3p into its phase-separated condensate in living skeletal muscle cells, which results in the inhibition of autophagy. It is unclear how the LLPS of PrP^C (Fig. 7) came into the picture after we found effects of PrP^C-associated miRNAs on autophagy (Fig. 6). Given that PrP^C selectively recruits a subset of miRNAs into its phase-separated condensate, which in turn greatly enhances in vitro LLPS of PrP^C (Fig. 7), we predicted that miR-214-3p, an example of these PrP^C-associated miRNAs, might regulate in vivo LLPS of PrP^C and thus suppress skeletal muscle cell autophagy (page 21, lines 437-443). 1,6-hexanediol is an aliphatic alcohol that disturbs weak hydrophobic interactions involved in phase separation^{50,52,56}. To test whether hydrophobic interactions play a role in our observed puncta formation (Fig. 8c–f), we treated C2C12 cells expressing the mCherry-Cry2-WT PrP^C construct with 1,6-hexanediol and took time-lapse images (0–300 s) of the living cells (Fig. 8k). We observed that the LLPS of PrP^C (red puncta) in living C2C12 cells was dampened by treatment of 2.5% 1,6-hexanediol (Fig. 8k), suggesting that hydrophobic interactions play an important role in PrP^C liquid-phase condensation. To test the second half of this hypothesis and the functional relevance of the observations that we made in vitro and in vivo with PrP^C LLPS, we used C2C12 cells stably expressing mCherry-Cry2-WT PrP^C, which were cultured until their confluence reached 80% (Fig. 8l), transfected with or without 10 μM miR-214-3p for 30 min, and incubated with or without 2.5% 1,6-hexanediol for 5 min after activation by 488-nm laser for 10 min, using C2C12 myoblasts stably expressing mCherry-Cry2 as a control. To gain a quantitative understanding of how miR-214-3p regulates the LLPS of PrP^C in living myoblasts and thus regulates skeletal muscle cell autophagy, we performed western blot analysis for the autophagy markers ATG5 and LC3B in the above cell lines (Fig. 8l). Incubation of*

C2C12 cells stably expressing mCherry-Cry2-WT PrP^C with 1,6-hexanediol inhibited the phase-separation ability of PrP^C in cells (Fig. 8k) and thus caused a significant increase in ATG5 and LC3B levels as proxies for autophagy (Fig. 8m,n). Notably, transfection of C2C12 myoblasts stably expressing mCherry-Cry2-WT PrP^C with miR-214-3p enhanced the phase-separation ability of PrP^C in cells (Fig. 8j), thus inhibiting autophagy (Fig. 8m,n). Therefore, the recruitment of miR-214-3p into PrP^C condensates enhances the LLPS of PrP^C in skeletal muscle cells and results in the inhibition of autophagy (pages 23-24, lines 478-501). We have added the following sentences into the Methods section of the revision. For analysis by western blotting treated with 1,6-hexanediol (Sigma, H11807), C2C12 cells expressing the mCherry-Cry2-WT PrP^C construct were cultured until their confluence reached 85% in 6-well plate, then transfected with 10 μ M miR-214-3p using TransIT-X2 (Mirus) for 30 min in a growing medium, activation by 488-nm laser for 10 min, and then treated with 2.5% 1,6-hexanediol for 5 min. Cells were washed twice with ice-cold PBS and lysed in 300 μ l (per well) cell lysis buffer containing 1 \times protease inhibitor cocktail (Target Mol) (page 64, lines 1143-1150). Accordingly, the following sentences have been revised and added into the revision. We demonstrate that PrP^C, and selectively recruits these miRNAs into its phase-separated condensate in living myoblasts, which in turn greatly enhances liquid-liquid phase separation (LLPS) of PrP^C, promotes pathological aggregation of PrP, and results in the inhibition of autophagy and muscle bundle formation in myopathy patients characterized by incomplete muscle regeneration (Lines 22-29, Abstract). The recruitment of miR-214-3p into PrP^C condensates enhances the LLPS of PrP^C in skeletal muscle cells, results in the inhibition of autophagy, and promotes pathological aggregation of PrP (Lines 605-607, Discussion section).

2. What is the FUNCTIONAL relevance of all the observations on LLPS?

REPLY: We sincerely thank the reviewer for his (her) expert suggestion that we should address the functional relevance of all the observations on LLPS!! To address the functional relevance of the observations that we made in vitro and in vivo with PrP^C LLPS, we have provided functional data of PrP^C LLPS (Fig. 8k-n, page 54, autophagy; and Fig. 9, page 56, pathological aggregation) and have revised and added the following sentences and paragraphs into the Results

section of the revision. To test the second half of this hypothesis and the functional relevance of the observations that we made in vitro and in vivo with PrP^C LLPS, we used C2C12 cells stably expressing mCherry-Cry2-WT PrP^C, which were cultured until their confluence reached 80% (Fig. 8l), transfected with or without 10 μ M miR-214-3p for 30 min, and incubated with or without 2.5% 1,6-hexanediol for 5 min after activation by 488-nm laser for 10 min, using C2C12 myoblasts stably expressing mCherry-Cry2 as a control. To gain a quantitative understanding of how miR-214-3p regulates the LLPS of PrP^C in living myoblasts and thus regulates skeletal muscle cell autophagy, we performed western blot analysis for the autophagy markers ATG5 and LC3B in the above cell lines (Fig. 8l). Incubation of C2C12 cells stably expressing mCherry-Cry2-WT PrP^C with 1,6-hexanediol inhibited the phase-separation ability of PrP^C in cells (Fig. 8k) and thus caused a significant increase in ATG5 and LC3B levels as proxies for autophagy (Fig. 8m,n). Notably, transfection of C2C12 myoblasts stably expressing mCherry-Cry2-WT PrP^C with miR-214-3p enhanced the phase-separation ability of PrP^C in cells (Fig. 8j), thus inhibiting autophagy (Fig. 8m,n). Therefore, the recruitment of miR-214-3p into PrP^C condensates enhances the LLPS of PrP^C in skeletal muscle cells and results in the inhibition of autophagy (pages 23-24, lines 485-501). To further test the functional relevance of the observations that we made in vitro and in vivo with PrP^C LLPS, we used C2C12 cells stably expressing mCherry-Cry2-WT PrP^C, which were cultured until their confluence reached 85% (Fig. 9a,b) or 80% (Fig. 9c), transfected with or without 10 μ M miR-214-3p for 30 min, and cultured for 12 h after activation by 488-nm laser for 10 min, using C2C12 myoblasts stably expressing mCherry-Cry2 as a control (page 24, lines 510-515).

The PrP aggregates in C2C12 cells were demonstrated by the amount of PrP in sarkosyl-insoluble pellets from cells overexpressing the chimeric mCherry-Cry2-WT PrP^C (Fig. 9a), and were further proved by the amount of PrP in sarkosyl-insoluble pellets from cells overexpressing WT PrP^C (Fig. 9f). C2C12 myoblasts stably overexpressing full-length WT mouse PrP^C were cultured until their confluence reached 85% and then incubated with differentiation medium for 2 days, transfected without or with miR-214-3p and cultured for 48 h (Fig. 9f,g). The sarkosyl-insoluble pellets from the above cells were probed with anti-PrP antibody, and the corresponding cell lysates were probed using anti-PrP antibody and anti- β

actin antibody (Fig. 9f). Notably, in C2C12 cells stably expressing WT PrP^C incubated with differentiation medium for 4 days, miR-214-3p specifically interacted with PrP^C in the cytoplasm and transfection of 20 μM miR-214-3p significantly promoted pathological aggregation of PrP (Fig. 9g).

Because PrP aggregates are more resistant (than normal PrP^C) to digestion with modest amounts of proteases (such as proteinase K)^{7,8}, we wanted to determine whether the PrP aggregates in C2C12 cells show some level of protease resistance. C2C12 cells stably expressing mCherry-Cry2-WT PrP^C were cultured until their confluence reached 85%, transfected without or with 10 μM miR-214-3p for 30 min, and cultured for 12 h after activation by 488-nm laser for 10 min, and mCherry-Cry2 fusion alone was used as a control (Fig. 9h). The above cells were digested with various concentrations of proteinase K and probed with anti-PrP antibody. We found that PrP aggregates in the cell lysates from cells transfected without miRNA were completely digested with a proteinase K concentration as low as 0.75 μg/ml. In sharp contrast, PrP aggregates in the cell lysates from cells transfected with 10 μM miR-214-3p became resistant to 1.0 μg/ml proteinase K, producing a fragment that migrated at 27–30 kDa (Fig. 9h). Therefore, WT PrP^C can be induced by light to form liquid condensates in C2C12 cells; such PrP^C condensates recruit miR-214-3p, which in turn promotes the LLPS of PrP^C in C2C12 cells to evolve into PrP aggregates with some level of protease resistance.

Altogether these data demonstrate that PrP^C recruits miR-214-3p into its phase-separated condensate in living skeletal muscle cells, which in turn promotes pathological aggregation of PrP to form protease-resistant aggregates.

(The above paragraphs, pages 25-27).

Fig. 8 | PrP^C recruits miR-214-3p into its phase-separated condensate in living skeletal muscle cells, which results in the inhibition of autophagy. k, Time-lapse images (0–300 s) of living C2C12 cells expressing mCherry-Cry2-WT PrP^C treated with 2.5% 1,6-hexanediol. Scale bar, 5 μ m. **l,** Western blot for the autophagy markers ATG5 and LC3B in C2C12 cells expressing mCherry-Cry2-WT PrP^C transfected without (-) or with (+) 10 μ M miRNA and then treated with (+) or without (-) 1,6-hexanediol. β -actin served as the protein loading control. **m,n,** The relative amount of ATG5 (**m**) and LC3B-II (**n**) in the above cell lines (solid black circles shown in

scatter plots) was expressed as the mean \pm S.D. (with error bars) of values obtained in three independent experiments. One-way ANOVA and multiple comparisons were performed by SPSS 19.0 and different letters indicate significant differences at the level of $p < 0.05$.

Fig. 9 | PrP^C recruits miR-214-3p into its phase-separated condensate in living skeletal muscle cells, which in turn promotes pathological aggregation of PrP. **a**, Western blot for PrP in the sarkosyl-insoluble pellets and the corresponding cell lysates from C2C12 cells stably overexpressing mCh-Cry2-WT PrP^C transfected without (-) or with (+) 10 μ M miR-214-3p and cultured for 12 h after activation by 488-nm laser for 10 min. β -actin served as the protein loading control. **b**, The normalized amount of insoluble PrP aggregates in the above cell lines (solid black circles shown in scatter plots) was expressed as the mean \pm S.D. (with error bars) of values obtained in three independent experiments. **c**, The cell viability (%) (solid black circles shown in scatter plots) was measured by MTT reduction assay and expressed as the mean \pm S.D. (with error bars) of values obtained in five independent

experiments. **d,e**, Immunogold electron microscopy of PrP fibrils purified from the same C2C12 myoblasts stably expressing mCh-Cry2-WT PrP^C transfected without (**d**) or with (**e**) 10 μ M miR-214-3p as in (**a**), and labeled by gold particles conjugated with anti-PrP antibody. Scale bars, 100 nm. **f**, Western blot for PrP in the sarkosyl-insoluble pellets and the corresponding cell lysates from C2C12 myoblasts stably overexpressing WT PrP^C incubated with differentiation medium for 2 days, transfected without or with 10 or 20 μ M miR-214-3p and cultured for 48 h. β -actin served as the protein loading control. **g**, the same as in (**b**). **b,c,g**, One-way ANOVA and multiple comparisons were performed by SPSS 19.0 and different letters indicate significant differences at the level of $p < 0.05$. **h**, Western blot for PrP in the cell lysates from the same C2C12 cells stably overexpressing mCh-Cry2-WT PrP^C transfected without or with 10 μ M miR-214-3p as in (**a**), and digested with 0.25, 0.50, 0.75, 1.0, and 2.0 μ g/ml proteinase K.

Reviewer #2

Remarks to the Author:

Summary:

The authors report very interesting findings on the roles of PrP, LLPS composed of PrP and miRNAs, and certain miRNAs on myopathies associated with PrP accumulation/overexpression. They first detected much higher levels of PrP in the cytoplasm of skeletal muscles from 6 myopathy patients with dermatomyositis, neurogenic myopathy or muscular atrophy, and showed that there were numerous “dots” where PrP and miRNA-214-3p co-localize. Similar co-localization of PrP and miRNA-214-3p was also observed in “dots” in the cytoplasm of differentiating C2C12 cells overexpressing wild type mouse PrP. They went on to demonstrate the following with a number of techniques:

- 1. In mouse C2C12 myoblast cells, PrP knockout prevents myoblast differentiation.*
- 2. Overexpression of wild type or mutant PrP also prevents myoblast differentiation and most of the PrP is in the cytoplasm.*
- 3. In C2C12 cells, overexpression of wild type PrP strongly inhibits autophagy related proteins (LC3B and ATG5).*
- 4. PrP^C selectively binds to a subset of miRNAs, especially miRNA-214-3p and miRNA-204-5p in differentiating myoblasts (C2C12)*

5. *Inhibition of miRNA-214-3p significantly enhances both autophagy (ATG5) and myoblast differentiation in C2C12 cells (even when PrP is knocked out), and such enhancement is reduced when PrP is overexpressed.*
6. *In PrP-KO C2C12 cells, inhibition of miRNA-205-5p significantly promotes autophagy (LC3B-II), but had no effect on myoblast differentiation (no differentiation); in C2C12 cells overexpressing PrP, inhibition of miRNA-205-5p enhances both autophagy (LC3B-II) and myoblast differentiation.*
7. *PrP overexpression enhanced stability of miRNA-204-5p and to a lesser degree for miRNA-214-3p.*
8. *Recombinant PrP selectively recruits miRNA204-5p and miRNA-214-3p into its phase-separated condensate, which stimulates LLPS of PrP and the fusion of PrP droplets in vitro.*
9. *A chimeric PrP-mCh-Cry protein that can be induced by light to form liquid condensate shows that PrP LLPS can form in C2C12 cells; such LLPS recruits miRNA-214-3p, which in turn promotes PrP to form LLPS and evolve into PrP aggregates.*

These data demonstrate that overexpression of wild type PrP (or mutant PrP) is toxic to myoblasts via impairing autophagy and differentiation, that miRNA-214-3p and miRNA-204-5p enhance such PrP toxicity, that PrP selectively binds to miRNA-214-3p and miRNA-204-5p and recruits them into PrP liquid droplets, and that miRNA-214-3p enhances LLPS of PrP and the fusion of PrP droplets in vitro and the formation of PrP aggregates in C2C12 cells overexpressing PrP. These results are consistent with the authors' conclusion that "PrP^C located in the cytoplasm blocks muscle cell differentiation via selectively recruiting a subset of miRNAs including miR-214-3p into phase-separated condensates in living myoblasts, which in turn greatly enhances the LLPS of PrP^C and the subsequent PrP aggregation in vivo and significantly inhibits ATG5-dependent autophagy", providing insights on the possible cytotoxic mechanisms of PrP overexpression in skeletal muscles. The observation of many co-localized dots of PrP and miRNA-214-3p in the cytoplasm of a group of myopathy patients provide clinical relevance. The design is good and the data of high quality overall. The findings are very significant and meaningful.

We sincerely thank the reviewer for recognizing the significance of our work. The reviewer's suggestion is very valuable for us to improve our manuscript. Right now, we have added the following sentences into the revised manuscript, as

followed the advice of reviewer #2. We report interesting findings on the roles of PrP^C, LLPS composed of PrP^C and miRNAs, and certain miRNAs on myopathies associated with accumulation or overexpression of PrP^C (page 29, lines 619-621). In this study, we detected a much higher level of PrP^C in skeletal muscles from six myopathy patients with dermatomyositis, neurogenic myopathy or muscular atrophy than in skeletal muscles from one healthy control, two controls with lipid storage myopathy, and one control with glycogen storage disease (page 27, lines 567-570). These data clearly demonstrate that in C2C12 myoblasts, PrP^C knockout prevents myoblast differentiation. Overexpression of wild type or mutant PrP^C also prevents myoblast differentiation and most of the PrP^C is in the cytoplasm (page 8, lines 150-153). In C2C12 cells, overexpression of WT PrP^C strongly inhibited autophagy-related proteins (ATG5 and LC3B) (pages 9-10, lines 185-186). Together, these results demonstrate that PrP^C selectively binds to a subset of miRNAs, especially miR-214-3p and miR-204-5p in differentiating C2C12 myoblasts (page 14, lines 279-281). Therefore, inhibition of miR-214-3p significantly enhances both autophagy (ATG5) and myoblast differentiation in C2C12 cells (even when PrP^C is knocked out), and such enhancement is reduced when PrP^C is overexpressed. In PrP^C-KO C2C12 cells, inhibition of miR-204-5p significantly promotes autophagy (LC3B-II), but had no effect on myoblast differentiation (no differentiation); in C2C12 cells overexpressing PrP^C, inhibition of miR-204-5p enhances both autophagy (LC3B-II) and myoblast differentiation. Moreover, PrP^C overexpression enhanced the stability of miR-204-5p and to a lesser degree for miR-214-3p, suggesting that the miRNA actions are dependent on their binding to PrP^C (page 17, lines 342-350). Therefore, recombinant PrP^C selectively recruits miR-214-3p and miR-204-5p into its phase-separated condensate, which stimulates the LLPS of PrP^C and the fusion of PrP droplets in vitro (page 20, lines 415-417). Therefore, WT PrP^C can be induced by light to form liquid condensates in C2C12 cells; such PrP^C condensates recruit miR-214-3p, which in turn promotes the LLPS of PrP^C in C2C12 cells to evolve into PrP aggregates with some level of protease resistance (page 26, lines 555-557). Therefore, accumulation or overexpression of PrP^C is toxic to myoblasts via impairing autophagy and differentiation, and that miR-214-3p and miR-204-5p enhance the PrP toxicity (page 30, lines 629-631). These results provide insights on the possible cytotoxic mechanisms of accumulation or overexpression of PrP^C in skeletal muscles. The observation of many colocalized

dots of PrP^C and miR-214-3p in the cytoplasm of a group of myopathy patients provides clinical relevance (page 30, lines 639-642).

However, there are some minor to modest caveats.

Comment #1 • First, the PrP aggregates in C2C12 cells were demonstrated by the amount of PrP in sarkosyl-insoluble pellets of cell lysate from cells overexpressing the chimeric PrP-mCh-Cry2 (Fig. 7k). It would be more authentic if the same can be proven with cells expressing wild type mouse PrP. In addition, aggregated PrP is more resistant (than normal PrP^C) to digestion with modest amount of proteases (such as proteinase K, 5-100 µg/ml), so it would be helpful to show some level of protease resistance.

REPLY: We sincerely thank reviewer #2 for this important suggestion that we should further demonstrate the PrP aggregates in C2C12 cells overexpressing full-length WT mouse PrP^C!! We also sincerely thank the reviewer for his (her) expert suggestion that we should measure proteinase K-resistant activity of the PrP aggregates in C2C12 cells!! According to the advice of reviewer #2, we have now provided PrP aggregation data in C2C12 cells stably overexpressing WT PrP^C assessed by western blot (Fig. 9f,g, page 56) and protease digestion data of the PrP aggregates in C2C12 cells examined by western blot (Fig. 9h, page 56). We have added the following two paragraphs into the Results section of the revision.

The PrP aggregates in C2C12 cells were demonstrated by the amount of PrP in sarkosyl-insoluble pellets from cells overexpressing the chimeric mCherry-Cry2-WT PrP^C (Fig. 9a), and were further proved by the amount of PrP in sarkosyl-insoluble pellets from cells overexpressing WT PrP^C (Fig. 9f). C2C12 myoblasts stably overexpressing full-length WT mouse PrP^C were cultured until their confluence reached 85% and then incubated with differentiation medium for 2 days, transfected without or with miR-214-3p and cultured for 48 h (Fig. 9f,g). The sarkosyl-insoluble pellets from the above cells were probed with anti-PrP antibody, and the corresponding cell lysates were probed using anti-PrP antibody and anti-β-actin antibody (Fig. 9f). Notably, in C2C12 cells stably expressing WT PrP^C incubated with differentiation medium for 4 days, miR-214-3p specifically

interacted with PrP^C in the cytoplasm and transfection of 20 μM miR-214-3p significantly promoted pathological aggregation of PrP (Fig. 9g).

Because PrP aggregates are more resistant (than normal PrP^C) to digestion with modest amounts of proteases (such as proteinase K)^{7,8}, we wanted to determine whether the PrP aggregates in C2C12 cells show some level of protease resistance. C2C12 cells stably expressing mCherry-Cry2-WT PrP^C were cultured until their confluence reached 85%, transfected without or with 10 μM miR-214-3p for 30 min, and cultured for 12 h after activation by 488-nm laser for 10 min, and mCherry-Cry2 fusion alone was used as a control (Fig. 9h). The above cells were digested with various concentrations of proteinase K and probed with anti-PrP antibody. We found that PrP aggregates in the cell lysates from cells transfected without miRNA were completely digested with a proteinase K concentration as low as 0.75 μg/ml. In sharp contrast, PrP aggregates in the cell lysates from cells transfected with 10 μM miR-214-3p became resistant to 1.0 μg/ml proteinase K, producing a fragment that migrated at 27–30 kDa (Fig. 9h). Therefore, WT PrP^C can be induced by light to form liquid condensates in C2C12 cells; such PrP^C condensates recruit miR-214-3p, which in turn promotes the LLPS of PrP^C in C2C12 cells to evolve into PrP aggregates with some level of protease resistance.

Altogether these data demonstrate that PrP^C recruits miR-214-3p into its phase-separated condensate in living skeletal muscle cells, which in turn promotes pathological aggregation of PrP to form protease-resistant aggregates.

(The above paragraphs, pages 25-27).

A subsection titled “Proteinase K digestion assay” (pages 77-78) and the following sentences have been added into Methods section of the revision. Sarkosyl-insoluble western blotting was also used to investigate pathological aggregation of PrP in skeletal muscle cells by PrP specific binding to miRNA. C2C12 myoblasts stably expressing WT mouse PrP^C were cultured until their confluence reached 85% in 60-mm plates and then incubated with differentiation medium for 2 days, transfected without or with miR-214-3p using TransIT-X2 (Mirus) and cultured for 48 h (Lines 1361-1366).

Fig. 9 | PrP^C recruits miR-214-3p into its phase-separated condensate in living skeletal muscle cells, which in turn promotes pathological aggregation of PrP. **f**, Western blot for PrP in the sarkosyl-insoluble pellets and the corresponding cell lysates from C2C12 myoblasts stably overexpressing WT PrP^C incubated with differentiation medium for 2 days, transfected without or with 10 or 20 μM miR-214-3p and cultured for 48 h. β-actin served as the protein loading control. **g**, the same as in **(b)**. One-way ANOVA and multiple comparisons were performed by SPSS 19.0 and different letters indicate significant differences at the level of $p < 0.05$. **h**, Western blot for PrP in the cell lysates from the same C2C12 cells stably overexpressing mCh-Cry2-WT PrP^C transfected without or with 10 μM miR-214-3p as in **(a)**, and digested with 0.25, 0.50, 0.75, 1.0, and 2.0 μg/ml proteinase K.

Comment #2 • Second, the age-matched controls (age 32, 70, 63, 64) do not match well with the myopathy patients (age 43-52). Moreover, only one control (age 70) is a healthy control and two of the controls are lipid storage myopathy patients. So it is not myopathy patients vs. non-myopathy patients, rather it is the myopathy patients with excessive PrP in skeletal muscles vs others. It would help to make this point explicit.

REPLY: We apologize for this confusion. We totally agree that in the previous version the age-matched controls (age 32, 70, 63, and 64) do not match well with

the myopathy patients (age 43-52). According to the reviewer's suggestion, we have now deleted the imprecise terms "the age-matched controls" and "four age-matched controls" from the manuscript (Line 22; Line 92; Line 362; Lines 374-375; Lines 855-857, and Lines 1056-1057) and have used the term "*these four controls*" or "*four controls with normal levels of PrP^C*" instead of "the age-matched controls" or "four age-matched controls" throughout the revised manuscript. To clarify and elaborate upon, we have reworded and added the following sentences into the revised manuscript. *and the colocalization of PrP^C and miR-214-3p was clearly observed in the skeletal muscle of six myopathy patients with excessive PrP^C (Lines 20-22). We first took confocal images of frozen skeletal muscle sections from six myopathy patients with dermatomyositis, neurogenic myopathy or muscular atrophy, one healthy control, two controls with lipid storage myopathy, and one control with glycogen storage disease (Table 1 and Figs. 1a and 2a) (Lines 88-92). To test this hypothesis, we took confocal images of frozen skeletal muscle sections from the six myopathy patients and these four controls (Figs. 1a and 2a), (Lines 361-363). Excess PrP^C (red) was mainly located in the cytoplasm, and abundant miR-214-3p (green dots) and the colocalization of excess PrP^C and miR-214-3p (yellow dots in the merged images) were clearly observed not only in the skeletal muscle of six myopathy patients with excessive PrP^C (Fig. 1a) but also in differentiating C2C12 cells stably expressing WT PrP^C (Fig. 2d). In sharp contrast, accumulation of PrP^C, miR-214-3p expression, and the colocalization of PrP^C and miR-214-3p were not observed in the skeletal muscle of four controls with normal levels of PrP^C (Fig. 2a). Thus, a higher level of PrP^C colocalized with miR-214-3p in the skeletal muscle of the six myopathy patients (Fig. 1a). In sharp contrast, such a phenomenon was not observed not only in these four controls (Fig. 2a) but also in control C2C12 cells (Fig. 2d) (Lines 368-381). Fig. 2 / Accumulation of PrP^C and the colocalization of PrP^C and miR-214-3p PrP^C were not observed in the skeletal muscle of one healthy control (Control 1), two controls with lipid storage myopathy (LSM) (Controls 2 and 3), and one control with glycogen storage disease (GSD) (Control 4). a, Confocal images of frozen skeletal muscle sections from these four controls. We have replaced panel a with a correct version of these four controls, in which the control samples do have normal levels of PrP^C (red). b,c, H&E staining and immunohistochemical staining of the frozen*

sections showed that muscle regeneration was not observed in the skeletal muscle of these four controls (Lines 855-859, Legend of Fig. 2). Frozen skeletal muscle sections from six myopathy patients with dermatomyositis, neurogenic myopathy or muscular atrophy, one healthy control, two controls with lipid storage myopathy, and one control with glycogen storage disease (Table 1) (Lines 1054-1057).

Comment #3 • Third, the figure legends are too long and contain some redundancy.

REPLY: We sincerely thank the reviewer for this important suggestion to improve our manuscript. We totally agree that the figure legends of the previous manuscript are too long and contain some redundancy. We apologize for such extraordinary repetition in the previous manuscript. According to the advice of the reviewers #2 and 1, we have now made the figure legends more concise and have deleted the data interpretation and conclusion that are already in Results section from figure legends as follows.

Fig. 1 | Accumulation of PrP^C and the colocalization of PrP^C and miR-214-3p were clearly observed in the skeletal muscle of six myopathy patients. **a**, Confocal images of frozen skeletal muscle sections from six myopathy patients. These myopathy patients included two patients with dermatomyositis (DM) (Cases 1 and 2), two with neurogenic myopathy (NM) (Cases 3 and 4), and two with muscular dystrophy (MD) (Cases 5 and 6). Shown are nuclei stained with DAPI (blue); signals were detected with the anti-PrP antibody 8H4 (red), and miR-214-3p was detected by FISH (green) using an FAM-labeled miR-214-3p probe (FAM-anti-miR). The enlarged regions (right) show 4-fold enlarged images from the merged images. Arrows indicate colocalization of PrP^C and miR-214-3p in granules. Scale bars, 75 μm. **b**, H&E staining of the frozen sections showed that the skeletal muscle of the six myopathy patients was characterized by incomplete muscle regeneration. Dashed loops indicate centrally nucleated fibers. Scale bar, 400 μm. **c**, Immunohistochemical analysis of NCAM expression in the frozen skeletal muscle sections showed a positive signal (brown) in the muscle bundles. Scale bar, 100 μm (**pages 40-41**).

Fig. 2 | Accumulation of PrP^C and the colocalization of PrP^C and miR-214-3p PrP^C were not observed in the skeletal muscle of one healthy control (Control 1), two controls with lipid storage myopathy (LSM) (Controls 2 and 3), and one control with glycogen storage disease (GSD) (Control 4). **a**, Confocal images of

frozen skeletal muscle sections from these four controls. We have replaced panel a with a correct version of these four controls, in which the control samples do have normal levels of PrP^C (red). **b,c**, H&E staining and immunohistochemical staining of the frozen sections showed that muscle regeneration was not observed in the skeletal muscle of these four controls. The experimental conditions are the same as those in Fig. 1a–c. **d**, Colocalization of PrP^C and miR-214-3p in C2C12 mouse myoblasts (control) and C2C12 myoblasts stably expressing WT PrP^C upon differentiation for 4 days. Shown are nuclei stained with DAPI (blue); signals were detected with the anti-PrP antibody 8H4 (red), and miR-214-3p was detected by FISH (green). Scale bar, 10 μ m (**pages 42-43**).

Fig. 3 | Maintenance of PrP^C homeostasis is important for myoblast differentiation. **a**, Western blot for PrP^C and the myogenic differentiation markers MyHC and MyoG during C2C12 mouse myoblasts (control), C2C12 myoblasts stably expressing full-length wild-type mouse PrP^C (WT PrP^C), C2C12 myoblasts stably expressing F198S PrP^C, and C2C12 myoblasts KO for PrP^C cultured until their confluence reached 90% and then incubated with differentiation medium for 0, 3, and 5 days, respectively. β -actin served as the protein loading control. **b–d**, The relative amount of PrP^C (**b**), MyHC (**c**) or MyoG (**d**) in the above cell lines (solid black circles shown in scatter plots) was expressed as the mean \pm S.D. (with error bars) of values obtained in three independent experiments. One-way ANOVA and multiple comparisons were performed by SPSS 19.0 and different letters indicate significant differences at the level of $p < 0.05$. **e,f**, Immunofluorescence imaging of the above four cell lines incubated with differentiation medium for 3 and 5 days, respectively, using antibody against PrP^C (red) (**e**) or MyHC (red) (**f**) and staining with DAPI (blue). Scale bars, 50 (**e**) and 75 (**f**) μ m, respectively. We have replaced the second row in panel f with a correct version of WT PrP^C, in which WT PrP^C-expressing cells did not have any MyHC signal at day 3 (**pages 44-45**).

Fig. 4 | Excess PrP^C strongly inhibits skeletal muscle cell autophagy and blocks myoblast differentiation. **a,b**, Immunofluorescence imaging of C2C12 mouse myoblasts (control), C2C12 myoblasts stably overexpressing WT PrP^C, and C2C12 myoblasts KO for PrP^C cultured until their confluence reached 90% and then incubated with differentiation medium for 3 (**a**) and 5 (**b**) days, respectively, using antibodies against PrP^C (red) and LC3B (green) and staining with DAPI (blue). The

enlarged regions in the lower left corner (**a**) or the lower right corner (**b**) of the merged images show 4-fold enlarged images from the same images. Scale bars, 7.5 (**a**) and 50 (**b**) μm , respectively. We have replaced the first row in panel a with a correct version of control, in which LC3B did not overlap with PrP^C. **c**, Western blot for PrP^C and the autophagy markers ATG5 and LC3B during the above three cell lines incubated with differentiation medium for 3 and 5 days, respectively. β -actin served as the protein loading control. **d,e**, The relative amount of ATG5 (**d**) or LC3B-II (**e**) in the above cell lines (solid black circles shown in scatter plots) was expressed as the mean \pm S.D. (with error bars) of values obtained in three independent experiments. One-way ANOVA and multiple comparisons were performed by SPSS 19.0 and different letters indicate significant differences at the level of $p < 0.05$ (**pages 46-47**).

Fig. 5 | PrP^C selectively binds to a subset of miRNAs during myoblast differentiation. **a**, Profile of total miRNAs and PrP^C IP-enriched miRNAs at day 4 of differentiation in C2C12 myoblasts stably overexpressing WT PrP^C (WT PrP^C) and C2C12 myoblasts (control). Heat map shows the expression levels of miRNAs and fold-enrichment compared to Inputs in the above two cell lines. **b**, Western blotting analysis of PrP^C immunoprecipitation in C2C12 myoblasts at day 4 of differentiation by probing PrP^C and GAPDH. **c**, Volcano plot of 31 PrP^C-bound miRNAs identified by small RNA-seq in differentiating C2C12 cells (control). **d**, Volcano plot of 51 PrP^C-bound miRNAs identified by small RNA-seq in differentiating C2C12 cells stably expressing WT PrP^C. Representative PrP^C RIP-enriched miRNAs are highlighted in orange. **e**, Venn diagram of miRNAs enriched by PrP^C RIP in C2C12 cells stably expressing WT PrP^C versus miRNAs enriched by PrP^C RIP in C2C12 myoblasts (control). Group I ($n = 21$, Control RIP only), $n = 10$ (overlap), and group II ($n = 41$, WT PrP RIP only). **f,g**, RT-qPCR validation of depleted versus enriched miRNAs in differentiating C2C12 myoblasts (control) (**f**) and those stably expressing WT PrP^C (**g**) normalized to total input. An axis break was introduced in the ordinate so that each orange bar is clearly paired with a gray bar as control (IgG). U6 snRNA served as a negative control. Data are presented as means \pm S.D. ($n = 3$ biologically independent measurements). **h,i**, The top six biological processes (BP) of Gene Ontology (GO) enrichment in the control cells (**h**) and the top eight BPs of GO enrichment in C2C12 cells stably overexpressing WT PrP^C (**i**). **j**, Pulldown of endogenous PrP^C and excess PrP^C with WT and mutant biotin-labeled miR-214-3p or

biotin-labeled miR-204-5p in the above two cell lines at day 4 of differentiation (pages 48-49).

Fig. 6 | PrP^C increases the stability of mature miR-214-3p to significantly inhibit autophagy and myoblast differentiation. **a,b**, Dual luciferase reporter assays, in which miR-214-3p (**a**) or miR-204-5p (**b**) represses the wild-type 3' end of untranslated region (WT 3'UTR) of the gene *ATG5* (**a**) or *LC3B* (**b**) associated with luciferase. Each pair of reporters was assayed in response to specific miRNA mimics at day 4 of differentiation in C2C12 myoblasts stably overexpressing WT PrP^C or KO for PrP^C. The relative luciferase activity of the reporters after transfection with NC (gray), miR-214-3p mimic (orange) or miR-204-5p mimic (blue) in the above cell lines (solid black circles shown in scatter plots) was expressed as the mean \pm S.D. (with error bars) of values obtained in three independent experiments. **c–e**, The relative stability of miR-214-3p (**c**), miR-204-5p (**d**), and nonspecific miR-183-5p (**e**) in C2C12 myoblasts stably overexpressing WT PrP^C (red) or KO for PrP^C (blue) incubated with differentiation medium for 4 days and then treated with 20 mg/ml α -amanitin was determined as the relative enrichment (to U6 snRNA) of miRNAs by RT-qPCR and expressed as the mean \pm S.D. (with error bars) of values obtained in three independent experiments. **f,g**, Western blot for the autophagy markers ATG5 (**f**) and LC3B (**g**) and the myogenic differentiation marker MyHC after transfection with NC, miR-214-3p inhibitor (**f**) or miR-204-5p inhibitor (**g**) at 10 μ M in the above cell lines upon differentiation for 4 days. β -actin served as the protein loading control. **h–k**, The relative amount of ATG5 (**h**), LC3B-II (**j**) or MyHC (**i,k**) in the above cell lines (solid black circles shown in scatter plots) was expressed as the mean \pm S.D. (with error bars) of values obtained in three independent experiments. Transfection of NC, miR-214-3p inhibitor (**h,i**) or miR-204-5p inhibitor (**j,k**) in C2C12 myoblasts KO for PrP^C (gray) and C2C12 myoblasts stably expressing WT PrP^C (orange or blue). **f–k**, Labeled as miR-214-3p inhibitor or miR-204-5p inhibitor to avoid confusion. **a–e**, Statistical analyses were performed using a two-tailed unpaired *t* test. Values of $p < 0.05$ indicate statistically significant differences. The following notation is used throughout: * $p < 0.05$; ** $p < 0.01$; *** $p < 0.001$; and **** $p < 0.0001$ relative to controls. n.s., no significance. **h–k**, One-way ANOVA and multiple comparisons were performed by SPSS 19.0 and different letters indicate significant differences at the level of $p < 0.05$ (pages 50-51).

Fig. 7 | PrP^C selectively recruits a subset of miRNAs into its phase-separated condensate, which in turn greatly enhances in vitro LLPS of PrP^C. **a**, Regulation of PrP^C LLPS by three miRNAs. Fluorescence images of 50 μ M recombinant wild-type mouse PrP^C (WT PrP^C) labeled by TAMRA (red) and incubated with 1 \times PBS (pH 7.4) containing 0, 1.25, 2.5, 5, 10 or 20 μ M miRNA on ice for 5 min. Scale bar, 7.5 nm. **b**, PrP^C selectively recruits miR-214-3p and miR-204-5p into its phase-separated condensate. Fluorescence images of in vitro phase-separated droplets (red; Merge: yellow) of 50 μ M TAMRA-labeled WT PrP^C incubated with 1 \times PBS containing 10 μ M FAM-labeled miRNA (green) on ice for 5 min. Scale bar, 2.5 nm. **c**, FRAP analysis on the selected liquid droplets of 50 μ M TAMRA-labeled WT PrP^C before (prebleach), during (0 s), and after photobleaching (1, 3, 6 and 30 s, respectively). The internal photobleaching is marked by a black square. Scale bar, 2.5 nm. **d**, Normalized kinetics of fluorescence recovery data of WT PrP^C (blue circle), WT PrP^C + miR-214-3p (red square), WT PrP^C + miR-204-5p (magenta square), and WT PrP^C + miR-183-5p (olive circle) obtained from FRAP intensity. The normalized fluorescence intensity is expressed as the mean \pm S.D. (with error bars) of values obtained in three independent experiments. The solid lines show the best single exponential fit for the fluorescence intensity-time curves (**pages 52-53**).

Fig. 8 | PrP^C recruits miR-214-3p into its phase-separated condensate in living skeletal muscle cells, which results in the inhibition of autophagy. **a**, Schematic of the optoIDR assay, depicting recombinant protein with an IDR (blue), mCherry (red), and Cry2 (orange) expressed in cells exposed to 488-nm excitation laser. **b,c**, Time-lapse images of living C2C12 cells expressing mCherry-Cry2-WT PrP^C construct containing PrP₁₋₃₇ IDR (residues 1–37) (blue) linked to mCherry (red) and Cry2 (orange) and then linked to PrP₃₈₋₂₃₀ IDR (residues 38–230) (blue) (**c**), and mCherry-Cry2 fusion alone was used as a control (**b**). **d,e**, Time-lapse images (0–100 s) of a C2C12 cell expressing the mCherry-Cry2-WT PrP^C construct. A droplet fusion event occurs in the region highlighted by the orange box (**d**). The droplet fusion event highlighted by using orange arrows (**e**). **b–e**, Cells were subjected to laser excitation every 2 s for the indicated time. Scale bars, 7.5 μ m. **f**, Representative live-cell images of PrP^C-miRNA speck formation when 10 μ M FAM-labeled miRNA (green) was transfected into C2C12 cells stably expressing mCherry (red) or WT PrP^C-mCherry (red), among \geq 10 cells. Scale bars, 7.5 μ m. **g,h**, FRAP analysis on the selected liquid

droplets of PrP^C (red) in living C2C12 cells expressing mCherry-Cry2-WT PrP^C transfected without (**g**) or with (**h**) miRNA before (prebleach), during (0 s), and after photobleaching (60 and 125 s, respectively). The dashed white circle highlights the punctum undergoing targeted bleaching. Scale bars, 7.5 μ m. **i**, Quantification of FRAP data for WT PrP^C puncta without or with miRNA ($n = 3$), as in **g,h**. Time 0 indicates the start of recovery after photobleaching. **j**, Normalized kinetics of fluorescence recovery data of WT PrP^C puncta (blue circle) and WT PrP^C puncta + miR-214-3p (red square). The normalized fluorescence intensity is expressed as the mean \pm S.D. (with error bars) of values obtained in three independent experiments. The solid lines show the best single exponential fit for the fluorescence intensity-time curves. **k**, Time-lapse images (0–300 s) of living C2C12 cells expressing mCherry-Cry2-WT PrP^C treated with 2.5% 1,6-hexanediol. Scale bar, 5 μ m. **l**, Western blot for the autophagy markers ATG5 and LC3B in C2C12 cells expressing mCherry-Cry2-WT PrP^C transfected without (-) or with (+) 10 μ M miRNA and then treated with (+) or without (-) 1,6-hexanediol. β -actin served as the protein loading control. **m,n**, The relative amount of ATG5 (**m**) and LC3B-II (**n**) in the above cell lines (solid black circles shown in scatter plots) was expressed as the mean \pm S.D. (with error bars) of values obtained in three independent experiments. One-way ANOVA and multiple comparisons were performed by SPSS 19.0 and different letters indicate significant differences at the level of $p < 0.05$ (pages 54-55).

Fig. 9 | PrP^C recruits miR-214-3p into its phase-separated condensate in living skeletal muscle cells, which in turn promotes pathological aggregation of PrP. **a**, Western blot for PrP in the sarkosyl-insoluble pellets and the corresponding cell lysates from C2C12 cells stably overexpressing mCh-Cry2-WT PrP^C transfected without (-) or with (+) 10 μ M miR-214-3p and cultured for 12 h after activation by 488-nm laser for 10 min. β -actin served as the protein loading control. **b**, The normalized amount of insoluble PrP aggregates in the above cell lines (solid black circles shown in scatter plots) was expressed as the mean \pm S.D. (with error bars) of values obtained in three independent experiments. **c**, The cell viability (%) (solid black circles shown in scatter plots) was measured by MTT reduction assay and expressed as the mean \pm S.D. (with error bars) of values obtained in five independent experiments. **d,e**, Immunogold electron microscopy of PrP fibrils purified from the same C2C12 myoblasts stably expressing mCh-Cry2-WT PrP^C transfected without (**d**)

or with (e) 10 μ M miR-214-3p as in (a), and labeled by gold particles conjugated with anti-PrP antibody. Scale bars, 100 nm. f, Western blot for PrP in the sarkosyl-insoluble pellets and the corresponding cell lysates from C2C12 myoblasts stably overexpressing WT PrP^C incubated with differentiation medium for 2 days, transfected without or with 10 or 20 μ M miR-214-3p and cultured for 48 h. β -actin served as the protein loading control. g, the same as in (b). b,c,g, One-way ANOVA and multiple comparisons were performed by SPSS 19.0 and different letters indicate significant differences at the level of $p < 0.05$. h, Western blot for PrP in the cell lysates from the same C2C12 cells stably overexpressing mCh-Cry2-WT PrP^C transfected without or with 10 μ M miR-214-3p as in (a), and digested with 0.25, 0.50, 0.75, 1.0, and 2.0 μ g/ml proteinase K (pages 56-57).

Fig. 10 | A hypothetical model shows how excess PrP^C inhibits muscle cell differentiation via miRNA-enhanced LLPS of PrP^C implicated in myopathy. Under pathological conditions (right), PrP^C (orange) is overexpressed in muscle satellite cells (orange) and myoblasts (pale blue). Then, PrP^C selectively recruits miR-214-3p and miR-204-5p into phase-separated condensates (gold balls) in living myoblasts to block autophagy and preserve myoblasts undifferentiated. This recruitment in turn greatly enhances PrP^C liquid phase condensation and the subsequent aggregation to produce PrP fibrils (gold bars), resulting in skeletal muscle cell death and muscle bundle formation in myopathy patients. Under physiological conditions (left), PrP^C (orange) is expressed in myoblasts (pale blue) and enhances muscle cell differentiation by selectively interacting with miR-486a-5p and miR-181b-5 to significantly promote autophagy, and finally, muscle bundles are assembled to create the whole muscle (page 58).

Comment #4 • Fourth, in Fig. 5c-k, the data labeled miR-214-3p or miRNA-204-5p actually mean “inhibitor” of miR-214-3p or miRNA-204-5p. It would be better to label as such to avoid confusion.

REPLY: We apologize for this confusion. We totally agree that previous Figure 5c–h should be labeled as miR inhibitor rather than miR to avoid confusion. According to the advice of the reviewers #2 and 1, we have now labeled Fig. 6f–k as miR-214-3p inhibitor or miR-204-5p inhibitor to avoid confusion (Fig. 6f–k,

page 50). *f-k*, Labeled as *miR-214-3p inhibitor* or *miR-204-5p inhibitor* to avoid confusion (Lines 947-948, Legend of Fig. 6).

Fig. 6 | PrP^C increases the stability of mature miR-214-3p to significantly inhibit autophagy and myoblast differentiation. *f,g*, Western blot for the autophagy markers ATG5 (*f*) and LC3B (*g*) and the myogenic differentiation marker MyHC after transfection with NC, miR-214-3p inhibitor (*f*) or miR-204-5p inhibitor (*g*) at 10 μ M in the above cell lines upon differentiation for 4 days. β -actin served as the protein loading control. *h-k*, The relative amount of ATG5 (*h*), LC3B-II (*j*) or MyHC (*i,k*) in the above cell lines (solid black circles shown in scatter plots) was expressed as the mean \pm S.D. (with error bars) of values obtained in three independent experiments. Transfection of NC, miR-214-3p inhibitor (*h,i*) or miR-204-5p inhibitor (*j,k*) in C2C12 myoblasts KO for PrP^C (gray) and C2C12 myoblasts stably expressing WT

PrP^C (orange or blue). **f–k**, Labeled as miR-214-3p inhibitor or miR-204-5p inhibitor to avoid confusion. **h–k**, One-way ANOVA and multiple comparisons were performed by SPSS 19.0 and different letters indicate significant differences at the level of $p < 0.05$.

Comment #5 • Fifth, many pictures are too small and of low resolutions that makes it hard to discern key details and structures (such as membrane vs cytoplasmic PrP distribution in Fig. 2c).

REPLY: Thanks the reviewer for the comments concerning the previous confocal images. We totally agree that in the previous version many pictures are too small and of low resolutions that makes it hard to discern key details and structures (such as membrane versus cytoplasmic PrP^C distribution in previous Fig. 2e). According to the advice of reviewer #2, we have now provided high-resolution confocal images of frozen skeletal muscle sections from six myopathy patients and these four controls (Fig. 1a, page 40; and Fig. 2a, page 42) and high-resolution confocal images of C2C12 cells incubated with differentiation medium for 3 and 5 days (Fig. 3e, page 44; and Fig. 4a,b, page 46), which clearly show membrane versus cytoplasmic PrP^C distribution.

Fig. 1 | Accumulation of PrP^C and the colocalization of PrP^C and miR-214-3p were clearly observed in the skeletal muscle of six myopathy patients. a, Confocal images of frozen skeletal muscle sections from six myopathy patients. These myopathy patients included two patients with dermatomyositis (DM) (Cases 1 and 2), two with neurogenic myopathy (NM) (Cases 3 and 4), and two with muscular dystrophy (MD) (Cases 5 and 6). Shown are nuclei stained with DAPI (blue); signals

were detected with the anti-PrP antibody 8H4 (red), and miR-214-3p was detected by FISH (green) using an FAM-labeled miR-214-3p probe (FAM-anti-miR). The enlarged regions (right) show 4-fold enlarged images from the merged images. Arrows indicate colocalization of PrP^C and miR-214-3p in granules. Scale bars, 75 μ m.

Fig. 2 | Accumulation of PrP^C and the colocalization of PrP^C and miR-214-3p PrP^C were not observed in the skeletal muscle of one healthy control (Control 1), two controls with lipid storage myopathy (LSM) (Controls 2 and 3), and one control with glycogen storage disease (GSD) (Control 4). **a**, Confocal images of frozen skeletal muscle sections from these 4 controls. We have replaced panel a with a correct version of these four controls, in which the control samples do have normal levels of PrP^C (red). Scale bar, 10 μ m.

Fig. 3 | Maintenance of PrP^C homeostasis is important for myoblast differentiation. e, Immunofluorescence imaging of C2C12 mouse myoblasts (control), C2C12 myoblasts stably expressing full-length wild-type mouse PrP^C (WT PrP^C), C2C12 myoblasts stably expressing F198S PrP^C, and C2C12 myoblasts KO for PrP^C cultured until their confluence reached 90% and then incubated with differentiation medium for 3 and 5 days, respectively, using antibody against PrP^C (red) and staining with DAPI (blue). Scale bar, 50 μ m.

Fig. 4 | Excess PrP^C strongly inhibits skeletal muscle cell autophagy and blocks myoblast differentiation. **a**, Immunofluorescence imaging of C2C12 mouse myoblasts (control), C2C12 myoblasts stably overexpressing WT PrP^C, and C2C12 myoblasts KO for PrP^C cultured until their confluence reached 90% and then incubated with differentiation medium for 3 days, using antibodies against PrP^C (red) and LC3B (green) and staining with DAPI (blue). The enlarged regions in the lower left corner of the merged images show 4-fold enlarged images from the same images. Scale bar, 7.5 μ m. We have replaced the first row in panel a with a correct version of control, in which LC3B did not overlap with PrP^C.

Fig. 4 | Excess PrP^C strongly inhibits skeletal muscle cell autophagy and blocks myoblast differentiation. **b**, Immunofluorescence imaging of C2C12 mouse myoblasts (control), C2C12 myoblasts stably overexpressing WT PrP^C, and C2C12 myoblasts KO for PrP^C cultured until their confluence reached 90% and then incubated with differentiation medium for 5 days, using antibodies against PrP^C (red) and LC3B (green) and staining with DAPI (blue). The enlarged regions in the lower right corner of the merged images show 4-fold enlarged images from the same images. Scale bar, 50 μ m.

Comment #6 • Sixth, some phrases/words are less than optimal or incorrect in the context. It would help to have a qualified scientific proofreader to edit the text.

REPLY: According to the suggestion of the reviewer, we have now used one of the many qualified scientific proofreaders available, American Journal Experts, to edit the text. Therefore, the English language in the text has benefited from improvement for clarity and readability. The followings are examples. *We demonstrate that PrP^C is overexpressed in skeletal muscle cells under pathological conditions, inhibits muscle cell differentiation by physically interacting with a subset of miRNAs to significantly inhibit autophagy-related protein 5-dependent autophagy, and selectively recruits these miRNAs into its phase-separated*

condensate in living myoblasts, which in turn greatly enhances liquid–liquid phase separation (LLPS) of PrP^C, promotes pathological aggregation of PrP, and results in the inhibition of autophagy and muscle bundle formation in myopathy patients characterized by incomplete muscle regeneration (Lines 22-29). Mammalian prion protein (PrP) has two forms that are distinct in their structure and function, the cellular prion protein (PrP^C) and its pathological aggregated form PrP^{Sc} (refs. 1-6) (Lines 32-34). Currently, pathological prion aggregates have been shown to trigger autophagy in skeletal muscle and can be degraded by autophagy³³. However, it is unclear whether the benign cellular form of PrP regulates autophagy and differentiation of skeletal muscle cells (Lines 52-55). However, it is unclear whether miRNAs regulate autophagy and differentiation of skeletal muscle cells via specific interactions with PrP^C (Lines 62-63). We next used HDOCK, a protein–protein/nucleic acid protein docking web server that combines template-based and free docking⁷⁰, (Lines 262-263). Notably, compared with that in C2C12 myoblasts KO for PrP^C, excess PrP^C significantly enhanced not only the binding affinity of miR-214-3p toward its downstream target, the autophagy marker ATG5 (Fig. 6a) ($p = 0.00021$), but also the binding affinity of miR-204-5p toward its downstream target, the autophagy marker LC3B (Fig. 6b) ($p = 0.0047$), in C2C12 myoblasts stably expressing WT PrP^C when incubated with the differentiation medium for 4 days (Lines 305-310). Based on these observations, we hypothesized that PrP^C colocalizes with miR-214-3p in the skeletal muscle of myopathy patients (Lines 359-361). PrP^C formed abundant liquid droplets, and protein condensates formed by PrP^C became much larger in the presence of low concentrations of miR-214-3p or miR-204-5p than in the absence of miRNA (Fig. 7a) (Lines 396-398). The addition of miR-214-3p or miR-204-5p at low concentrations dramatically promoted but the addition of miR-183-5p or miR-214-3p mutant did not dramatically enhance the phase separation (Fig. 7a) and droplet fusion ability (Fig. 7b) of PrP^C (Lines 412-415). Together, the data showed that a subset of miRNAs recruited by PrP^C forms a positive feed-forward loop with PrP^C to greatly enhance in vitro LLPS of PrP^C (Lines 417-419). FRAP of phase-separated PrP^C droplets without miRNA or with a negative control miR-183-5p revealed fluorescence recovery of $(82.5 \pm 0.3)\%$ or $(65.1 \pm 0.3)\%$ within 30 s (Fig. 7d). In sharp contrast, FRAP of phase-separated PrP^C droplets coacervated with miR-214-3p or miR-204-

5p revealed much lower fluorescence recovery, (34.5 ± 0.8)% or (24.9 ± 0.3)%, within 30 s (Fig. 7d). According to Fig. 7c,d, miR-214-3p and miR-204-5p reduced fluorescence recovery. This means that miR-214-3p and miR-204-5p decrease the fluidity of LLPS condensates, possibly because these miRNAs could modulate liquid-to-solid transitions in phase-separated PrP^C condensates. The above experiments help drive the narrative that miR-214-3p and miR-204-5p at low concentrations decrease fluorescence recovery and modulate the liquid nature of PrP^C droplets in vitro (Lines 421-431). Therefore, PrP^C regulates autophagy and differentiation of skeletal muscle cells via multiple mechanisms, the first being physical interaction with these specific miRNAs (Lines 591-593). We show that PrP^C undergoes LLPS in vitro and within cells. PrP^C condensates selectively recruit a couple of miRNAs (miR-214-3p and miR-204-5p) in vitro and within skeletal muscle cells, which in turn dramatically promotes the LLPS of PrP^C under both conditions (Lines 596-599). The selective interaction of miRNAs with PrP^C during cell differentiation will be valuable to understanding the functional basis underlying LLPS of proteins and inspiring future research on protein condensation diseases caused by abnormal liquid-like or solid-like states of proteins⁸³ and regulated by RNA⁴³⁻⁴⁶ (Lines 642-646).

Below are some examples:

- Lines 141-142, “After 5 days of differentiation, however, excess WT PrP^C is completely located in the cytoplasm” (Fig. 2e) is an overstatement. It is nearly impossible to be sure that PrP is not on the cell membrane when there are plenty of PrP in the cytoplasm and examined by confocal microscopy alone.

REPLY: Thank for pointing this out. Right now, we have replaced the overstatement “After 5 days of differentiation, however, excess WT PrP^C is completely located in the cytoplasm (Fig. 2e)” with “After 5 days of differentiation, however, excess WT PrP^C was mainly located in the cytoplasm (Fig. 3e)” in our manuscript (Lines 147-148), as followed reviewer’s suggestion.

- Line 145, “maintenance of PrP^C homeostasis is essential for myoblast differentiation” is an overstatement. It is well known that PrP-KO mice are basically normal, including their skeletal muscles, under normal conditions. So “essential” should be replaced with something like “important”.

REPLY: Thank for pointing this out. Right now, we have replaced “essential” with “important” in our manuscript (Lines 154, 114, 622, and 864), as followed reviewer’s suggestion.

- Line 47, “conservative” should be “conserved”.

REPLY: Thank for pointing this out. Right now, we have replaced “conservative” with “conserved” in our manuscript (Line 45), as followed reviewer’s suggestion.

- Line 119, “until confluence was reached 90%” should be “until confluence reached 90%”.

REPLY: Thank for pointing this out. Right now, we have replaced the imprecise phrase “until confluence was reached 90%” with “until their confluence reached 90%” in our manuscript (Lines 123-124), as followed reviewer’s suggestion.

- Line 123, “more than 90% sequence identity” should be “more than 90% sequence homology”.

REPLY: Thank for pointing this out. Right now, we have replaced the imprecise phrase “more than 90% sequence identity” with “more than 90% sequence homology” in our manuscript (Lines 126-127), as followed reviewer’s suggestion.

- Line 98, “PrP^C (red) was overexpressed” is inaccurate since no study was done to demonstrate overexpression of the PrP gene in this study. It is more accurate to state “a higher level of PrP was accumulated”.

REPLY: Thank for pointing this out. Right now, we have replaced the inaccurate statement “PrP^C (red) was overexpressed and mainly located ...” with “A higher level of PrP^C (red) accumulated ...” in our manuscript (Line 94), as followed reviewer’s suggestion.

REVIEWER COMMENTS

Reviewer #1 (Remarks to the Author):

The authors' significant efforts to address the previous criticisms are appreciated. However, concerns remain for the current version of the manuscript.

1. In response to my questions about data in the previous version, several times authors claim they "made a mistake" (without stating the nature of the mistake) and have now replaced the old data with new ones – showing what I indicated would have been expected, but without explaining what caused the "mistakes". (See Fig. 1, Fig. 2, and Fig. 3.) This does not bolster overall confidence in the rigor and/or truthfulness of the study.

2. Why does the DAPI staining vary so much from sample to sample in Figs. 1A and 2A? All the samples are muscle sections, and yet some images had numerous and strong DAPI signals and others had very few.

3. To address my question #3 regarding the previous Fig. 3, authors used rapamycin to induce autophagy (current Fig. 4). However, rapamycin does not seem to be an appropriate tool here because rapamycin is well documented to inhibit C2C12 differentiation through various mTOR mechanisms that are independent of autophagy.

4. Regarding my point #3 on previous Fig. 5, "The effect of miR inhibitor is there with either KO or PrP overexpression. Would it suggest that the miRNA actions are independent of their binding to PrP?", the authors responded by simply not mentioning the PrP KO data. Therefore, my original question is not addressed.

5. "It is unclear how the LLPS of PrPC (Fig. 7) came into the picture after we found effects of PrPC-associated miRNAs on autophagy (Fig. 6)." – this sentence should be deleted from the manuscript. The authors are repeating my previous question, which was meant to ask for rationale for the transition from Fig. 6 to experiments in Fig. 7.

6. The authors added new data (Figs. 8 and 9) to address the role of LLPS. I do not have sufficient expertise to evaluate these experiments.

Reviewer #2 (Remarks to the Author):

The authors have adequately addressed my main concerns, including conducting the recommended additional experiments with satisfactory results. The manuscript has been significantly improved. The writing can still benefit from further professional editing.

Reviewer #1

Remarks to the Author:

The authors' significant efforts to address the previous criticisms are appreciated.

We sincerely thank reviewer #1 for recognizing the significance of our work. The reviewer's suggestion is very valuable for us to improve our manuscript.

However, concerns remain for the current version of the manuscript.

Comment #1 • In response to my questions about data in the previous version, several times authors claim they “made a mistake” (without stating the nature of the mistake) and have now replaced the old data with new ones – showing what I indicated would have been expected, but without explaining what caused the “mistakes”. (See Fig. 1, Fig. 2, and Fig. 3.) This does not bolster overall confidence in the rigor and/or truthfulness of the study.

REPLY: Thank the reviewer for this great point! We apologize for this confusion. We totally agree that we should state the nature of the mistake and explain what caused the “mistakes” in the previous Fig. 1b, Fig. 2f, and Fig. 3a. The nature of the mistake is that our analysis is not comprehensive enough in the previous version of our manuscript. The not-comprehensive-enough analysis caused the “mistakes” in the previous Fig. 1b, Fig. 2f, and Fig. 3a. Therefore, we have replaced the old data with new ones with a comprehensive enough analysis (current Fig. 2a, Fig. 3f, and Fig. 4a). This replacement does not change our conclusions that accumulation of PrP^C was not observed in the skeletal muscle of these four controls and excess WT PrP^C located in the cytoplasm significantly inhibited and blocked muscle cell differentiation when C2C12 myoblasts stably expressing PrP^C were incubated with differentiation medium for 3 days. To address the concern from reviewer #1 and to bolster overall confidence in the rigor and/or truthfulness of our study, we have now reworded and added the following sentences into the revised manuscript. ... *and the control samples only had normal levels of PrP^C (red) in most fields of view (Fig. 2a) (Lines 97-98). Please also see the following Fig. S1. We have replaced the second row for Day 3 in panel f with a correct version of WT PrP^C, in which WT PrP^C-expressing cells did not have any MyHC signal at day 3 in most fields of view (Lines 898-900, Legend of Fig. 3). Please also see the following Fig. S2. We have replaced the first row in panel a with a correct version of control, in which LC3B did not overlap with PrP^C*

in most fields of view (Lines 910-911, Legend of Fig. 4). Please also see the following Fig. S3.

Fig. S1 | Accumulation of PrP^C was not observed in the skeletal muscle of these four controls and the control samples only had normal levels of PrP^C (red) in most fields of view. Scale bar, 10 μ m.

Fig. S2 | WT PrP^C-expressing cells did not have any MyHC signal at day 3 in most fields of view. Scale bar, 75 μ m.

Fig. S3 | LC3B did not overlap with PrP^C in most fields of view. Scale bar, 7.5 μ m.

Comment #2 • Why does the DAPI staining vary so much from sample to sample in Figs. 1A and 2A? All the samples are muscle sections, and yet some images had numerous and strong DAPI signals and others had very few.

REPLY: Thank for pointing this out. The major reason why the DAPI staining could vary so much from sample to sample in Figs. 1a and 2a is that the nuclei of muscle bundles are not evenly distributed in the periphery of the muscle bundles. First, the number of nuclei in different frozen sections from the same skeletal muscle sample could be different, for example the number of nuclei in the first row of Case 1 in Fig. 1a is much different from that in the second row. Second, the number of nuclei in frozen sections from different skeletal muscle samples could be different, for example the images of Cases 4 and 5 in Fig. 1a, had numerous and strong DAPI signals, and others, for example the images of Cases 2 and 3, had very few.

Comment #3 • To address my question #3 regarding the previous Fig. 3, authors used rapamycin to induce autophagy (current Fig. 4). However, rapamycin does not seem to be an appropriate tool here because rapamycin is well documented to inhibit C2C12 differentiation through various mTOR mechanisms that are independent of autophagy.

REPLY: We sincerely thank reviewer #1 for his (her) expert suggestion that rapamycin does not seem to be an appropriate tool here because rapamycin is well documented to inhibit C2C12 differentiation through various mTOR mechanisms that are independent of autophagy!! According to the advice of reviewer #1, we have now used trehalose, instead of rapamycin, as an autophagy enhancer and have updated autophagy data for the effect of PrP^C overexpression on differentiation (current Extended Data Fig. 1, page 83). We have revised and added the following sentences into the Results section of the revision. *Rapamycin, a widely used autophagy enhancer, does not seem to be an appropriate tool here because rapamycin is well documented to inhibit C2C12 cell differentiation through various mammalian (or mechanistic) target of rapamycin (mTOR) mechanisms that are independent of autophagy^{61,62}. Instead, trehalose, a mTOR-independent autophagy enhancer^{63,64}, was used to induce ATG5-dependent and LC3B-dependent autophagy in differentiating C2C12 cells stably expressing WT PrP^C (Extended Data Fig. 1a,d,e). Overexpression of PrP^C significantly inhibited skeletal muscle cell autophagy and blocked myoblast differentiation (Extended Data Fig. 1a–e). Compared with C2C12 myoblasts (control), incubation of C2C12 myoblasts stably expressing WT PrP^C with 50 mM trehalose restored their autophagic activity and thus partially restored muscle cell differentiation (Extended Data Fig. 1a–e). Therefore, the inhibitory effect of excess PrP^C on autophagy in skeletal muscle partially results in an inhibitory effect of excess PrP^C on myoblast differentiation. Compared with C2C12 myoblasts (control), incubation of C2C12 myoblasts stably expressing WT PrP^C with 100 mM trehalose significantly increased their autophagic activity and thus blocked muscle cell differentiation (Extended Data Fig. 1a–e) (Lines 197-212). For analysis by western blotting treated with trehalose, C2C12 cells were grown in a 6-well plate and cultured in a differentiation medium of 2% equine serum supplemented with 50 or 100 mM D-(+)-trehalose (MedChem Express, HY-N1132) for 5 days, and the medium was changed daily (Lines 1158-1162). Accordingly, four related publications (Refs. 61-64) have been added into the revision.*

Extended Data Fig. 1 | Autophagy is partially responsible for the effect of overexpression of PrP^C on myoblast differentiation. **a**, Western blot for PrP^C, the myogenic differentiation markers MyHC and MyoG, and the autophagy markers ATG5 and LC3B during C2C12 mouse myoblasts (control) and C2C12 myoblasts stably expressing full-length wild-type mouse PrP^C (WT PrP^C) cultured until their confluence reached 90% and then incubated with 50 or 100 mM trehalose and differentiation medium for 5 days. β-actin served as the protein loading control. **b–e**, The relative amount of MyHC (**b**), MyoG (**c**), ATG5 (**d**) or LC3B-II (**e**) in the above cell lines (solid black circles shown in scatter plots) was expressed as the mean ± S.D. (with error bars) of values obtained in three independent experiments. One-way ANOVA and multiple comparisons were performed by SPSS 19.0 and different letters indicate significant differences at the level of $p < 0.05$.

Comment #4 • Regarding my point #3 on previous Fig. 5, “The effect of miR inhibitor is there with either KO or PrP overexpression. Would it suggest that the miRNA actions are independent of their binding to PrP?”, the authors responded by simply not mentioning the PrP KO data. Therefore, my original question is not addressed.

REPLY: Thank the reviewers #1’s comments regarding his (her) point #3 on previous Fig. 5. To address the original question from reviewer #1, we have now revised and added the following sentences into the Results section of the revision.

Moreover, PrP^C overexpression enhanced the stability of miR-204-5p and to a lesser degree for miR-214-3p, and the inhibitory effect of these miRNAs on their target proteins (ATG5 for miR-214-3p and LC3B for miR-204-5p) under PrP^C overexpression conditions is significantly stronger than that of the KO group, suggesting that the miRNA actions are enhanced by their binding to PrP^C. However, because the effect of miRNA inhibitor is there with either KO or PrP^C overexpression, the possibility that the miRNA actions are not dependent on their binding to PrP^C cannot be excluded (Lines 353-360).

Comment #5 • “It is unclear how the LLPS of PrP^C (Fig. 7) came into the picture after we found effects of PrP^C-associated miRNAs on autophagy (Fig. 6).” – this sentence should be deleted from the manuscript. The authors are repeating my previous question, which was meant to ask for rationale for the transition from Fig. 6 to experiments in Fig. 7.

REPLY: We sincerely thank reviewer #1 for this important suggestion that we should provide a rationale for the transition from Fig. 6 to experiments in Fig. 7!! According to the advice of the reviewer, we have now deleted the sentence “It is unclear how the LLPS of PrP^C (Fig. 7) came into the picture after we found effects of PrP^C-associated miRNAs on autophagy (Fig. 6)” from the manuscript (Line 449) and provided a rationale for the transition from Fig. 6 to experiments in Fig. 7 (current Fig. 8, page 55). The main rationale for the transition from Fig. 6 to experiments in Fig. 7 is that PrP^C recruits miR-214-3p into its phase-separated condensate in living skeletal muscle cells, which results in the inhibition of autophagy (Lines 448-449). We have now revised and added the following sentence into the Results section of the revision. *Given that PrP^C specifically interacts with a subset of miRNAs, such as miR-214-3p and miR-204-5p, during myoblast differentiation (Fig. 5), and the miRNA actions are enhanced by their binding to PrP^C (Fig. 6), we predicted that miR-214-3p and miR-204-5p might regulate the LLPS of PrP^C via interaction with the protein (Lines 398-402).*

Fig. 8 | PrP^C recruits miR-214-3p into its phase-separated condensate in living skeletal muscle cells, which results in the inhibition of autophagy. **a**, Schematic of the optoIDR assay, depicting recombinant protein with an IDR (blue), mCherry (red), and Cry2 (orange) expressed in cells exposed to 488-nm excitation laser. **b,c**, Time-lapse images of living C2C12 cells expressing mCherry-Cry2-WT PrP^C construct containing PrP₁₋₃₇ IDR (residues 1–37) (blue) linked to mCherry (red) and Cry2 (orange) and then linked to PrP₃₈₋₂₃₀ IDR (residues 38–230) (blue) (**c**), and mCherry-Cry2 fusion alone was used as a control (**b**). **d,e**, Time-lapse images (0–100 s) of a C2C12 cell expressing the mCherry-Cry2-WT PrP^C construct. A droplet fusion event occurs in the region highlighted by the orange box (**d**). The droplet fusion event highlighted by using orange arrows (**e**). **b–e**, Cells were subjected to laser excitation every 2 s for the indicated time. Scale bars, 7.5 μm. **f**, Representative live-cell images of PrP^C-miRNA speck formation when 10 μM FAM-labeled miRNA (green) was transfected into C2C12 cells stably expressing mCherry (red) or WT PrP^C-mCherry (red), among ≥ 10 cells. Scale bars, 7.5 μm. **g,h**, FRAP analysis on the selected liquid droplets of PrP^C (red) in living C2C12 cells expressing mCherry-Cry2-WT PrP^C

transfected without (**g**) or with (**h**) miRNA before (prebleach), during (0 s), and after photobleaching (60 and 125 s, respectively). The dashed white circle highlights the punctum undergoing targeted bleaching. Scale bars, 7.5 μm . **i**, Quantification of FRAP data for WT PrP^C puncta without or with miRNA ($n = 3$), as in **g,h**. Time 0 indicates the start of recovery after photobleaching. **j**, Normalized kinetics of fluorescence recovery data of WT PrP^C puncta (blue circle) and WT PrP^C puncta + miR-214-3p (red square). The normalized fluorescence intensity is expressed as the mean \pm S.D. (with error bars) of values obtained in three independent experiments. The solid lines show the best single exponential fit for the fluorescence intensity-time curves. **k**, Time-lapse images (0–300 s) of living C2C12 cells expressing mCherry-Cry2-WT PrP^C treated with 2.5% 1,6-hexanediol. Scale bar, 5 μm . **l**, Western blot for the autophagy markers ATG5 and LC3B in C2C12 cells expressing mCherry-Cry2-WT PrP^C transfected without (-) or with (+) 10 μM miRNA and then treated with (+) or without (-) 1,6-hexanediol. β -actin served as the protein loading control. **m,n**, The relative amount of ATG5 (**m**) and LC3B-II (**n**) in the above cell lines (solid black circles shown in scatter plots) was expressed as the mean \pm S.D. (with error bars) of values obtained in three independent experiments. One-way ANOVA and multiple comparisons were performed by SPSS 19.0 and different letters indicate significant differences at the level of $p < 0.05$.

Comment #6 • The authors added new data (Figs. 8 and 9) to address the role of LLPS. I do not have sufficient expertise to evaluate these experiments.

REPLY: Yes, we have added new data (Figs. 8 and 9) to address the role of LLPS, as followed the advice of reviewers #1 and #2. The main purpose of these data is to provide the FUNCTIONAL relevance of all the observations on LLPS of PrP^C. Thank you!

Reviewer #2

Remarks to the Author:

The authors have adequately addressed my main concerns, including conducting the recommended additional experiments with satisfactory results. The manuscript has been significantly improved. The writing can still benefit from further professional editing.

We sincerely thank reviewer #2 for recognizing the significance of our work. The reviewer's suggestion is very valuable for us to improve our manuscript.

REVIEWERS' COMMENTS

Reviewer #1 (Remarks to the Author):

The authors have addressed my previous concerns satisfactorily.